# GOTabPFN: From Feature Ordering to Compact Tokenization for Tabular Foundation Models on High-Dimensional Data

**Al Zadid Sultan Bin Habib** [1]  **Md Younus Ahamed** [1]  **Prashnna Kumar Gyawali** [1]  **Gianfranco Doretto** [2]
**Donald A. Adjeroh** [1]

## Abstract

We investigate how to make small tabular foundation models effective for High-Dimensional, Low-Sample Size (HDLSS) tabular prediction without retraining large backbones. We introduce Graph-guided Ordering with Local Refinement (GO-LR), show its equivalence to weighted Minimum Linear Arrangement, and interpret the practical solver as a TSP-path-style surrogate. We propose GOTabPFN, which builds on GO-LR, and a Neuro-Inspired Subunit Compression (NSC) unit to pool locally adjacent ordered features into meta-features, yielding a compact representation that makes TabPFN-style prediction practical in HDLSS regimes. Across tabular benchmarks, GOTabPFN improves stability and accuracy under tight token budgets.

## 1. Introduction

High-Dimensional, Low-Sample Size (HDLSS) tabular prediction remains a challenge: when $m \gg n$ (with $m$=no. of features, $n$=no. of samples), both learning and representation become costly. Tabular foundation models such as TabPFN and its variants are strong general-purpose baselines, but popular versions (e.g., TabPFN-2.5 (Grinsztajn et al., 2025)) are designed and benchmarked for inputs with up to roughly 2,000 features, leaving many HDLSS domains (e.g., gene expression with $m \gg 2{,}000$) outside their intended operating range without prior feature selection or compression. This motivates representation strategies that reduce dimensionality under tight sample budgets while preserving predictive structure, so TabPFN-style learners remain effective in truly high-dimensional regimes.

Permutation learning seeks an ordering of a finite set that improves a downstream objective, typically via differentiable relaxations that approximate discrete permutations in end-to-end neural training (Barthel et al., 2025; Jurewicz & Derczynski, 2022). For tabular data, the lack of inherent spatial or temporal structure weakens inductive bias relative to vision or language, especially in HDLSS settings. Although tree-based methods remain strong baselines, learning cross-feature dependencies without overfitting is difficult; even simple models (e.g., MLPs or Lasso) can outperform advanced tabular approaches in $n \ll m$ regimes (ProtoGate (Jiang et al., 2024)). This suggests that feature selection alone is often insufficient; we also need a learnable feature ordering that organizes correlated features into neighborhoods amenable to structured compression. We therefore formulate the Column Permutation Problem (CPP) (Fogel et al., 2013; Lima et al., 2024; Tegze & Vlach, 1986; Liiv, 2010; Behrisch et al., 2016): learn a data-driven column order that reduces redundancy, reveals long-range dependencies, and induces a useful sequential structure for downstream modules. In practice, CPP can be tackled via attention-based pointer mechanisms and graph-aware variants that generate permutations while encoding relational structure (Vinyals et al., 2015; Yang et al., 2022b; Veličković et al., 2020).

Feature ordering has a long history in pattern recognition and is central to Incremental Attribute Learning (IAL), where features arrive sequentially and must be ranked before training (Wang & Guan, 2013). Unlike set-based models that assume order invariance (Zaheer et al., 2017), column order can expose redundancy and shape how models capture dependencies; even simple Fisher/correlation/entropy rankings reduce interference and error over unordered baselines (Wang et al., 2015c;b), motivating learned, task-aware ordering (Wang et al., 2015a; 2014). In deep tabular learning, Mambular (Thielmann et al., 2024) underscored the impact of ordering and Habib et al. (2024; 2026b) introduced explicit ordering algorithms in TabSeq and DynaTab, respectively. Other related efforts show brittleness to col-

---

[1] Lane Department of Computer Science and Electrical Engineering, West Virginia University, Morgantown, WV 26506, USA
[2] Scientific Computing and Imaging Institute & Department of Biomedical Informatics, The University of Utah, Salt Lake City, UT 84112, USA . Correspondence to: Al Zadid Sultan Bin Habib <ah00069@mix.wvu.edu>.

*Proceedings of the 43rd International Conference on Machine Learning*, Seoul, South Korea. PMLR 306, 2026. Copyright 2026 by the author(s).

umn permutations, prompting permutation-invariant architectures (Eremeev et al., 2025; Brahmavar et al., 2025) and TabICL (Jingang et al., 2025), which ensembles across permutations. Beyond supervised prediction, COPER (Eisenberg et al., 2025) uses a permutation-based correlation objective for multi-view (image-table) clustering, and ROTATOR-LLM (Wang et al., 2025) studies feature ordering for LLM-based tabular inference.

While ordering can expose local structure, HDLSS tables introduce a second bottleneck: even a "good" permutation still leaves $m$ raw features to process, which is prohibitive when $m \gg n$. To make TabPFN-style predictors practical in this regime without changing the backbone, we introduce Neuro-Inspired Subunit Compression (NSC), motivated by subunit-style integration in cortical dendrites (Poirazi et al., 2003; Schiller et al., 2000; Major et al., 2013; Kastellakis et al., 2015; Kirchner & Gjorgjieva, 2021; Ujfalussy & Makara, 2020; Wu et al., 2018). NSC groups adjacent features along the GO-LR (Graph-guided Ordering with Local Refinement) axis into contiguous subunits and pools each into a meta-feature, reducing dimensionality from $m$ to $M$ ($M \ll m$), with $M$ tied to intrinsic dimension estimates from the covariance spectrum (Roy & Vetterli, 2007; Halko et al., 2011; Levina & Bickel, 2004). Naïve compression often produces latent components without a stable coordinate system, yielding run and subsample-dependent representations that are not effective for TabPFN-style models, which assume a fixed, consistently parameterized input space (Hollmann et al., 2023; 2025). We therefore design a structure-constrained compression interface that yields reproducible latent features within the feature budgets targeted by recent TabPFN variants (Grinsztajn et al., 2025; Liu & Ye, 2025; Kolberg et al., 2025).

**Our contributions:**

- We cast feature ordering as a combinatorial optimization problem, prove its NP-hardness, and propose MinLA-grounded ordering via GO-LR.

- We introduce scalable HDLSS compression via NSC, a neuro-inspired subunit-style pooling that is controlled by intrinsic-dimension estimates.

- Building on the above, we propose GOTabPFN for analyzing HDLSS tabular data. Across HDLSS benchmarks, GOTabPFN improves accuracy and stability under tight feature budgets in high-dimensions.

## 2. Related Work

In Appendix A, we provide more details on related work, including on tabular foundation models, the TabPFN family, HDLSS-specific models, and LLM-based tabular models.

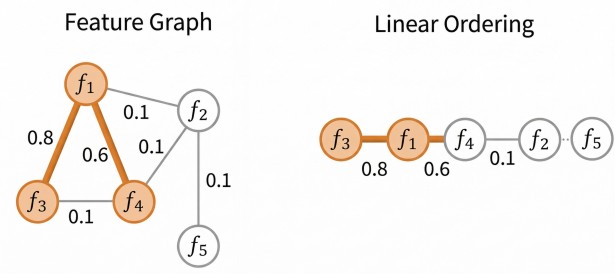

**Goal:** Place connected nodes on a line such that the most strongly related nodes are close to each other

*Figure 1.* **Graph-based feature ordering.** GO-LR linearizes a weighted feature graph to keep related features nearby for local segmentation and compression. It uses NNPath for local initialization, then refines the order with a global MinLA-style objective over pairwise placements. See Appendix T for more clarifications.

Existing approaches often struggle in HDLSS settings with $m \gg n$, since they either assume moderate feature counts or rely primarily on feature selection and task-specific tuning to cope with very high dimensionality. GOTabPFN bridges this gap by coupling MinLA-grounded ordering (GO-LR) with subunit-style compression (NSC), yielding stable, low-dimensional representations that enable TabPFN-style predictors to operate effectively in truly high-dimensional regimes without modifying the TabPFN backbone.

## 3. Methodology

**Problem formulation.** Let $X \in \mathbb{R}^{n \times m}$ be the input matrix with $n$ samples and $m$ features. We define the sample partition $\{I_c\}_{c=1}^k$ obtained by clustering the samples, and the cluster-restricted matrices in Eq. 1.

$$X^{(c)} = X[I_c, :] \in \mathbb{R}^{n_c \times m}, \qquad n_c = |I_c| \qquad (1)$$

For each $X^{(c)}$, we construct the corresponding cluster-wise feature graph $G_c = (V, E, w^{(c)})$, where $V = \{1, \ldots, m\}$ is the shared feature set and $w_{ij}^{(c)}$ measures feature dissimilarity within cluster $c$. The local permutation $\pi_c$ is obtained by minimizing a MinLA-style dispersion objective on $G_c$, and the final global permutation $\Pi^*$ is obtained by aggregating local ranks across clusters. All permutations are over features, and GO-LR outputs a single global feature ordering $\Pi^*$, not separate feature spaces that must later be rearranged across clusters. $\Pi^*$ is then used for NSC segmentation and compression. Figs. 1, 2, and 3 summarize the pipeline: GO-LR linearizes feature graphs, NSC segments and compresses contiguous ordered neighborhoods into meta-features, and the resulting tokens are passed to a frozen TabPFN-2.5 head within GOTabPFN.

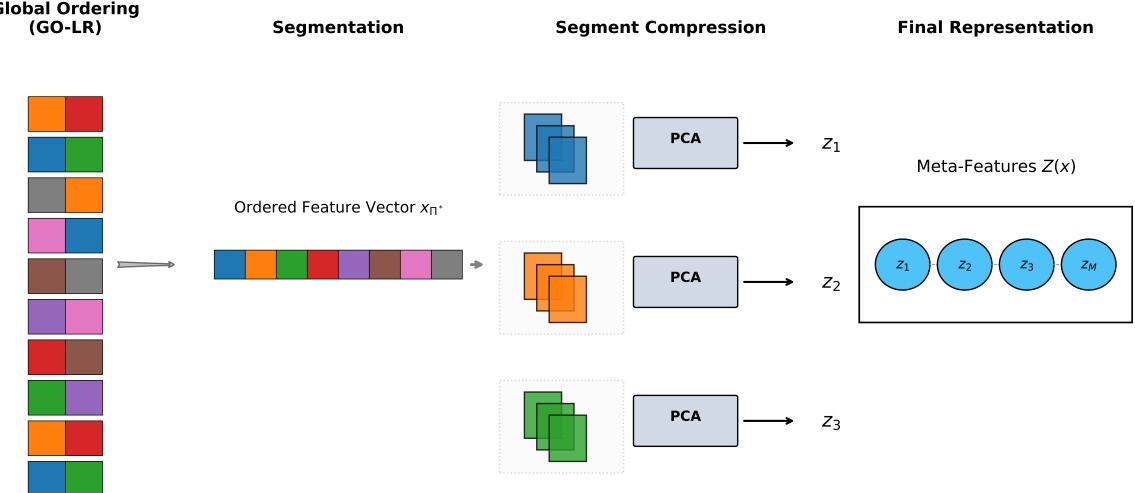

*Figure 2.* **Meta-feature construction.** GO-LR first orders features globally; NSC then segments the ordered axis into contiguous neighborhoods and compresses each segment by PCA into a scalar meta-feature. The final vector $Z(x) = (z_1, \ldots, z_M)$ is passed to the frozen TabPFN-2.5 head. See Appendix T for additional clarifications.

### 3.1. Feature Ordering as a Combinatorial Optimization Problem.

**Problem Setup: Feature Ordering by Graph Dispersion.** In this section, we show that GO-LR-based feature ordering corresponds to the Minimum Linear Arrangement (MinLA) problem, is NP-hard, and strictly generalizes TSP-path. Here, TSP-path refers to the Traveling Salesman (TSP) path problem: given a complete weighted graph, find a Hamiltonian path $\sigma$ that minimizes $\mathrm{PathCost}(\sigma) = \sum_{t=1}^{m-1} d_{\sigma_t, \sigma_{t+1}}$. We further show that the practical GO-LR algorithm provides a TSP-path style initialization, which is then locally refined under the dispersion objective. We connect GO-LR-based feature ordering to classical combinatorial optimization, including linear arrangement and seriation problems (Díaz et al., 2002; Seminaroti, 2016; Fogel et al., 2013). It is MinLA (NP-hard), admits a TSP-path heuristic implementation, and strictly generalizes TSP-path via an exact embedding.

**Theorem 3.1** (Theoretical Characterization of GO-LR). *GO-LR-based feature ordering corresponds to a weighted MinLA problem, is NP-hard in the number of features, and strictly generalizes the TSP-path problem.*

*Proof sketch.* Theorem follows from Lemma 3.8, Lemma 3.9, and Theorem 3.12 as described below. □

Moreover, the practical GO-LR algorithm uses a nearest-neighbor TSP-path heuristic for initialization and then applies a local refinement step (direction selection and adjacent swaps) that monotonically decreases the MinLA dispersion objective. The remainder of this section establishes this characterization through a sequence of equivalence and reduction results.

**Definition 3.2** (Local Feature Graph). Given cluster $c$ with samples $X^{(c)} \in \mathbb{R}^{n_c \times m}$, we define a weighted feature graph $G_c = (V, E, w)$ where $V = \{1, \ldots, m\}$ indexes features and $w_{ij} \geq 0$ quantifies dissimilarity between features $i$ and $j$ computed from $X^{(c)}$ (e.g., $1 - |\mathrm{corr}|$; see App. T, JS(Jensen-Shannon divergence)/KL(Kullback-Leibler divergence), cosine/Euclidean/Manhattan). We write $(i, j) \in E$ whenever a pair is included (typically $E = V \times V \setminus \{(i, i)\}$ for a complete graph, or a sparse neighborhood graph).

**Definition 3.3** (Dispersion Objective (GO-LR Local Ordering)). A local ordering is a bijection $\pi : V \to \{0, \ldots, m-1\}$ assigning each feature to a position. The cluster-wise dispersion of $\pi$ is in Eq. 2. The GO-LR local ordering problem is to compute the local order with minimum dispersion, as shown in Eq. 3.

$$D_{G_c}(\pi) = \sum_{(i,j) \in E} w_{ij} |\pi(i) - \pi(j)| \tag{2}$$

$$\pi_c^* \in \arg\min_\pi D_{G_c}(\pi) \tag{3}$$

**Definition 3.4** (GO-LR Local Refinement Operator). Let $\pi^{(0)} \leftarrow \mathrm{NNPath}(G_c)$ be the nearest-neighbor initialization (a permutation of $V$). We define $\mathrm{rev}(\pi)$ as the reversed permutation and let $\mathcal{N}(\pi)$ denote the set of permutations obtained by one adjacent transposition (Eq. 4). GO-LR first performs direction selection (Eq. 5) and then applies $P$ passes of adjacent-swap descent (Eq. 6), with early stopping if $\pi^{(p+1)} = \pi^{(p)}$. The refined local ordering is $\pi_c \leftarrow \pi^{(P)}$. In Eq. 4, $\mathrm{swap}_t(\pi)$ is the adjacent-transposition operator that returns the permutation obtained by swapping the entries at positions $t$ and $t + 1$ in $\pi$.

$$\mathcal{N}(\pi) = \{\text{swap}_t(\pi) : t = 0, \ldots, m - 2\} \qquad (4)$$

$$\pi^{(0)} \leftarrow \arg\min \left\{ D_{G_c}(\pi^{(0)}), D_{G_c}(\text{rev}(\pi^{(0)})) \right\} \quad (5)$$

$$\pi^{(p+1)} \leftarrow \text{SweepRefine}\left(\pi^{(p)}, G_c\right), \qquad p = 0, \ldots, P-1 \tag{6}$$

**SweepRefine.** Initialize $\tilde{\pi} \leftarrow \pi^{(p)}$ and scan $t = 0, \ldots, m - 2$. Compute swap gain $\Delta_t := D_{G_c}(\text{swap}_t(\tilde{\pi})) - D_{G_c}(\tilde{\pi})$ via an incremental update (no full recomputation). If $\Delta_t < 0$, set $\tilde{\pi} \leftarrow \text{swap}_t(\tilde{\pi})$ immediately. Return $\pi^{(p+1)} \leftarrow \tilde{\pi}$ and early-stop if a full sweep makes no changes.

*Remark* 3.5. Each update in Eqs. (5)–(6) is chosen to not increase $D_{G_c}$; hence GO-LR yields $D_{G_c}(\pi_c) \leq D_{G_c}(\pi^{(0)})$.

*Remark* 3.6. Eq. (2) (together with Eq. (3)) defines the dispersion objective used in GO-LR. It is a standard linear arrangement / seriation-type criterion (Díaz et al., 2002; Fogel et al., 2013): pairs with larger weights $w_{ij}$ are penalized more when placed far apart, hence the ordering tends to place high-weight pairs closer in their index.

**Complexity: Equivalence to Minimum Linear Arrangement.** The MinLA problem is a classical graph layout problem, and has been well studied (Shiloach, 1979).

**Definition 3.7** (Weighted Minimum Linear Arrangement (MinLA)). Given a weighted graph $G = (V, E, w)$, the weighted MinLA problem is

$$\min_{\pi:V \to \{0,\ldots,|V|-1\} \text{ bijective}} \sum_{(i,j)\in E} w_{ij} |\pi(i) - \pi(j)| \quad (7)$$

**Lemma 3.8** (GO-LR Local Ordering is MinLA). *For each cluster $c$, the GO-LR local ordering objective in Eq. (3) is exactly the weighted MinLA objective on $G_c$.*

*Proof sketch.* Both problems optimize over bijections $\pi : V \to \{0, \ldots, m - 1\}$ and share the identical objective $\sum_{(i,j)\in E} w_{ij}|\pi(i) - \pi(j)|$ (Eq. (2) and Eq. (7)). Hence they are the same optimization problem. $\square$

**Lemma 3.9** (NP-hardness). *The GO-LR local feature ordering problem (Eq. (3)) is NP-hard in $m$.*

*Proof sketch.* Weighted MinLA is NP-hard (Garey et al., 1976); since GO-LR local ordering is exactly MinLA, it is NP-hard. $\square$

**An Exact Equivalence Case: TSP-path as a Special Case of Feature Ordering.**

**Definition 3.10** (TSP-path Objective on a Complete Graph). Given a complete weighted graph $\mathcal{K} = (V, \binom{V}{2}, d)$ with

edge weights $d_{ij} \geq 0$, define the path cost of a permutation $\sigma = (\sigma_1, \ldots, \sigma_m)$ by

$$\text{PathCost}(\sigma) = \sum_{t=1}^{m-1} d_{\sigma_t, \sigma_{t+1}} \tag{8}$$

First, we establish GO-LR as a TSP-path heuristic that outputs a Hamiltonian path. (see Appendix B). Then, below, we show the connection between our feature ordering and TSP-path.

The GO-LR objective in Eq. (2) is a general seriation / linear arrangement criterion. We now exhibit a non-circular special case in which Feature Ordering becomes exactly the TSP-path problem, implying that Feature Ordering strictly generalizes TSP-path (Carmona et al., 2023).

**Definition 3.11** (Path-Edge Feature Ordering (Adjacency-by-Position)). Fix $m$ and define the path-edge set on positions

$$E_{\text{path}} = \{(t, t + 1) : t = 1, \ldots, m - 1\} \tag{9}$$

Given a complete weighted graph $\mathcal{K} = (V, \binom{V}{2}, d)$ with $|V| = m$, a permutation $\sigma = (\sigma_1, \ldots, \sigma_m)$ induces an ordering map $\pi_\sigma : V \to \{1, \ldots, m\}$ via $\pi_\sigma(\sigma_t) = t$. We define the path-edge feature ordering objective:

$$D_{\text{path}}(\pi_\sigma) = \sum_{t=1}^{m-1} d_{\sigma_t, \sigma_{t+1}} \tag{10}$$

This is equivalent to restricting Eq. (2) to adjacency-by-position interactions.

**Theorem 3.12** (Exact Equivalence to TSP-path). *Minimizing the path-edge feature ordering objective in Eq. (10) over all permutations $\sigma$ is exactly the TSP-path problem on $\mathcal{K}$ with path cost given by $\text{PathCost}(\sigma)$ in Eq. (8).*

*Proof sketch.* For any permutation $\sigma$, Eq. (10) can be rewritten as $\sum_{t=1}^{m-1} d_{\sigma_t, \sigma_{t+1}} = \text{PathCost}(\sigma)$ by definition. Thus, the minimizers coincide. $\square$

**Corollary 3.13** (TSP-path embeds into Feature Ordering). *TSP-path is a special case of feature ordering. Consequently, feature ordering (strictly) generalizes TSP-path.*

*Remark* 3.14. This equivalence holds for the path-edge special case. In GO-LR, the practical objective remains the full dispersion in Eq. (2) (MinLA), while the nearest-neighbor constructor corresponds to a TSP-path heuristic for initialization, and GO-LR then applies local refinement under the full dispersion objective in Eq. (2) on a complete graph built from the chosen dissimilarity metric.

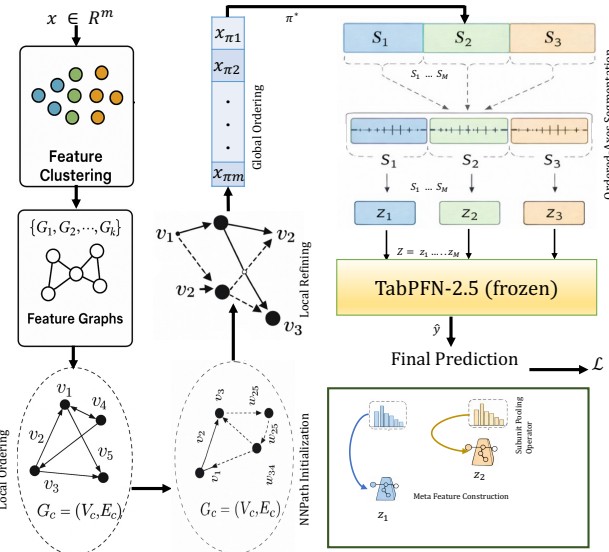

*Figure 3.* **End-to-end architecture of GOTabPFN.** The feature clustering block denotes the discovery of local feature-dependence groups, implemented by estimating cluster-wise feature graphs $G_c$ from local sample contexts; GO-LR then obtains a global order $\Pi^*$, and NSC compresses contiguous ordered segments into meta-features $Z(x)$, which are passed to a frozen TabPFN-2.5 head.

**Global Aggregation (Mean-Rank Integration).** Let $\pi_c$ be a local ordering for cluster $c$ and let $r_c(j)$ be the rank (position) of feature $j$ in $\pi_c$. With cluster weights $\alpha_c \geq 0$ and $\sum_{c=1}^{k} \alpha_c = 1$, GO-LR forms a global order by Eq. 11. This aggregation produces a single global permutation consistent with the set of local cluster-wise permutations.

$$\bar{r}(j) = \sum_{c=1}^{k} \alpha_c r_c(j), \quad \Pi^* = \underset{j=1}{\overset{m}{\operatorname{argsort}}} \, \bar{r}(j) \qquad (11)$$

Algorithm 1 captures the steps in the proposed GO-LR algorithm.

### 3.2. Neuro-Inspired Subunit Compression (NSC)

**Motivation.** We design a representation interface that allows TabPFN to scale to HDLSS tabular data without retraining or architectural modification. Cortical pyramidal neurons receive on the order of 20,000-30,000 synaptic inputs (Poirazi et al., 2003), yet these inputs are not integrated as a single linear sum (Major et al., 2013). Instead, inputs are organized into multiple dendritic subunits (Kastellakis et al., 2015), each acting as a nonlinear integration compartment. Here, correlated synapses may exhibit local clustering (Ujfalussy & Makara, 2020) and trigger N-methyl-D-aspartate (NMDA)-mediated plateau potentials (Schiller et al., 2000) that pool dozens to hundreds of inputs into a single subunit-level signal (Kirchner & Gjorgjieva, 2021). This subunit-based organization provides a canonical biological mechanism (Beniaguev et al., 2021) for compressing extremely high-dimensional inputs into a compact set of functional representations (Wu et al., 2018). This locality-

---

**Algorithm 1** Graph-guided Ordering with Local Refinement (GO-LR)

**Require:** $X \in \mathbb{R}^{n \times m}$, clusters $k$, metric $\phi$, passes $P$
**Ensure:** Global order $\Pi^*$ and local orders $\{\pi_c\}_{c=1}^{k}$
1: $\{X^{(c)}\}_{c=1}^{k} \leftarrow \text{Cluster}(X, k)$; $\mu^{(c)} \leftarrow \text{mean}(X^{(c)})$
2: **for** $c = 1$ to $k$ **do**
3: $\quad G_c \leftarrow \text{Sym}(\text{FeatureDissimilarity}(X^{(c)}, \phi)) \in \mathbb{R}^{m \times m}$ // undirected
4: $\quad \pi_c \leftarrow \text{NNPath}(G_c)$
5: $\quad \pi_c \leftarrow \text{Refine}(\pi_c, G_c, P)$ // direction-select + $P$ passes
6: $\quad r_c(j) \leftarrow \text{rank}(j \text{ in } \pi_c)$ **for** $j = 1, \ldots, m$
7: **end for**
8: $\tilde{\alpha}_c \leftarrow \left(\varepsilon + \text{mean}_{c'} \|\mu^{(c)} - \mu^{(c')}\|_2\right)^{-1}$; $\alpha_c \leftarrow \tilde{\alpha}_c / \sum_{c'} \tilde{\alpha}_{c'}$
9: $\bar{r}(j) \leftarrow \sum_{c=1}^{k} \alpha_c r_c(j)$; $\Pi^* \leftarrow \text{argsort}_j \bar{r}(j)$
10: **return** Global feature order $\Pi^*$ and local orders $\{\pi_c\}_{c=1}^{k}$

---

driven compression view is also consistent with prior signal-compression work, where edge-aware prediction has been used to exploit local structure in high-dimensional hyperspectral imagery (Jain & Adjeroh, 2007). We adopt this principle as an algorithmic inductive bias for HDLSS tabular data (Balın et al., 2019). See Alg. 2 for steps in NSC.

**Ordered-Axis Segmentation.** Let $x \in \mathbb{R}^m$ denote a tabular sample and let $\Pi^*$ be the global feature permutation produced by GO-LR (Section 3.1). We define the reordered feature vector in Eq. 12. Given a target number of meta-features $M$, we set the segment length $s = \lceil m/M \rceil$ (Eq. 13) and define contiguous segments $\{\mathcal{S}_t\}_{t=1}^{M}$ by Eq. 14, which partition $\{1, \ldots, m\}$ into ordered neighborhoods (subunits).

$$x^{\Pi} = \left(x_{\Pi^*(1)}, x_{\Pi^*(2)}, \ldots, x_{\Pi^*(m)}\right) \qquad (12)$$

$$s = \left\lceil \frac{m}{M} \right\rceil \qquad (13)$$

$$\mathcal{S}_t = \{(t-1)s + 1, \ldots, \min(ts, m)\}, \qquad t = 1, \ldots, M \qquad (14)$$

**Adaptive segmentation.** To let segment boundaries follow "transitions" in the ordered feature axis, we first summarize pairwise dissimilarities into a 1D signal. We reuse the global feature dissimilarity matrix $\bar{W} \in \mathbb{R}^{m \times m}$ already computed for GO-LR on the dataset (and keep it fixed at inference), e.g., $\bar{W} := \text{FeatureDissimilarity}(X, \phi)$ or $\bar{W} := \sum_{c=1}^{k} \alpha_c W_c$, so NSC itself does not introduce any additional $O(m^2)$ cost. For adjacent positions along the GO-LR order, we define the transition dissimilarity $\delta_t := \bar{W}_{\Pi^*(t), \Pi^*(t+1)}$ for $t = 1, \ldots, m-1$; large $\delta_t$ indicates a sharp change between neighboring features. We then form the cumulative transition mass $c_t := \sum_{i=1}^{t-1} \delta_i$ for $t = 1, \ldots, m$ with total $C := c_m = \sum_{i=1}^{m-1} \delta_i$. Given a desired number of segments $M$, we place cutpoints

---

**Algorithm 2** Neuro-Inspired Subunit Compression (NSC)

**Require:** Training matrix $X_{\text{train}} \in \mathbb{R}^{n \times m}$, sample $x \in \mathbb{R}^m$, global order $\Pi^*$, ID threshold $\tau \in (0, 1)$, bypass threshold $m_0$, $M$-rule hyperparameters $(\gamma, M_{\min}, M_{\max})$, segmentation rule $\text{Seg}(\cdot)$ with $\ell_{\min}$, and (if transition-aware) dissimilarities $\Delta \in \mathbb{R}_+^{m-1}$.

**Ensure:** Compressed tokens $Z(x) \in \mathbb{R}^M$.

1: **Reorder:** $X^\Pi \leftarrow X_{\text{train}}[:, \Pi^*]$, $x^\Pi \leftarrow x[\Pi^*]$.
2: **PCA-ID:** compute $G = \frac{1}{n-1} X^\Pi (X^\Pi)^\top$; let $\{\lambda_i\}$ be eigenvalues of $G$ (descending).
3: $\hat{d} \leftarrow \min\{k : \sum_{i=1}^k \lambda_i \geq \tau \sum_i \lambda_i\}$;   $\text{IDF} \leftarrow \hat{d}/m$
4: **if** $m \leq m_0$ **then**
5:    $M \leftarrow m$
6: **else**
7:    $M \leftarrow \text{clip}(\lceil 2\hat{d} \rceil, M_{\min}, \min(M_{\max}, m))$     // or $\lceil \gamma \hat{d} \rceil$ / IDF-rule
8: **end if**
9: **Segment:** $\{\mathcal{S}_t\}_{t=1}^M \leftarrow \text{Seg}(m, M, \Delta, \ell_{\min})$ // uniform / equal-mass / largest-jump
10: **for** $t = 1$ to $M$ **do**
11:    **Fit (once):** center $X^\Pi_{[:, \mathcal{S}_t]}$ to get mean $\mu_t$; compute first PC direction $v_t$ (CPU SVD), fix sign deterministically.
12:    **Tokenize:** $z_t(x) \leftarrow (x^\Pi_{\mathcal{S}_t} - \mu_t)^\top v_t$     // scalar
13: **end for**
14: **return** $Z(x) \leftarrow (z_1(x), \ldots, z_M(x))$

---

$1 \leq \tau_1 < \cdots < \tau_{M-1} < m$ along this 1D signal in two ways: (i) a largest-jump rule that selects the indices of the $M-1$ largest $\delta_t$ values (subject to a minimum segment length $\ell_{\min}$), and (ii) an equal-mass rule that treats $c_t$ as a discrete CDF and chooses cutpoints by Eq. 15.

$$\tau_\ell := \min\left\{ t \in \{2, \ldots, m-1\} : c_t \geq (\ell/M)\, C \right\}, \quad (15)$$
$$\ell = 1, \ldots, M-1$$

Again enforcing $\ell_{\min}$ with a uniform fallback if needed. Finally, we materialize segments as $\mathcal{S}_1 = \{1, \ldots, \tau_1\}$, $\mathcal{S}_t = \{\tau_{t-1} + 1, \ldots, \tau_t\}$ for $t = 2, \ldots, M-1$, and $\mathcal{S}_M = \{\tau_{M-1} + 1, \ldots, m\}$, so that each subunit is an ordered neighborhood bounded by large transitions in the feature axis.

**Subunit Pooling and Meta-Feature Construction.**

**Segment descriptors.** Beyond mean and variance, we optionally summarize each segment $u_t$ using a richer descriptor $\psi(u_t)$ that includes higher-order and robust statistics (e.g., skewness, kurtosis, median, and interquartile range), enabling NSC to capture distributional shape within each ordered region at negligible extra cost.

Let $\psi : \mathbb{R}^{|\mathcal{S}_t|} \to \mathbb{R}^q$ denote a (possibly learn-free) segment descriptor of dimension $q$, and let $g_\theta : \mathbb{R}^q \to \mathbb{R}^d$ be a shared lightweight pooling network that maps each descriptor to a $d$-dimensional meta-feature. In practice, $g_\theta$

may be a shallow MLP (or linear map) applied to $\psi(u_t)$; the same $g_\theta$ is reused across segments to enforce parameter sharing and stability. The $t$-th meta-feature is defined in Eq. 16. NSC outputs the compressed meta-feature sequence $Z(x) = (z_1, \ldots, z_M)$, which is subsequently provided to the TabPFN predictor head.

$$z_t = g_\theta\Big(\psi(u_t)\Big), \qquad u_t := x^\Pi_{\mathcal{S}_t} \qquad (16)$$

$$Z(x) = (z_1, \ldots, z_M) \in \mathbb{R}^{M \times d} \qquad (17)$$

**NSC variants.** We instantiate NSC in four variants (details in Appendix D): (i) NSC: uniform segments + learned pooling, (ii) NSC-P: same with PCA-based intrinsic-dimension rule for $M$, (iii) NSC-SP: PCA-based segment (SegPCA) pooling with a fixed $M$, and (iv) NSC-pSP: PCA-based intrinsic-dimension rule for $M$ combined with SegPCA pooling. GOTabPFN uses NSC-pSP in the experiments.

**Choosing the Number of Meta-Features (NSC-pSP).** To adapt the compression level to dataset complexity, we tie the meta-feature budget $M$ to an estimate of the intrinsic dimensionality $\hat{d}$ of the training data. Let $\tilde{X} \in \mathbb{R}^{n \times m}$ denote the standardized training matrix (zero mean, unit variance per feature), and let $\Sigma = \frac{1}{n-1} \tilde{X}^\top \tilde{X}$ be its empirical covariance (or correlation) matrix with nonzero eigenvalues $\{\lambda_i\}_{i=1}^r$, $r \leq \min(n, m)$.

For the NSC-pSP variant used in our main experiments, we estimate $\hat{d}$ via a PCA cumulative-variance rule (Hotelling, 1933). We define the explained-variance ratio and its cumulative sum in Eqns. 18 and 19. Given a target variance-retention level $\tau \in (0, 1)$ (e.g., $\tau \in \{0.90, 0.95, 0.99, 0.9975\}$), the PCA-based intrinsic dimension is defined by Eq. 20 and NSC-pSP sets $\hat{d} = \hat{d}_{\text{PCA}}(\tau)$. We then choose the meta-feature budget via Eq. 21 where $\text{clip}(x, a, b) = \min(\max(x, a), b)$. For non-HDLSS regimes (e.g., $m \leq 400$), we bypass compression by setting $M = m$. This rule ensures that the number of meta-features scales with intrinsic, rather than ambient, dimensionality, yielding aggressive compression in highly redundant HDLSS settings while avoiding unnecessary bottlenecks when features are already compact. Implementation details and alternative intrinsic-dimension rules used by the other NSC variants are given in Appendix C.

$$\text{EVR}_i = \frac{\lambda_i}{\sum_{j=1}^r \lambda_j}, \qquad i = 1, \ldots, r \qquad (18)$$

$$\text{CUM}(k) = \sum_{i=1}^k \text{EVR}_i, \qquad k = 1, \ldots, r \qquad (19)$$

$$\hat{d}_{\text{PCA}}(\tau) = \min\Big\{ k \in \{1, \ldots, r\} : \text{CUM}(k) \geq \tau \Big\} \qquad (20)$$

$$M = \text{clip}\big(\lceil 2\hat{d}\rceil,\ 32,\ \min(512, m)\big) \qquad (21)$$

**PCA-centric post-segmentation pooling (SegPCA).** For the PCA-centric NSC variants (NSC-SP and NSC-pSP), once the meta-feature budget $M$ is determined (Sec. 3.2) and the ordered segments $\{\mathcal{S}_t\}_{t=1}^M$ are formed (Eqs. 13-14), we construct one scalar token per segment by projecting onto a segment specific first principal direction learned on the training set. Let $u_t(x) := x_{\mathcal{S}_t}^{\Pi} \in \mathbb{R}^{|\mathcal{S}_t|}$ and let $X^{\Pi} \in \mathbb{R}^{n \times m}$ denote the standardized training matrix after applying $\Pi^*$. We define the training submatrix for segment $t$ as $X_t := X_{:,\mathcal{S}_t}^{\Pi} \in \mathbb{R}^{n \times |\mathcal{S}_t|}$. We compute the segment mean and covariance by Eq. 22 and take the first principal direction using Eq. 23. The $t$-th meta-feature is then the centered projection (Eq. 24), yielding a $d{=}1$ token sequence $Z_{\text{SegPCA}}(x)$ (Eq. 25). Optionally, we apply a deterministic sign convention to $v_t$ (e.g., flipping $v_t$ so that segment scores positively correlate with a fixed reference such as the within-segment sample mean), which leaves the subspace unchanged but improves reproducibility.

$$\mu_t = \frac{1}{n}\sum_{i=1}^n X_{t,i:} \in \mathbb{R}^{|\mathcal{S}_t|},$$

$$\Sigma_t = \frac{1}{n-1}\big(X_t - \mathbf{1}\mu_t^\top\big)^\top\big(X_t - \mathbf{1}\mu_t^\top\big) \qquad (22)$$

$$v_t = \arg\max_{\|v\|_2=1} v^\top \Sigma_t v \in \mathbb{R}^{|\mathcal{S}_t|} \qquad (23)$$

$$z_t(x) = \big(u_t(x) - \mu_t\big)^\top v_t \in \mathbb{R}, \qquad t = 1,\dots,M \qquad (24)$$

$$Z_{\text{SegPCA}}(x) = (z_1(x),\dots,z_M(x)) \in \mathbb{R}^{M \times 1} \qquad (25)$$

**Summary.** NSC acts as a shared piecewise pooling operator, defined by Prop. C.1 (App. C). NSC transforms GO-LR-ordered high-dimensional tabular inputs into a compact sequence of structured meta-features through contiguous segmentation and shared pooling, introducing an HDLSS-friendly inductive bias inspired by subunit-based cortical computation while remaining purely statistical and computationally efficient. By compressing $m$ raw features into $M \ll m$ meta-features, NSC reduces effective sequence length presented to TabPFN-style backbones (e.g., TabPFN-2.5 (Grinsztajn et al., 2025) or other variants), yielding lower compute & memory cost while preserving order-induced locality to make original TabPFN versions usable for HDLSS regime. Per sample, NSC is $O(m)$ when $\psi$ uses linear-time statistics (e.g., moments); robust summaries such as quantiles can be computed approximately in linear time/exactly with a mild $O(|\mathcal{S}_t| \log |\mathcal{S}_t|)$ overhead if sorting is used.

**TabPFN-2.5 Head (non-differentiable).** NSC module

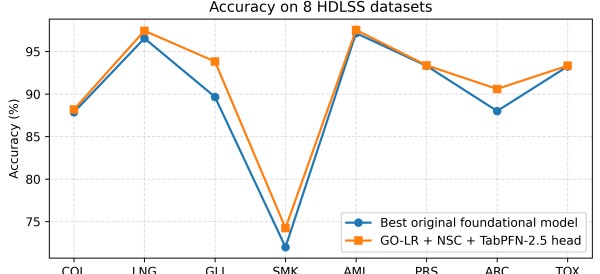

*Figure 4.* **HDLSS ablation accuracy.** GOTabPFN vs. tabular foundation models on 8 HDLSS datasets.

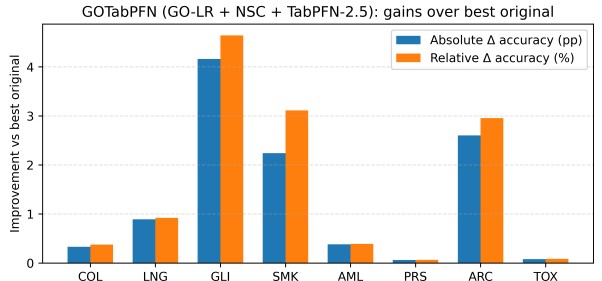

*Figure 5.* **Ablation gains.** Absolute and relative gains of GOTabPFN over the best original foundation-model head.

compresses each sample into a fixed-dimensional representation $Z(x) \in \mathbb{R}^M$ (or $Z(x) \in \mathbb{R}^{M \times d}$, flattened to $\mathbb{R}^{Md}$). We then use TabPFN-2.5 as the predictor head: for each train/validation split, we fit TabPFN-2.5 on $\{(Z(x_i), y_i)\}_{i \in \mathcal{I}_{\text{train}}}$ and evaluate on $Z(x_j)$ for $j \in \mathcal{I}_{\text{val}}$ without backpropagation through the head. This design treats NSC as a compression interface that maps HDLSS inputs into a feature budget compatible with TabPFN variants, while retaining strong tabular foundation models by Hollmann et al. (2023; 2025); Grinsztajn et al. (2025).

## 4. Experimental Results

**Algorithms and datasets used.** We use eight biomedical HDLSS datasets (e.g., Arcene, Colon, GLI-85, Lung, etc.) from the repository of Li et al. (2018), also used by Proto-Gate (Jiang et al., 2024). We follow the standard HDLSS setting where features far exceed samples ($m \gg n$). To study when feature ordering helps, we propose an empirical locality-based criterion; see Appendix F for the criterion and usage guidance. We compare against 55 baselines spanning classical ML/GBDT, HDLSS feature selection, deep tabular models, and small tabular foundation models. Modern methods including TANDEM (Naor & Lindenbaum, 2025), TabPFN Wide (Kolberg et al., 2025), TabDPT (Ma et al., 2025), TabICL (Jingang et al., 2025), BETA (Liu & Ye, 2025), TuneTables (Feuer et al., 2024), and ProtoGate perform strongly on HDLSS classification, while classical baselines (MLP, Lasso) remain competitive, consistent with ProtoGate. Full baseline details are in Appendix G.

**Experimental set up.** We use $5{\times}5$ nested cross-validation (25 repeats) on the HDLSS datasets, matching ProtoGate's protocol, for all baselines. Experiments run on the TI-

*Table 1.* Top-10 performance on 8 HDLSS datasets (mean accuracy with subscripted standard deviation over $5 \times 5$ CV). **Bold** denotes the best result per dataset and underline denotes the second-best. Rank is the average rank across datasets (lower is better), computed with standard tie-breaking. Dataset abbreviations: COL = Colon, LNG = Lung, GLI = GLI-85, SMK = SMK_CAN_187, AML = ALLAML, PRS = Prostate-GE, ARC = Arcene, TOX = TOX-171. Model abbreviations: *GOTabPFN = our method, TabPFN-W = TabPFN Wide, TTables = TuneTables, BETA = TabPFN Unleashed. See Table G.1 in Appendix G for full results against 55 baselines.

| MODEL/DB | COL | LNG | GLI | SMK | AML | PRS | ARC | TOX | RANK |
|---|---|---|---|---|---|---|---|---|---|
| #SAMPLES | 62 | 203 | 85 | 187 | 72 | 102 | 200 | 171 | – |
| #FEATURES | 2000 | 3312 | 22283 | 19993 | 7129 | 5966 | 10000 | 5748 | – |
| #CLASSES | 2 | 5 | 2 | 2 | 2 | 2 | 2 | 4 | – |
| *GOTabPFN | $\mathbf{88.18}_{\pm 10.05}$ | $\mathbf{97.44}_{\pm 2.32}$ | $\mathbf{93.82}_{\pm 5.81}$ | $\mathbf{74.23}_{\pm 5.17}$ | $\mathbf{97.54}_{\pm 3.86}$ | $\mathbf{93.37}_{\pm 4.48}$ | $\mathbf{90.60}_{\pm 3.97}$ | $\mathbf{93.33}_{\pm 4.74}$ | $\mathbf{1.00}_{\pm 0.00}$ |
| TANDEM | $86.15_{\pm 7.75}$ | $96.46_{\pm 2.88}$ | $\underline{91.53}_{\pm 6.02}$ | $\underline{72.72}_{\pm 5.69}$ | $95.81_{\pm 5.53}$ | $91.55_{\pm 4.32}$ | $86.90_{\pm 6.34}$ | $93.08_{\pm 2.61}$ | $\underline{3.63}_{\pm 1.32}$ |
| TABPFN-W | $\underline{87.85}_{\pm 7.28}$ | $\underline{96.55}_{\pm 2.15}$ | $88.47_{\pm 5.75}$ | $68.78_{\pm 8.60}$ | $\underline{97.16}_{\pm 4.10}$ | $93.10_{\pm 5.92}$ | $\underline{88.00}_{\pm 5.20}$ | $89.35_{\pm 4.95}$ | $3.75_{\pm 2.38}$ |
| TABDPT | $86.26_{\pm 7.27}$ | $96.05_{\pm 2.57}$ | $87.76_{\pm 5.86}$ | $71.99_{\pm 7.32}$ | $96.32_{\pm 4.15}$ | $90.94_{\pm 5.72}$ | $82.10_{\pm 6.48}$ | $\underline{93.25}_{\pm 3.44}$ | $4.88_{\pm 1.69}$ |
| TABICL | $84.62_{\pm 10.52}$ | $96.36_{\pm 2.61}$ | $87.06_{\pm 6.23}$ | $68.73_{\pm 7.28}$ | $95.52_{\pm 5.59}$ | $90.17_{\pm 5.93}$ | $82.60_{\pm 6.14}$ | $88.78_{\pm 5.92}$ | $7.63_{\pm 2.29}$ |
| BETA | $84.73_{\pm 9.36}$ | $94.38_{\pm 3.34}$ | $86.21_{\pm 8.91}$ | $70.21_{\pm 5.61}$ | $95.67_{\pm 7.54}$ | $87.53_{\pm 4.67}$ | $86.45_{\pm 5.92}$ | $90.38_{\pm 6.42}$ | $8.13_{\pm 4.31}$ |
| TTABLES | $86.80_{\pm 2.14}$ | $94.37_{\pm 2.35}$ | $89.66_{\pm 3.12}$ | $70.28_{\pm 6.46}$ | $95.80_{\pm 2.14}$ | $\underline{93.31}_{\pm 2.83}$ | $81.40_{\pm 3.66}$ | $77.96_{\pm 2.55}$ | $8.38_{\pm 7.70}$ |
| LASSO | $79.40_{\pm 10.18}$ | $94.47_{\pm 4.39}$ | $85.88_{\pm 4.71}$ | $61.19_{\pm 13.72}$ | $87.24_{\pm 3.39}$ | $91.18_{\pm 6.39}$ | $81.00_{\pm 3.39}$ | $91.86_{\pm 6.03}$ | $11.13_{\pm 5.06}$ |
| MLP | $83.95_{\pm 9.80}$ | $96.47_{\pm 2.69}$ | $85.41_{\pm 8.00}$ | $59.05_{\pm 7.44}$ | $89.98_{\pm 9.17}$ | $89.20_{\pm 6.07}$ | $78.40_{\pm 4.05}$ | $92.48_{\pm 4.28}$ | $11.63_{\pm 5.45}$ |
| PROTOGATE | $83.95_{\pm 9.82}$ | $93.44_{\pm 6.37}$ | $82.48_{\pm 5.68}$ | $60.16_{\pm 5.10}$ | $86.12_{\pm 3.34}$ | $90.58_{\pm 5.72}$ | $81.50_{\pm 5.10}$ | $92.34_{\pm 5.67}$ | $12.06_{\pm 4.77}$ |

*Table 2.* Performance on 8 cross-domain datasets (mean accuracy with subscripted standard deviation over $5 \times 5$ CV). **Bold** denotes the best result per dataset and underline denotes the second-best. Rank is the avg. rank across datasets (lower is better), computed with standard tie-breaking; Some datasets use only 50- 60 Optuna trials due to compute limits; "-" denotes OOM/unsupported runs, ranked last. ProtoGate targets very few sample and scales less favorably with larger $n$. Dataset abbreviations: ORL = orlraws10P, BAS = BASEHOCK, REL = RELATHE, PCM = PCMAC, CCY = Cell Cycle, CIF = CIFAR-10, DF-R = DrivFace-Regression, DF-C = DrivFace-Classification, REG = Regression ($R^2$). Model abbreviations: *GOTabPFN = our method, TabPFN-W = TabPFN Wide, TTables = TuneTables.

| MODEL/DB | ORL | BAS | REL | PCM | CCY | CIF | DF-R | DF-C | RANK |
|---|---|---|---|---|---|---|---|---|---|
| #SAMPLES | 100 | 1993 | 1427 | 1943 | 1067 | 11000 | 606 | 606 | – |
| #FEATURES | 10304 | 4862 | 4322 | 3289 | 42728 | 2048 | 6400 | 6400 | – |
| #CLASSES | 10 | 2 | 2 | 2 | 3 | 10 | REG. | 7 | – |
| *GOTabPFN | $\mathbf{100.00}_{\pm 0.00}$ | $\mathbf{97.11}_{\pm 1.00}$ | $88.87_{\pm 1.32}$ | $\mathbf{89.51}_{\pm 2.24}$ | $\mathbf{79.94}_{\pm 2.53}$ | $\mathbf{88.45}_{\pm 0.89}$ | $\mathbf{0.6548}_{\pm 0.0992}$ | $\mathbf{86.70}_{\pm 2.48}$ | $\mathbf{1.25}_{\pm 0.66}$ |
| TABDPT | $97.32_{\pm 0.68}$ | $97.00_{\pm 0.68}$ | $88.26_{\pm 1.49}$ | $87.58_{\pm 0.96}$ | $77.52_{\pm 2.44}$ | $88.00_{\pm 0.40}$ | $\underline{0.6505}_{\pm 0.0820}$ | $83.57_{\pm 3.04}$ | $4.44_{\pm 1.49}$ |
| TABPFN-W | $96.10_{\pm 6.52}$ | $\underline{97.04}_{\pm 0.72}$ | $87.84_{\pm 3.90}$ | $88.56_{\pm 0.71}$ | $\underline{78.67}_{\pm 2.42}$ | $88.12_{\pm 0.60}$ | $0.6430_{\pm 0.0772}$ | $85.42_{\pm 2.66}$ | $\underline{3.88}_{\pm 1.62}$ |
| TTABLES | $93.14_{\pm 1.02}$ | $93.14_{\pm 1.02}$ | $88.26_{\pm 2.31}$ | $84.27_{\pm 3.06}$ | $78.30_{\pm 1.13}$ | $78.05_{\pm 0.15}$ | $0.6332_{\pm 0.0675}$ | $84.62_{\pm 2.43}$ | $5.94_{\pm 1.91}$ |
| TABICL | $\underline{99.20}_{\pm 1.87}$ | $96.84_{\pm 0.75}$ | $88.15_{\pm 1.68}$ | $88.68_{\pm 1.94}$ | - | $87.61_{\pm 1.00}$ | - | $\underline{85.55}_{\pm 2.45}$ | $5.31_{\pm 2.46}$ |
| TANDEM | $99.00_{\pm 2.45}$ | $96.72_{\pm 0.86}$ | $\mathbf{89.42}_{\pm 1.42}$ | $\underline{88.99}_{\pm 1.33}$ | $77.31_{\pm 1.97}$ | $87.80_{\pm 0.36}$ | $0.6488_{\pm 0.0770}$ | $84.42_{\pm 2.55}$ | $4.00_{\pm 1.94}$ |
| LASSO | $96.00_{\pm 3.82}$ | $96.74_{\pm 0.88}$ | $86.87_{\pm 1.46}$ | $88.85_{\pm 1.80}$ | $77.86_{\pm 2.42}$ | $86.42_{\pm 1.10}$ | $0.3194_{\pm 0.0668}$ | $77.95_{\pm 3.03}$ | $6.13_{\pm 1.69}$ |
| MLP | $91.00_{\pm 8.37}$ | $96.91_{\pm 1.15}$ | $\underline{89.12}_{\pm 1.85}$ | $87.71_{\pm 1.95}$ | $76.38_{\pm 1.42}$ | $\underline{88.15}_{\pm 0.54}$ | $0.5682_{\pm 0.1258}$ | $80.59_{\pm 3.72}$ | $5.25_{\pm 2.17}$ |
| PROTOGATE | $66.40_{\pm 11.32}$ | - | $58.40_{\pm 3.15}$ | - | - | - | $-0.1636_{\pm 0.0738}$ | $66.70_{\pm 10.20}$ | $8.81_{\pm 0.35}$ |

*Table 3.* **Colon ablation.** Accuracy over $5 \times 5$ CV. All NSC variants use NSC-pSP unless noted.

| Variant | Acc. (%) |
|---|---|
| GO-LR+NSC-pSP (tokens) | $\mathbf{88.18} \pm 10.05$ |
| No NSC (TabPFN-2.5) | $86.85 \pm 9.16$ |
| Rand. order + NSC-pSP | $84.21 \pm 10.34$ |
| GO-LR+NSC-pSP + LogReg | $82.67 \pm 11.05$ |
| Id. order + NSC-pSP | $81.67 \pm 11.91$ |
| GO-LR+NSC-pSP (uniform seg.) | $80.00 \pm 11.30$ |
| Mean-pool NSC-pSP, no GO-LR | $64.59 \pm 3.67$ |

TAN cluster (x86_64 CPU, 188,GB RAM, TITAN RTX 24,GB) with PyTorch 2.4.1+cu121. We disable AMP for GOTabPFN, but some transformer baselines require AMP and/or DP/DDP to fit GPU memory. We tune only GO-LR and NSC in GOTabPFN via Optuna (Akiba et al., 2019) (150 trials/dataset), following standard tabular tuning practice (Gorishniy et al., 2025; 2024; 2022; 2021); the TabPFN-2.5 head remains frozen. For baselines, we use authors' recommended settings when tuning is unnecessary; when tuning is recommended, we also run Optuna (150 trials) for fairness. Among TabPFN-based models, TuneTables likewise uses lightweight Optuna tuning. See Fig. A.1.

**HDLSS classification performance.** Table 1 reports mean accuracy ($\pm$std) over $5 \times 5$ repeated CV on eight HDLSS benchmarks. GOTabPFN is best on all datasets (8/8) with the lowest average rank ($1.00 \pm 0.00$), indicating consistent dominance. Relative to the strongest TabPFN variants (TabPFN-Wide, TuneTables, BETA), gains are largest on harder/noisier datasets (e.g., SMK, TOX) and smaller on near-saturated ones (e.g., ALLAML, Prostate), where headroom is limited. GOTabPFN also shows comparable or lower split-to-split variance, suggesting improved robustness in the low-sample regime. Full results against 55 baselines are in Table G.1 (Appendix G). Across the 8

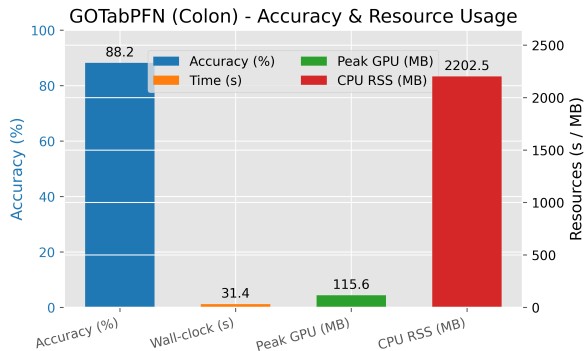

*Figure 6.* **Accuracy-resource profile on Colon.** Wall-clock time, peak GPU memory, and CPU RSS for GOTabPFN.

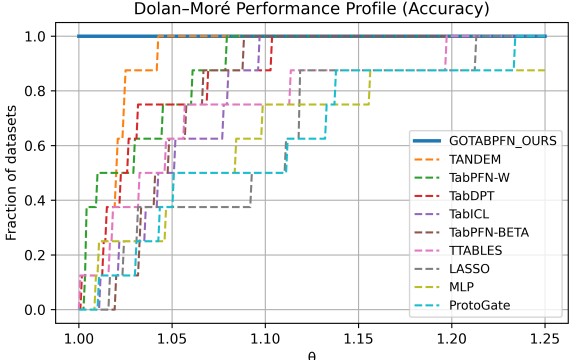

*Figure 7.* **Dolan–Moré profiles.** Performance profiles over 8 HDLSS datasets against the top-10 baselines.

cross-domain(App. T) high-dimensional datasets (Table 2), GOTabPFN achieves the best average rank ($1.25 \pm 0.66$) and obtains the top result on 7/8 tasks, including image-derived, biological, text-like, and camera-sensor datasets. The strongest competing methods are TabPFN-W, TAN-DEM, TabDPT, and MLP, but their gains are less consistent across domains. These results suggest that GO-LR+NSC provides a robust ordering-aware compression interface for diverse HDLSS and related high-dimensional regimes.

**Statistical significance.** Across 8 HDLSS datasets, GOTabPFN achieves the best average rank and is separated from competing baselines by Friedman (Friedman, 1937)/Nemenyi (Nemenyi, 1963) critical-difference analysis (Fig. I.1). While paired Wilcoxon signed-rank tests (Demšar, 2006) yield consistent directional improvements (all $p_{\mathrm{raw}} = 0.00781$), significance does not always survive Holm correction due to the small number of datasets and the resulting conservativeness of multiple-comparison control (Table I.1). Additional statistical details are in Appendix I.

**Runtime and computational complexity.** For $n$ samples, $m$ features, $k$ sample-clusters, and $M$ NSC tokens, GO-LR costs $\mathcal{O}(nmkI + m^2n)$, plus $\mathcal{O}(km^2b)$ for KL graph construction; refinement/integration adds $\mathcal{O}(kPm^2 + k^2m)$, where $P$ is no. of Sweep Refine passes. In HDLSS ($n \ll m$), NSC fits per-segment PC1 directions in $\mathcal{O}(n^2m)$ and tokenizes in $\mathcal{O}(nm)$. The TabPFN-2.5 head on $M$ tokens is dominated by attention over $n$ context points, scaling as $\tilde{\mathcal{O}}(n^2M)$ per split. On Colon, GOTabPFN achieves 88.2%

*Table 4.* **GOTabPFN gains over foundation-model heads.** Accuracy on 8 HDLSS datasets under $5 \times 5$ CV. "Best orig" is the best among TabDPT, TabPFN-Wide, BETA, TuneTables, and TabICL.

| Dataset | Best orig | GOTabPFN | $\Delta_{\mathrm{abs}}$ | $\Delta_{\mathrm{rel}}$ |
|---|---|---|---|---|
| Colon | 87.85 | 88.18 | 0.33 | 0.38 |
| Lung | 96.55 | 97.44 | 0.89 | 0.92 |
| GLI85 | 89.66 | 93.82 | 4.16 | 4.64 |
| SMK | 71.99 | 74.23 | 2.24 | 3.11 |
| ALLAML | 97.16 | 97.54 | 0.38 | 0.39 |
| Prostate | 93.31 | 93.37 | 0.06 | 0.06 |
| Arcene | 88.00 | 90.60 | 2.60 | 2.95 |
| TOX | 93.25 | 93.33 | 0.08 | 0.09 |

accuracy in 31.4s with modest peak GPU use (115.6MB; Fig. 6); memory is primarily CPU-side (RSS $\approx$ 2202.5MB), consistent with GO-LR graph construction/refinement.

**Ablations.** Figs. 4-5 and Table 4 quantify adding GO-LR ordering and NSC compression before a frozen TabPFN-2.5 head: GOTabPFN matches or exceeds the best original tabular foundation model on all 8 HDLSS datasets, with largest gains on GLI-85 (+4.16 pp), Arcene (+2.60 pp), and SMK (+2.24 pp), and near-saturated improvements on ALLAML/Prostate. On Colon (Table 3), removing NSC lowers accuracy, and replacing GO-LR with identity/random orders drops more, indicating NSC benefits from structure-revealing orderings; transition-aware segmentation and PCA-based token embeddings outperform uniform segmentation or mean-pooling, while swapping TabPFN-2.5 for logistic regression substantially degrades performance, suggesting both the ordering+compression pipeline and a strong TabPFN-style predictor are needed. The Dolan-Mor'e profile (Dolan & Moré, 2002) (Fig. 7, following TANDEM (Naor & Lindenbaum, 2025)) shows stronger cross-dataset consistency: GOTabPFN stays closest to the per-dataset best (curve at 1.0), whereas others need larger tolerance $\theta$. Additional ablations appear in App. J.

**Limitations.** GOTabPFN inherits constraints from its frozen TabPFN-2.5 backbone, including its limit of up to 10 classes and sample-size limit of 50K samples. GO-LR+NSC adds ordering and compression before TabPFN inference; runtime can increase for larger sample sizes. Thus, GOTabPFN is most suitable for HDLSS and related low-sample, high-dimensional regimes, rather than high-sample regimes.

## 5. Conclusion

We present **GOTabPFN**, which makes TabPFN-style small tabular foundation models effective in HDLSS regimes. GOTabPFN couples MinLA-grounded feature ordering (GO-LR) with a neuro-inspired stable, locality-preserving compression interface (NSC) that converts high-dimensional tables into compact token sequences. Without retraining or modifying the TabPFN-2.5 backbone, this ordering-to-tokenization pipeline improves accuracy and robustness under tight token budgets across diverse HDLSS benchmarks. GOTabPFN provides a theory-grounded, practical route to scalable in-context tabular prediction when $m \gg n$.

## Acknowledgements

This work was supported in part by the US National Science Foundation under Awards #1920920, #2125872, and #2223793. We thank the anonymous ICML reviewers for their valuable feedback and suggestions.

## Impact Statement

GOTabPFN aims to make tabular foundation models more usable in HDLSS settings by introducing an ordering-aware compression interface that reduces feature dimensionality while preserving local structure. This can benefit scientific and biomedical domains where data are scarce but feature spaces are large. However, as with any predictive model, deployment in sensitive domains should include careful validation, bias assessment, and domain-expert oversight.

## Software and Data

The project webpage is available at `https://www.zadidhabib.com/gotabpfn.html`. Code, notebooks, and installation instructions are available at `https://github.com/zadid6pretam/GOTabPFN`; the package can also be installed with `pip install gotabpfn`. Our experiments use TabPFN-2.5 as the frozen backbone with `tabpfn==6.3.1`; newer default `tabpfn` installations may install TabPFN-3 or later, so reproducing our results requires `pip install tabpfn==6.3.1` and may require Prior Labs/Hugging Face checkpoint access. Most HDLSS datasets are from the scikit-feature dataset repository (Li et al., 2018) (`https://jundongl.github.io/scikit-feature/datasets.html`); DrivFace is from UCI (Hernández-Sabat et al., 2016); CIFAR-10 embeddings are derived from the Kaggle CIFAR-10 dataset (Cukierski, 2013); and Cell Cycle is from Mahdessian et al. (2021) via GEO (NCBI, 2021).

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

- Detailed Related Work in Sec. A

- Theoretical Characterization of GO-LR: Complexity and TSP Connections in Sec. B

- NSC Variants and NSC as a Shared Piecewise Pooling Operator in Sec. C

- NSC as a Structured Dimensionality Reduction Layer in Sec. D

- Why Feature Ordering? Local Neighborhoods Enable Structure-Aware Compression in Sec. E

- Feature Ordering - When to Use? Through the Lens of Locality in Sec. F

- Detailed Comparative Results in Sec. G

- GOTabPFN Hyperparameters in Sec. H

- Statistical Significance Analysis in Sec. I

- Additional Ablation Analysis in Sec. J

- Representation Quality via t-SNE in Sec. K

- Inference Level Ablation on Calibration and Robustness in Sec. L

- Sanity and Stress Diagnostics in Sec. M

- Additional Reliability and Interpretability Diagnostics in Sec. N

- Theory-Inspired Representation Diagnostics in Sec. O

- OOD and Local Sensitivity Diagnostics in Sec. P

- Deployment-Oriented Triage Diagnostics in Sec. Q

- Extension beyond TabPFN in Sec. R

- TabPFN Seed Sensitivity in Sec. S

- Additional Clarifications in Sec. T

## A. Detailed Related Work

**Tabular Models.** Recent tabular deep learning has also advanced rapidly beyond early attention/MLP baselines. In-context learning-based model such as TabICL (Jingang et al., 2025) aim to provide strong out-of-the-box performance (especially in low-data regimes), while methods like TabR (Gorishniy et al., 2024) and TabM (Gorishniy et al., 2025) improve classical deep tabular pipelines via nearest-neighbor augmentation or parameter-efficient ensembling. Hybrid and low-label settings are explored by TANDEM (Naor & Lindenbaum, 2025), and context optimization for scalable prior-fitted networks is studied in TuneTables (Feuer et al., 2024). TabDPT pre-trains a row-token tabular foundation model by combining retrieval-based in-context learning with self-supervised column-masking on large-scale real datasets, enabling strong generalization to unseen tasks without per-dataset tuning (Ma et al., 2025). At the same time, strong "simple" baselines remain highly competitive: carefully pre-tuned MLPs such as Real-MLP (Holzmüller et al., 2024) can be surprisingly hard to beat, and gradient-boosted decision trees, especially XGBoost (Chen & Guestrin, 2016), LightGBM (Ke et al., 2017), and CatBoost (Prokhorenkova et al., 2018) continue to thrive as robust, high-performing workhorses on many tabular benchmarks.

**Feature Selection-based HDLSS Specific Models.** Feature-selection methods are particularly relevant for HDLSS tabular learning, where selecting a compact, informative subset of features is often more critical than scaling model capacity. ProtoGate (Jiang et al., 2024) addresses this regime with prototype-guided gating that selects features by aligning samples with class prototypes, improving robustness when samples are scarce. Several works perform instance-wise or differentiable

subset selection: INVASE (Yoon et al., 2019) learns a sample-dependent feature selector trained with prediction and selection regularization, while STG (Yamada et al., 2020) uses stochastic gates to enable end-to-end feature selection with sparsity control. LSPIN/LLSPIN (Yang et al., 2022a) further emphasizes interpretable, per-instance gating for tabular inputs. Complementarily, L2X (Chen et al., 2018) and REAL-X (Jethani et al., 2021) learn differentiable selection/explanation mechanisms that identify a small set of features sufficient for prediction, offering a principled way to trade accuracy for sparsity and interpretability in low-sample settings. RKNN-FS (Li et al., 2011) addresses HDLSS learning through feature selection, using a random $k$-nearest neighbor (RKNN) strategy to identify informative features before prediction.

**LLMs for Tabular Data.** Motivated by general-purpose LLMs (Brown et al., 2020; Achiam et al., 2023; Guo et al., 2024; 2025), recent work adapts them to tabular prediction via fine-tuning (e.g., LIFT (Dinh et al., 2022) on GPT-3 (Brown et al., 2020), TabLLM (Hegselmann et al., 2023) on T0 (Sanh et al., 2022)) and tabular-centric pretraining for transfer/instruction following (e.g., TP-BERTa (Yan et al., 2024), GTL (Wen et al., 2024)). Other lines use prompting or hybrid pipelines, including correlated-text semi-supervision (P2T (Nam et al., 2024b)), weak-learner boosting (Summary (Manikandan et al., 2023)), feature synthesis (FeatLLM (Han et al., 2024)), synergy learning with tabular backbones (SERSAL (Yan et al., 2025)), LLM-guided rule/feature generation (Nam et al., 2024a; Zhang et al., 2023), rule refinement without LLM fine-tuning (Ye et al., 2025a), metadata-driven distillation (Shi et al., 2025), and order-bias mitigation (ROTATOR-LLM (Wang et al., 2025)). Despite encouraging efforts, current tabular LLMs remain poorly suited to HDLSS and very high-dimensional tables, since attention-based architectures incur quadratic cost in the token/feature dimension.

**TabPFN and Variants.** TabPFN introduced the idea of a tabular foundation model that performs in-context learning for small tabular classification problems (Hollmann et al., 2023). This was later substantially extended and validated at scale in TabPFN v2 (Hollmann et al., 2025). Recent follow-ups push accuracy and scalability along multiple axes: TabPFN-2.5 advances the state of the art in tabular foundation models (Grinsztajn et al., 2025), TabPFN Unleashed (BETA) targets practical scalability and effectiveness on broader settings (Liu & Ye, 2025), and TabPFN-Wide explores continued pre-training for extreme feature counts (Kolberg et al., 2025). Orthogonally, LoCalPFN investigates retrieval and fine-tuning mechanisms for in-context tabular models (Thomas et al., 2024). Our work is complementary to these efforts: rather than modifying TabPFN itself, we develop a stable, structure-aware compression interface that enables TabPFN-style predictors to operate reliably in HDLSS regimes with $m \gg n$ and very large feature counts.

**Other Models.** MLP-PLR is a strong MLP baseline for tabular data that replaces raw continuous inputs with learnable piecewise-linear (PLR) feature embeddings, improving expressivity and performance over standard MLPs (Gorishniy et al., 2022). Other tabular models such as TabNet (Arik & Pfister, 2021), TabTransformer (Huang et al., 2020)/FT-Transformer (Gorishniy et al., 2021), SAINT (Somepalli et al., 2022), AutoInt (Song et al., 2019), and interaction-centric networks (DeepFM (Guo et al., 2017), DCN (Wang et al., 2017)) differ primarily in how they represent columns and capture cross-feature dependencies: TabNet performs step-wise attentive feature selection via sparse masks for interpretability, while transformer families tokenize columns (especially categorical features) and use self-attention to model contextual interactions, with FT-Transformer providing a lighter mixed-type variant; SAINT further strengthens tabular transformers via augmentation and contrastive/self-supervised pretraining, and AutoInt targets high-order interactions directly through attention. A complementary line treats tabular inputs as feature sequences, including ordering-based approaches (TabSeq (Habib et al., 2024)) and recurrent processing (TabulaRNN (Thielmann & Samiee, 2024)), which impose position-aware inductive bias but may depend on the quality of the chosen order. More recent sequence backbones replace attention with state-space modeling e.g., Mambular (Thielmann et al., 2024), MambaTab (Ahamed & Cheng, 2024), and hybrids like MambAttention (Thielmann & Samiee, 2024) leveraging Mamba-style selective state-space layers for efficient long-range dependency modeling. Tree-inspired inductive biases remain competitive through differentiable ensembles (NODE (Popov et al., 2020), ENODE (OpenTabular Contributors, 2025), NDTF (Kontschieder et al., 2015)) that emulate decision trees with end-to-end training, alongside classical ensembles (Random Forest, AdaBoost, GBM) that provide strong, stable baselines. Finally, representation and regularization advances such as DANets (Chen et al., 2022), ResNetTabular (OpenTabular Contributors, 2025), CategoryEmbedding (Gorishniy et al., 2022), TANGOS (Jeffares et al., 2023), and metric/contrastive methods (ModernNCA (Ye et al., 2025c), Trompt (Chen et al., 2023)) improve robustness and optimization on noisy, small-sample settings, while standard baselines (Naive Bayes, KNN, SVM, Decision Tree, Lasso, MLP, 1-D CNN) remain important reference points due to their interpretability and well-characterized trade-offs. GeoAggregator (Deng et al., 2025) and ZAYAN (Habib et al., 2026c) represent recent domain-specific geospatial tabular deep learning models, with GeoAggregator targeting spatially aware geospatial regression and ZAYAN focusing on feature-level contrastive learning for tabular remote sensing and environmental data.

**Feature Ordering and Permutation Sensitivity.** Mambular (Thielmann et al., 2024) first highlighted that random permutations of tabular feature order can induce brittleness in predictive performance, even under fixed seeds. Concurrently, TabSeq (Habib et al., 2024) explicitly introduced a feature ordering algorithm for tabular deep learning, establishing

# SOTA tabular models grouping as per tuning requirement

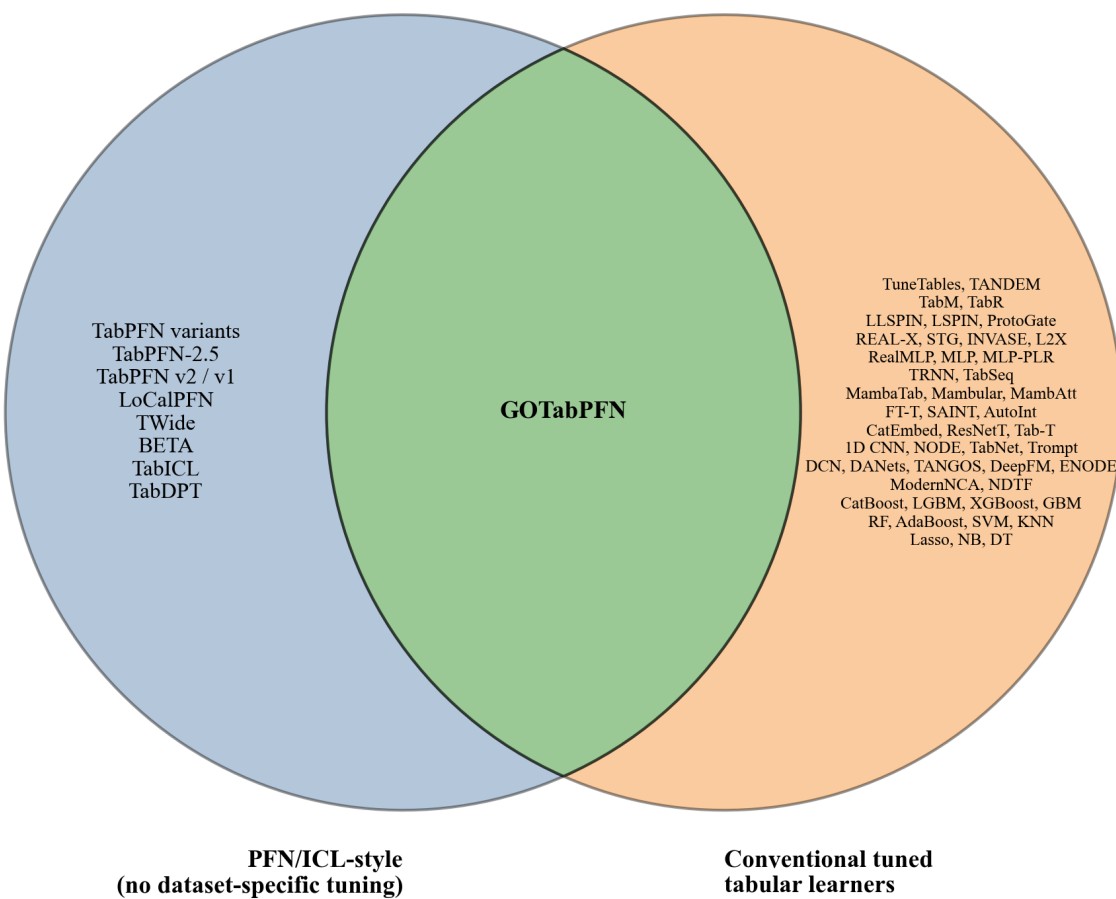

*Figure A.1.* **GOTabPFN tuning regime.** GOTabPFN sits between frozen PFN/ICL-style tabular foundation models and fully tuned tabular learners: only the GO-LR+NSC front-end is tuned, while TabPFN-2.5 remains frozen. See Appendix T for more clarifications.

column permutation as a learnable design choice rather than a nuisance factor. Later ROTATOR-LLM (Wang et al., 2025) extended this direction by studying feature ordering for LLM-based tabular inference. DynaTab (Habib et al., 2026b) systematically studied when feature ordering matters in high-dimensional tabular learning, introducing an Intrinsic Dimensionality Factor (IDF) and feature-to-sample ratio $\rho = m/n$ based categorization of dataset regimes. It proposed a neuroscience-inspired Dynamic Feature Ordering (DFO) algorithm and showed that sequence-sensitive backbones such as Transformers, LSTMs, denoising autoencoders, and SSM-style models can benefit from adaptive ordering. However, its use of vanilla sequence backbones incurs substantial memory and runtime costs, motivating more compact ordering-aware tokenization and compression strategies. We use Fig. A.1 to position GOTabPFN within this landscape according to hyperparameter tuning requirements for tabular models: unlike PFN/ICL-style models that require no dataset-specific tuning and conventional tabular learners that are fully tuned, GOTabPFN occupies the middle regime by tuning only the GO-LR+NSC front-end while retaining a frozen TabPFN-2.5 backbone.

## B. Theoretical Characterization of GO-LR: Complexity and TSP Connections

### B.1. GO-LR as a TSP-Style Initialization with Local Refinement.

GO-LR does not solve MinLA exactly; instead, our implementation constructs a permutation by greedy seriation over a pairwise dissimilarity matrix. The initialization coincides with a nearest-neighbor heuristic for a TSP-path objective defined on a complete graph, and GO-LR then locally refines the resulting permutation under the dispersion objective.

**Definition B.1** (TSP-path Objective on a Complete Graph). Given a complete weighted graph $\mathcal{K} = (V, \binom{V}{2}, d)$ with edge

*Table B.1.* **GO-LR vs. classic metaheuristics on Colon.** Lower runtime, TSP cost, and MinLA are better; higher accuracy is better. Accuracy is reported over $5 \times 5$ CV. Christofides and Simulated Annealing are implemented using NetworkX (Hagberg et al., 2008).

| Ordering Method$_{+\text{NSC}}$ | Runtime | TSP $\downarrow$ | MinLA $\downarrow$ | Accuracy $\uparrow$ |
|---|---|---|---|---|
| GO-LR (ours) | **10.07** | 21958.78 | **1.4743e10** | **0.8818 $\pm$ 0.1005** |
| Simulated Annealing | 15.01 | **11712.75** | 1.4803e10 | 0.8405 $\pm$ 0.0944 |
| Genetic Algorithm | 206.59 | **11712.75** | 1.4803e10 | 0.8405 $\pm$ 0.0944 |
| Ant Colony Optimization | 1501.44 | 11792.08 | 1.4760e10 | 0.8595 $\pm$ 0.1000 |
| Christofides | 1424.83 | 11715.06 | 1.4994e10 | 0.8364 $\pm$ 0.0911 |

weights $d_{ij} \geq 0$, we define the path cost of a permutation $\sigma = (\sigma_1, \ldots, \sigma_m)$ by Eq. 8, repeated below as Eq. 26 for brevity.

$$\text{PathCost}(\sigma) \;=\; \sum_{t=1}^{m-1} d_{\sigma_t, \sigma_{t+1}} \tag{26}$$

**Lemma B.2** (Nearest-Neighbor Heuristic). *The greedy procedure "start at* $\arg\min_i \sum_j d_{ij}$ *and repeatedly append the nearest unvisited node" returns a Hamiltonian path in* $\mathcal{K}$ *and is a standard nearest-neighbor heuristic for minimizing Eq.* (8) *(Rosenkrantz et al., 1977).*

*Proof sketch.* At each step exactly one new unvisited node is appended; thus the resulting sequence visits each node exactly once and forms a Hamiltonian path. The choice "nearest unvisited" is precisely the nearest-neighbor rule for minimizing Eq. (8). $\square$

**Theorem B.3** (The Initialization Step of GO-LR Can Be Used as a TSP-path Heuristic). *For any complete weighted graph* $\mathcal{K}$, *the GO-LR local seriation rule (nearest-neighbor) can be used as a TSP-path heuristic that outputs a Hamiltonian path* $\sigma$ *for* $\mathcal{K}$. *By Lemma B.2, the output is a Hamiltonian path.*

*Proof sketch.* Given $\mathcal{K}$, run the greedy nearest-neighbor construction on its distance matrix $[d_{ij}]$. By Lemma B.2, the output is a Hamiltonian path and corresponds to a permutation $\sigma$. $\square$

*Remark* B.4 (What is (and is not) "equivalent to TSP" here). GO-LR's local objective Eq. (2) is MinLA, while the nearest-neighbor constructor optimizes a TSP-path surrogate for initialization, while GO-LR subsequently refines the ordering using the dispersion objective in Eq. (2). Empirically, the surrogate tends to produce low-dispersion permutations under Eq. (2) when $d_{ij}$ is chosen to be compatible with $w_{ij}$ (e.g., monotone transforms or neighborhood sparsification).

**GO-LR vs. classic metaheuristics.** To assess whether GO-LR is overly limited by its greedy construction, we replace GO-LR with stronger stochastic/metaheuristic orderings while keeping the downstream NSC + TabPFN-2.5 pipeline fixed. On Colon, although Simulated Annealing (Kirkpatrick et al., 1983), Genetic Algorithm (Larranaga et al., 1999), Ant Colony Optimization (Dorigo & Gambardella, 2002), and Christofides-based ordering (Christofides, 2022) often achieve lower TSP-style surrogate cost, they do not improve downstream accuracy; GO-LR obtains the best $5 \times 5$ CV accuracy ($0.8818 \pm 0.1005$), the lowest runtime (10.07s), and the best MinLA-style dispersion objective. We observe the same trend on the larger Cell Cycle (Mahdessian et al., 2021; NCBI, 2021) RNA-seq dataset ($n = 1067, m = 42728$): GO-LR+NSC+TabPFN-2.5 achieves $79.94 \pm 2.53$ accuracy, $92.36 \pm 1.36$ AUC, and $79.95 \pm 2.51$ macro-F1, compared to $76.45 \pm 2.29$, $92.76 \pm 1.10$, and $76.42 \pm 2.29$ for Simulated Annealing+NSC+TabPFN-2.5. Consistent with our theory, GO-LR attains the lower MinLA cost on Cell Cycle ($8.14 \times 10^{11}$ vs. $8.51 \times 10^{11}$), while Simulated Annealing attains the lower TSP-path cost; this is expected, since GO-LR is designed to optimize the MinLA-style dispersion objective rather than a TSP surrogate alone. These results suggest that optimizing a TSP surrogate more aggressively does not necessarily yield better feature orderings for NSC+TabPFN-2.5; GO-LR is better aligned with the MinLA-style objective and provides a stronger accuracy-efficiency tradeoff in practice.

**On global optimality.** GO-LR does not guarantee a globally optimal ordering, since the underlying MinLA-style feature ordering problem is combinatorial and NP-hard. Instead, it is a structured approximation strategy: clustering induces local feature graphs, NNPath provides an efficient initialization, and local refinement explicitly reduces the MinLA-style dispersion objective. Empirically, replacing GO-LR with stronger metaheuristics such as Simulated Annealing, Genetic Algorithm, Ant Colony Optimization, and Christofides-based ordering does not improve downstream NSC+TabPFN-2.5 performance. On both Colon and the larger Cell Cycle transcriptomic dataset, GO-LR achieves better downstream accuracy

and lower MinLA cost despite some alternatives attaining lower TSP-style surrogate cost, suggesting that GO-LR is better aligned with the objective that matters for ordering-aware compression and prediction.

**GO-LR cluster-size sensitivity.** We ablate the GO-LR cluster size $k$ on Colon while fixing all other hyperparameters to the best tuned configuration. As shown in Table B.2, performance peaks at $k = 10$, reproducing the best Colon result ($88.18_{\pm 10.05}$) in a fresh run. The trend is non-monotonic: small $k$ values appear too coarse to capture local sample heterogeneity, while overly large $k$ values fragment the data into noisier local graphs, weakening the aggregated global ordering. This supports using a moderate cluster size as the best trade-off.

*Table B.2.* **GO-LR cluster-size sensitivity on Colon.** All settings except $k$ are fixed to the best tuned configuration. Values are mean accuracy with subscripted standard deviation over $5 \times 5$ CV.

| $k$ | 3 | 4 | 5 | 7 | 10 | 12 | 15 |
|---|---|---|---|---|---|---|---|
| Acc. | $82.72_{\pm 10.69}$ | $84.38_{\pm 9.82}$ | $83.95_{\pm 9.54}$ | $84.33_{\pm 8.95}$ | $\mathbf{88.18}_{\pm 10.05}$ | $83.72_{\pm 10.16}$ | $82.69_{\pm 11.08}$ |

# C. NSC Variants and NSC as a Shared Piecewise Pooling Operator

## C.1. Intrinsic-Dimension Rules and Budgets for NSC Variants

For completeness we describe the intrinsic-dimension estimators and budget rules used by the remaining NSC variants. Recall that $\tilde{X} \in \mathbb{R}^{n \times m}$ is the standardized training matrix and we get Eq. 27 which denotes denotes its covariance (or correlation) matrix with nonzero eigenvalues $\{\lambda_i\}_{i=1}^r$, $r \leq \min(n, m)$.

$$\Sigma = \frac{1}{n-1} \tilde{X}^\top \tilde{X} \in \mathbb{R}^{m \times m} \tag{27}$$

**Effective-rank intrinsic dimension.** Besides the PCA cumulative-variance rule in Eqs. (18)-(20), we also consider an effective-rank estimate (Roy & Vetterli, 2007; Halko et al., 2011). We get Eq. 28 and define Eq. 29 with a small constant $\epsilon > 0$ for numerical stability. Some variants set $\hat{d} = d_{\text{eff}}$ instead of $\hat{d}_{\text{PCA}}(\tau)$.

$$p_i = \frac{\lambda_i}{\sum_{j=1}^r \lambda_j} \tag{28}$$

$$d_{\text{eff}} = \exp\left(-\sum_{i=1}^r p_i \log(p_i + \epsilon)\right) \tag{29}$$

**IDF-based budget modulation.** We define the IDF (Habib et al., 2026b) in Eq. 30 which compares intrinsic and ambient dimensionality. An IDF-based budget rule optionally used in NSC and NSC-P is defined by Eq. 31 where $\gamma$, $M_{\min}$, and $M_{\max}$ control redundancy allowance and token-budget bounds.

$$\text{IDF} = \frac{\hat{d}}{m}, \qquad \hat{d} \in \{d_{\text{eff}}, \hat{d}_{\text{PCA}}(\tau)\} \tag{30}$$

$$M = \text{clip}\Big(\lceil(1 + \beta(1 - \text{IDF}))\,\hat{d}\rceil, \ M_{\min}, \ \min(M_{\max}, m)\Big),$$
$$\beta \in [0, 1] \tag{31}$$

**Variant summary.**

- **NSC**: uses a fixed, user-chosen $M$ (no intrinsic-dimension estimate).
- **NSC-P**: chooses $\hat{d}$ via either $d_{\text{eff}}$ or $\hat{d}_{\text{PCA}}(\tau)$ and then applies an IDF- or $\gamma$-based rule such as Eq. 32

$$M = \text{clip}\big(\lceil \gamma \hat{d} \rceil, \ M_{\min}, \ \min(M_{\max}, m)\big) \tag{32}$$

- **NSC-SP**: uses a fixed $M$ but applies SegPCA pooling (Sec. 3.2).
- **NSC-pSP**: uses the PCA-based rule in Eqs. (20)-(21) (described in the main text).

**Computational note.** In HDLSS regimes ($m \gg n$), we avoid forming a full eigen-decomposition of the $m \times m$ matrix $\Sigma$ in Eq. (27) by computing the nonzero spectrum via the $n \times n$ Gram matrix $\tilde{X}\tilde{X}^\top$ or using randomized methods (Halko et al., 2011). The resulting eigenvalues are then re-used in the intrinsic-dimension rules above.

**Proposition C.1** (NSC as a Shared Piecewise Pooling Operator). *Let $\Pi^*$ be a fixed global feature permutation and let $\{\mathcal{S}_t\}_{t=1}^M$ be the contiguous segments defined by NSC. The Neuro-Inspired Subunit Compression defines a mapping in Eq. 33 which is given by Eq. 34 where the same pooling function $g_\theta$ is shared across all segments.*

$$F_{NSC} : \mathbb{R}^m \to \mathbb{R}^{M \times d} \tag{33}$$

$$F_{NSC}(x) = \left( g_\theta\big(\psi(x_{\mathcal{S}_1}^\Pi)\big), \ldots, g_\theta\big(\psi(x_{\mathcal{S}_M}^\Pi)\big) \right) \tag{34}$$

*Then $F_{NSC}$ is a piecewise pooling operator with the following properties: (1)* **Locality.** *Each output token depends only on a contiguous subset of the ordered feature axis; (2)* **Parameter sharing.** *The number of trainable parameters is independent of $m$ and depends only on $g_\theta$; and (3)* **Linear complexity.** *For fixed pooling depth, $F_{NSC}$ can be evaluated in $O(m)$ time and $O(Md)$ space per sample.*

Proposition C.1 shows that NSC induces a structured, order-aware compression operator that preserves locality while remaining scalable to extreme HDLSS regimes.

*Proof sketch.* By construction, the ordered feature vector $x^\Pi$ is partitioned into $M$ disjoint contiguous segments whose union covers $\{1, \ldots, m\}$. Each segment is processed independently by the same pooling function $g_\theta$, establishing locality and parameter sharing. Since each feature appears in exactly one segment and $g_\theta$ has constant depth, the total number of operations scales linearly with $m$, yielding $O(m)$ time complexity. The output representation stores $M$ vectors of dimension $d$, giving $O(Md)$ space complexity. $\square$

## D. NSC as a Structured Dimensionality Reduction Layer

In this section we view the NSC (Section 3.2) as an explicit Dimensionality Reduction (DR) layer that maps high-dimensional GO-LR-ordered features into a low-dimensional latent space. We first formalize NSC as a structured DR map, then relate it to classical DR methods, and finally outline and instantiate an empirical protocol comparing NSC to Principal Component Analysis (PCA) (Hotelling, 1933), Random Projections (RP) (Johnson & Lindenstrauss, 1984), Uniform Manifold Approximation and Projection (UMAP) (McInnes et al., 2018), Pairwise Controlled Manifold Approximation (PaCMAP) (Wang et al., 2021), and Autoencoders (AE) (Hinton & Salakhutdinov, 2006) quantitatively.

### D.1. NSC as a Structured DR Map

Recall that GO-LR produces a global feature permutation $\Pi^*$ that approximately solves a weighted MinLA problem (Section 3.1), bringing highly correlated or low-dissimilarity features into local neighborhoods along the ordered axis. NSC then segments this ordered axis into $M$ contiguous subunits $\{\mathcal{S}_t\}_{t=1}^M$ and applies a shared pooling operator $g_\theta \circ \psi$ to each segment (Eqs. 12-17). For a sample $x \in \mathbb{R}^m$ we again write Eq. 35 and define segments $\mathcal{S}_t \subset \{1, \ldots, m\}$ that partition the ordered axis. The $t$-th meta-feature is defined by Eq. 36 yielding a meta-feature sequence $Z(x) = (z_1, \ldots, z_M) \in \mathbb{R}^{M \times d}$ (Eqs. 16-17). In our implementation we often set $d = 1$ (scalar tokens), and flatten $Z(x)$ into a vector in $\mathbb{R}^M$. Formally, NSC again defines a mapping in Eq. 37 as summarized in Proposition C.1. When $d = 1$ and we flatten across segments, this reduces to a DR map by Eq. 38. Thus NSC acts as a structured DR layer whose output dimensionality $M \ll m$ is explicitly controlled by the meta-feature budget.

$$x^\Pi = \big( x_{\Pi^*(1)}, \ldots, x_{\Pi^*(m)} \big) \tag{35}$$

$$z_t = g_\theta\big(\psi(u_t)\big), \qquad u_t := x_{\mathcal{S}_t}^\Pi \tag{36}$$

$$F_{\text{NSC}} : \mathbb{R}^m \to \mathbb{R}^{M \times d}, \qquad F_{\text{NSC}}(x) = \left( g_\theta\big(\psi(x_{\mathcal{S}_1}^\Pi)\big), \ldots, g_\theta\big(\psi(x_{\mathcal{S}_M}^\Pi)\big) \right) \tag{37}$$

$$\Phi_{\text{NSC}} : \mathbb{R}^m \to \mathbb{R}^M, \qquad \Phi_{\text{NSC}}(x) = \text{flatten}\big(F_{\text{NSC}}(x)\big) \tag{38}$$

**Structured locality.**    Unlike global DR methods such as PCA (Hotelling, 1933), $\Phi_{\text{NSC}}$ has built-in locality: each coordinate of the compressed representation depends only on a contiguous block of GO-LR–ordered features. In particular, the $t$-th coordinate of $\Phi_{\text{NSC}}(x)$ is a pooled summary of the subunit $u_t = x_{\mathcal{S}_t}^{\Pi}$, where $\mathcal{S}_t$ corresponds to a locally redundant neighborhood in the GO-LR ordering. This induces a piecewise pooling structure: different coordinates of the latent vector correspond to different ordered blocks rather than global linear mixtures of all features.

**Linear and nonlinear regimes.**    If both $\psi$ and $g_\theta$ are linear maps, NSC reduces to a structured linear DR operator. Writing $\psi(u_t) = A_t u_t$ and $g_\theta(v) = w^\top v$, we obtain Eq. 39 where $P_\Pi$ is the permutation matrix for $\Pi^*$ and $P_{\mathcal{S}_t}$ selects segment indices. Stacking across $t$ yields a linear map $W_{\text{NSC}}x$ with a structured block-sparse pattern aligned with the ordered segments. When $\psi$ or $g_\theta$ include nonlinear statistics (e.g., quantiles, skewness, kurtosis, shallow MLP), NSC becomes a local nonlinear DR layer with the same segmentation structure.

$$z_t = w^\top A_t u_t = w^\top A_t\, x_{\mathcal{S}_t}^{\Pi} = w^\top A_t\, P_{\mathcal{S}_t} P_\Pi x \tag{39}$$

**Intrinsic dimension-aware budget.**    Section 3.2 ties the meta-feature budget $M$ to an estimate of intrinsic dimensionality $\hat{d}$ via effective rank (Eqs. 27-29). Our default rule (Eq. 21) sets Eq. 40 or optionally uses the IDF-based budget (Eq. 31) to modulate $M$ by redundancy. In HDLSS regimes with $m \gg n$, we estimate $\hat{d}$ via the $n \times n$ Gram matrix or randomized methods to avoid forming a full $m \times m$ covariance. In all cases, $M$ scales with intrinsic rather than ambient dimension, so NSC compresses more aggressively when the effective rank is low.

$$M = \text{clip}\big(\lceil 2\hat{d}\rceil,\ 32,\ \min(512, m)\big) \tag{40}$$

### D.2. Relation to Classical Dimensionality Reduction

NSC is conceptually related to, but distinct from, standard DR techniques.

**PCA and RP.**    PCA (Hotelling, 1933) computes a global linear projection $x \mapsto U_k^\top x$ that optimizes variance preservation in $\mathbb{R}^k$. RP maps $x \mapsto Rx$ with a dense random matrix $R$ and approximately preserves pairwise distances by Johnson–Lindenstrauss guarantees (Johnson & Lindenstrauss, 1984). Both treat coordinates symmetrically and mix all features into each latent dimension. By contrast, NSC first reorders features via GO-LR so that correlated variables are neighbors, then pools each contiguous neighborhood into a meta-feature. This yields:

- **Local support:** Each latent coordinate depends on a small, interpretable block of features rather than all $m$.

- **Order-aware pooling:** GO-LR ensures that each block tends to group features with small dispersion under the MinLA objective, so pooled statistics are computed over structurally coherent neighborhoods.

- **Flexible statistics:** NSC variants can incorporate robust or higher-order statistics via $\psi$ without changing the dimensionality $M$. NSC variants also inherit the power of PCA for IDF or post segmentation summarization.

In the special case of linear $\psi$ and $g_\theta$, NSC is a constrained linear DR method whose projection matrix has a block structure aligned with the GO-LR ordering, whereas PCA uses a dense orthogonal mixing and RP uses an unstructured dense matrix.

**Autoencoders.**    Autoencoders (AE) (Hinton & Salakhutdinov, 2006) learn a parametric encoder-decoder pair $(f_\phi, h_\psi)$ with a low-dimensional bottleneck $z \in \mathbb{R}^k$ optimized to minimize reconstruction error. While AE can capture nonlinear structure, they require training, are sensitive to sample size, and their bottleneck dimensions often entangle information from all features. NSC can be seen as a deterministic, data-dependent encoder with no decoder: it compresses features into $M$ meta-features using shared, shallow pooling and no reconstruction objective. In HDLSS regimes, NSC has two advantages: (i) it does not require fitting a heavy parametric model on small $n$, and (ii) its pooling structure is constrained by GO-LR ordering, which reduces overfitting and enforces a biologically motivated inductive bias.

**UMAP.**    UMAP (McInnes et al., 2018) is a nonlinear manifold-learning method that constructs a neighborhood graph and optimizes a low-dimensional embedding to preserve local topological structure. Although this makes UMAP effective for visualization and neighborhood preservation, the learned coordinates are global embedding dimensions whose relation to the original features is indirect. In contrast, NSC preserves an explicit feature-axis interpretation: each meta-feature is computed from a contiguous GO-LR-induced neighborhood, making the compression more directly tied to feature locality and block structure in HDLSS settings.

**PaCMAP.** PaCMAP (Wang et al., 2021) is another nonlinear DR method that balances nearby, mid-near, and far pair relationships to preserve both local and global geometry in the embedded sample space. Like UMAP, it learns low-dimensional coordinates over samples rather than constructing interpretable feature-block summaries. NSC instead operates along the feature dimension: after GO-LR induces a locality-aware ordering, NSC pools contiguous feature segments into meta-features, which is better aligned with our goal of preserving order-induced feature neighborhoods for downstream TabPFN-style prediction.

### D.3. Empirical Behavior on HDLSS Benchmarks

We instantiate the protocol below on block structured synthetic HDLSS datasets, using $M = 32$ as an aggressive DR setting. For each dataset we compare NSC (GO-LR ordering + Optuna (Akiba et al., 2019) tuned segmentation, descriptor, and pooling) with PCA, AE, UMAP, PaCMAP, and RP using:

- Linear-probe accuracy (Logistic Regression) (Cox, 1958),

- $k$NN accuracy in latent space (Cover & Hart, 1967),

- Silhouette (Rousseeuw, 1987) and Davies–Bouldin (Davies & Bouldin, 1979) scores for class separability.

### D.4. Block-Structured Synthetic HDLSS Model

To better understand when NSC should dominate classical DR, we also study a synthetic family of HDLSS distributions with explicit block structure (Ver Steeg et al., 2019). This controlled ablation is inspired by the block-subunit HDLSS modeling philosophy of BSTabDiff (Habib et al., 2026a), which views high-dimensional tabular data as arising from latent feature blocks governed by shared subunit factors. In our setting, we adapt this idea only as a diagnostic synthetic benchmark: features are generated from block-correlated latent factors, then randomly permuted, allowing us to test whether GO-LR + NSC can recover useful local neighborhoods for compression and downstream prediction. Concretely, we construct datasets where $m$ features are partitioned into $B$ contiguous blocks, each governed by a shared latent factor plus Gaussian noise (Devijver & Gallopin, 2018). Class information is injected via mean shifts on a small subset of blocks, while the remaining blocks are pure noise. After generation, we randomly permute feature indices so that class-informative blocks are no longer contiguous in the raw feature space (Simon & Tibshirani, 2012). Under this model, GO-LR with a correlation-based metric tends to recover an ordering that approximately re-groups highly correlated features into contiguous neighborhoods, effectively reconstructing the underlying blocks. NSC then segments along this ordering and pools each block into one or a few meta-features. As a result, the NSC embedding approximates block-level sufficient statistics (block means and low-order summaries), whereas PCA, AE, RP, UMAP, and PaCMAP operate through global mixing, parametric encoding, random projection, or sample-space manifold embedding, and have no explicit bias toward recovering feature-block boundaries.

We instantiate this block-structured synthetic model in the HDLSS regime ($n \ll m$) and compare four NSC variants (NSC, NSC-P, NSC-SP, NSC-pSP) against PCA, AE, RP, UMAP, and PaCMAP using the same evaluation suite as above (linear and $k$NN probe accuracy, silhouette, and Davies-Bouldin; see section D.3). Over 10 independent repetitions (Table D.1), the NSC family achieves the best mean result on all four metrics: NSC-pSP obtains the highest mean linear accuracy ($0.8488 \pm 0.0406$), best silhouette ($0.0460 \pm 0.0257$), and lowest DB ($4.1198 \pm 0.9657$), while NSC-SP achieves the highest mean $k$NN accuracy ($0.7313 \pm 0.0722$). Thus, NSC-pSP is the strongest overall variant, with NSC-SP providing the best neighborhood-probe performance (Table D.1, Panel B). Consistently, Friedman tests detect significant differences across the nine methods for all metrics (all $p \le 9.79 \times 10^{-5}$; Table D.2). Using one-sided Wilcoxon tests with NSC-pSP as the reference, NSC-pSP shows a clear linear-probe advantage over all baselines and NSC variants, including PCA ($p=9.77 \times 10^{-4}$), AE ($p=0.003906$), RP ($p=9.77 \times 10^{-4}$), UMAP ($p=0.004883$), PaCMAP ($p=0.002930$), NSC ($p=0.001953$), NSC-P ($p=0.003906$), and NSC-SP ($p=0.03711$).

For neighborhood-based structure, NSC-pSP significantly improves $k$NN accuracy over PCA and RP (both $p=0.001953$), NSC ($p=0.04492$), and NSC-P ($p=0.004883$), while differences versus AE ($p=0.3467$), NSC-SP ($p=0.9561$), UMAP ($p=0.6152$), and PaCMAP ($p=0.5771$) are not significant at $\alpha=0.05$. For clustering separability, NSC-pSP yields higher silhouette than PCA, RP, and NSC-P (all $p=9.77 \times 10^{-4}$), AE ($p=0.002930$), and NSC ($p=0.01953$), whereas gains versus NSC-SP, UMAP, and PaCMAP are not significant. Finally, NSC-pSP achieves significantly lower DB than PCA, RP, and NSC-P (all $p=9.77 \times 10^{-4}$), AE ($p=0.002930$), NSC ($p=0.009766$), and PaCMAP ($p=0.01367$), with no significant

difference versus NSC-SP or UMAP. Overall, this controlled experiment highlights that the PCA-IDF + segment-wise PCA tokenization in NSC-pSP better matches the block-correlated generative structure, yielding improved predictive separability and strong local/clustering structure relative to both classical and nonlinear DR baselines, as well as earlier NSC variants. Table D.3 further shows that NSC-pSP consistently outperforms PCA, AE, UMAP, and PaCMAP on linear accuracy, with positive $\Delta$ CIs and favorable effect sizes; it also clearly improves DB over PCA, AE, and PaCMAP, while kNN accuracy and silhouette are essentially tied against UMAP/PaCMAP.

### D.4.1. A STYLIZED BLOCK–SUBUNIT HDLSS MODEL

Inspired by BSTabDiff (Habib et al., 2026a), we formalize a simple generative model that matches the inductive bias of GO-LR + NSC.

**Definition D.1** (Block–subunit HDLSS model). Let $m$ features be partitioned into $M$ disjoint blocks $\{S_t\}_{t=1}^{M}$ of equal size $s$, so that $m = Ms$ and $S_t \subset \{1, \ldots, m\}$, $S_t \cap S_{t'} = \emptyset$ for $t \neq t'$. For each block $t$, we define a latent block signal $h_t \in \mathbb{R}$ and independent noise variables $\{\epsilon_j\}_{j \in S_t}$ with Eq. 41. The observed features are defined by Eq. 42. The label $Y$ is conditionally independent of $X$ given the block signals in Eq. 43.

$$\epsilon_j \sim \mathcal{N}(0, \sigma^2), \quad \text{i.i.d. across } j \tag{41}$$

$$X_j = h_t + \epsilon_j, \qquad j \in S_t, \ t = 1, \ldots, M \tag{42}$$

$$P(Y \mid X) = P(Y \mid h_1, \ldots, h_M) \tag{43}$$

We consider a setting where GO-LR, applied with a correlation-based metric, recovers an ordering $\Pi^*$ that makes the blocks contiguous (up to permutation of the blocks). NSC then segments the GO-LR axis so that each segment $S_t$ corresponds to one block.

**NSC configuration in the block model.** In this setting, NSC with: (i) segmentation aligned with the blocks $\{S_t\}_{t=1}^{M}$, and (ii) a simple mean descriptor $\psi(u_t) = \frac{1}{|S_t|} \sum_{j \in S_t} u_t[j]$ with identity pooling $g_\theta(v) = v$, produces meta-features by Eq. 44. Stacking across $t$ yields the NSC embedding $\Phi_{\text{NSC}}(X) = (z_1, \ldots, z_M) \in \mathbb{R}^M$.

$$z_t = \frac{1}{|S_t|} \sum_{j \in S_t} X_j, \qquad t = 1, \ldots, M \tag{44}$$

**Proposition D.2** (Block means are sufficient for Bayes classification). *Consider the block model in Definition D.1 in a two-class mean-shift setting where Eq. 45 with $\sigma^2$ known and $\mu_{t,0}, \mu_{t,1}$ possibly varying across blocks $t$. Then for each block $S_t$ the block sum (Casella & Berger, 2024) in Eq. 46 is a sufficient statistic for $(\mu_{t,0}, \mu_{t,1})$, and the joint log-likelihood ratio (Neyman & Pearson, 1933) for $Y$ based on all features $X$ depends only on the collection $\{S_t\}_{t=1}^{M}$. Consequently, any Bayes-optimal classifier (Devroye et al., 1996) based on $X$ can be written as a function of the NSC block means $z_t = S_t/s$.*

$$X_j \mid Y = y \ \sim \ \mathcal{N}(\mu_{t,y}, \sigma^2), \qquad j \in S_t, \ t = 1, \ldots, M, \ y \in \{0, 1\} \tag{45}$$

$$S_t := \sum_{j \in S_t} X_j \tag{46}$$

*Proof sketch.* Within each block $t$, conditional on $Y = y$, the observations $\{X_j\}_{j \in S_t}$ are i.i.d. Gaussian with mean $\mu_{t,y}$ and variance $\sigma^2$. The joint likelihood within block $t$ is defined by Eq. 47. Rewriting the exponent shows that this likelihood depends on the data only through the block sum $S_t = \sum_{j \in S_t} X_j$ (equivalently the block mean $z_t$). Thus $S_t$ (or $z_t$) is a sufficient statistic for $\mu_{t,y}$ in the exponential-family sense. Across blocks, conditional independence implies Eq. 48 so the global log-likelihood ratio $\log p(X \mid Y = 1) - \log p(X \mid Y = 0)$ depends on $X$ only through $\{S_t\}_{t=1}^{M}$, i.e., only through $\{z_t\}_{t=1}^{M}$. Therefore any Bayes-optimal decision rule $\text{sign}(\log p(Y = 1 \mid X) - \log p(Y = 0 \mid X))$ can be written as a function of $(z_1, \ldots, z_M)$.

$$p(X_{S_t} \mid Y = y) = \prod_{j \in S_t} \frac{1}{\sqrt{2\pi\sigma^2}} \exp\left( -\frac{(X_j - \mu_{t,y})^2}{2\sigma^2} \right) \tag{47}$$

$$p(X \mid Y = y) = \prod_{t=1}^{M} p(X_{\mathcal{S}_t} \mid Y = y) \tag{48}$$

$\square$

*Remark* D.3. Proposition D.2 exhibits a family of HDLSS distributions where there exists an NSC configuration (aligned segments, block means) such that the $M$-dimensional NSC embedding $\Phi_{\mathrm{NSC}}(X)$ is information-preserving for Bayes classification, even though the ambient dimension $m = Ms$ can be arbitrarily large.

**Lemma D.4** (SNR gain of NSC vs Random Projection in a block). *Consider one block $\mathcal{S}$ of size $s$ under a simple two-class Gaussian mean-shift model (Eq. 49) with independent coordinates and fixed $\Delta \neq 0$, $\sigma^2 > 0$ (Dasgupta & Gupta, 2003; Devroye et al., 1996; Johnson & Lindenstrauss, 1984).*

$$X_j \mid Y = 0 \sim \mathcal{N}(0, \sigma^2), \qquad X_j \mid Y = 1 \sim \mathcal{N}(\Delta, \sigma^2), \qquad j \in \mathcal{S} \tag{49}$$

1. *The NSC block mean (Eq. 50) has class-conditional mean difference $\mathbb{E}[z_{\mathrm{NSC}} \mid Y = 1] - \mathbb{E}[z_{\mathrm{NSC}} \mid Y = 0] = \Delta$ and variance $\mathrm{Var}(z_{\mathrm{NSC}} \mid Y) = \sigma^2/s$, hence we get Eq. 50 and 51*

$$z_{\mathrm{NSC}} = \frac{1}{s} \sum_{j \in \mathcal{S}} X_j \tag{50}$$

$$\mathrm{SNR}_{\mathrm{NSC}} := \frac{\left(\mathbb{E}[z_{\mathrm{NSC}} \mid Y = 1] - \mathbb{E}[z_{\mathrm{NSC}} \mid Y = 0]\right)^2}{\mathrm{Var}(z_{\mathrm{NSC}} \mid Y)} = \frac{\Delta^2 s}{\sigma^2} \tag{51}$$

2. *Let $u = (u_j)_{j \in \mathcal{S}}$ be a random projection direction with $\sum_{j \in \mathcal{S}} u_j^2 = 1$ and entries of order $1/\sqrt{s}$, and define Eq. 52. Then we get Eq. 52 and Eq. 53. For typical random $u$, $\mathbb{E}\left[\left(\sum_j u_j\right)^2\right] = 1$, so the typical SNR of $z_{\mathrm{RP}}$ is defined by Eq. 54.*

$$z_{\mathrm{RP}} = \sum_{j \in \mathcal{S}} u_j X_j \tag{52}$$

$$\mathbb{E}[z_{\mathrm{RP}} \mid Y = 1] - \mathbb{E}[z_{\mathrm{RP}} \mid Y = 0] = \Delta \sum_{j \in \mathcal{S}} u_j, \qquad \mathrm{Var}(z_{\mathrm{RP}} \mid Y) = \sigma^2 \tag{53}$$

$$\mathrm{SNR}_{\mathrm{RP}} := \frac{\left(\mathbb{E}[z_{\mathrm{RP}} \mid Y = 1] - \mathbb{E}[z_{\mathrm{RP}} \mid Y = 0]\right)^2}{\mathrm{Var}(z_{\mathrm{RP}} \mid Y)} \approx \frac{\Delta^2}{\sigma^2} \tag{54}$$

*Consequently, in this block model (Eq. 55 i.e., NSC's block mean enjoys an SNR gain of a factor $s$ over a typical random projection coordinate.*

$$\frac{\mathrm{SNR}_{\mathrm{NSC}}}{\mathrm{SNR}_{\mathrm{RP}}} \approx s \tag{55}$$

*Proof sketch.* Part (1) follows by linearity of expectation and the variance of the average of $s$ i.i.d. Gaussians. For part (2), the mean and variance of $z_{\mathrm{RP}}$ are obtained by linearity and the constraint $\sum_j u_j^2 = 1$. For random $u$ with roughly i.i.d. components of variance $1/s$, we have $\mathbb{E}[(\sum_j u_j)^2] = s \cdot \mathbb{E}[u_j^2] = 1$, so the typical squared signal is $\Delta^2$, leading to the stated SNR. The ratio then simplifies to $s$. $\square$

*Remark* D.5. Lemma D.4 shows that, even at the level of a single block, NSC pooling yields a multiplicative SNR improvement of order $s$ over random projections. In HDLSS regimes where classification is heavily SNR-limited, this translates into a substantial advantage in sample efficiency and Bayes error.

**Proposition D.6** (HDLSS stability of NSC block statistics). *In the block model of Definition D.1, suppose the block size $s$ is fixed while the ambient dimension $m = Ms$ may grow. For each block $\mathcal{S}_t$, the NSC block mean (Eq. 56) satisfies Eq. 57 and its empirical estimate based on $n$ samples converges to its population value at rate $O_p(n^{-1/2})$, independently of $m$. By contrast, global linear DR methods such as PCA or dense linear encoders must estimate directions in $\mathbb{R}^m$ based on the empirical covariance matrix or large weight matrices of size $O(m)$ or $O(m^2)$, which is known to be unstable in the HDLSS regime $m \gg n$ without strong structural assumptions (Jung & Marron, 2009; Yata & Aoshima, 2009; Strawderman, 2014;*

*Cover & Thomas, 2006). Thus, in this stylized setting, NSC's local block statistics remain well-conditioned as $m$ grows, while unconstrained global DR can become ill-posed.*

$$z_t = \frac{1}{s} \sum_{j \in \mathcal{S}_t} X_j \tag{56}$$

$$z_t = h_t + \bar{\epsilon}_t, \qquad \bar{\epsilon}_t := \frac{1}{s} \sum_{j \in \mathcal{S}_t} \epsilon_j \sim \mathcal{N}(0, \sigma^2/s) \tag{57}$$

### D.5. Comparison and Evaluation Protocol

To empirically position NSC as a structured DR layer, we adopt the following protocol for both real and synthetic experiments.

**Experimental setup.** For each dataset (real HDLSS or synthetic block-structured):

1. **GO-LR ordering.** Compute the GO-LR global feature permutation $\Pi^*$ on the training set using a chosen metric (e.g., correlation, cosine, euclidean, manhattan, or KL divergence), with local refinement passes as in Algorithm 1.

2. **Intrinsic dimension.** Estimate $\hat{d}$ via effective rank (Eqs. 27-29), and compute IDF $= \hat{d}/m$ (Eq. 30).

3. **NSC configuration.** Configure NSC with $\Pi^*$, using:
   - *Default compression:* $M$ chosen by Eq. 21 (e.g., yielding $M \approx 2\hat{d}$ and resulting in tens to a few hundred meta-features).
   - *Aggressive compression:* fixed $M \in \{32, 64\}$ to emulate strong dimensionality reduction.

   Apply NSC to obtain embeddings $X_{\text{NSC}} \in \mathbb{R}^{n \times M}$.

**Baselines.** For each dataset and target dimension $M$ (e.g., $M = 32$), we construct the following DR baselines:

- **PCA**: compute $k$ principal components with $k = M$, producing $X_{\text{PCA}} \in \mathbb{R}^{n \times M}$.

- **RP**: apply a dense Gaussian projection $R \in \mathbb{R}^{M \times m}$, normalized to preserve variance, yielding $X_{\text{RP}} = XR^\top$.

- **AE**: train a shallow autoencoder with bottleneck dimension $M$ on the training set, and use the bottleneck activations $X_{\text{AE}} \in \mathbb{R}^{n \times M}$ as the DR representation.

- **UMAP**: learn a nonlinear manifold embedding with target dimension $M$, preserving local neighborhood structure in the sample space and producing $X_{\text{UMAP}} \in \mathbb{R}^{n \times M}$.

- **PaCMAP**: learn a nonlinear embedding with target dimension $M$ by balancing nearby, mid-near, and far pair constraints, yielding $X_{\text{PaCMAP}} \in \mathbb{R}^{n \times M}$.

**NSC variants and naming conventions.** We evaluate a family of **NSC** variants that share a common pipeline (i) determine an IDF to set the internal granularity which is a ratio of Intrinsic Dimension (ID) to the actual dimension, (ii) segment the feature sequence into contiguous blocks, and (iii) compress each block into an $M$-dimensional representation, but differ in how IDF is estimated and how each segment is summarized. We denote the default variant as NSC, which uses an effective-rank (data-driven) ID estimate and applies statistical descriptor-based pooling within each segment. To isolate the impact of a PCA-inspired ID heuristic while keeping the same descriptor-based segment summarization, we use **NSC-P** (also written as NSC-PCA). To study the role of replacing descriptors with explicit linear projection inside segments, we define **NSC-SP** (also written as NSC-SegPCA), which retains the effective rank-based IDF but applies PCA within each segment after segmentation. Finally, our proposed **NSC-pSP** (also written as NSC-PIDF-SegPCA) combines both modifications: a PCA-inspired IDF estimate together with per-segment PCA compression after segmentation. In all cases, the suffixes indicate the modification relative to NSC: **P** denotes PCA-inspired IDF, **SP** denotes segmented per-block PCA, and **pSP** denotes the combination of both.

**Quantitative comparisons.**    To compare NSC variants against PCA/RP/AE/UMAP/PaCMAP, we use:

1. **Linear-probe accuracy.**    Train a logistic regression classifier on each latent space $X_{\text{NSC}}, X_{\text{PCA}}, X_{\text{RP}}, X_{\text{AE}}, X_{\text{UMAP}}, X_{\text{PaCMAP}}$ using the same stratified cross-validation protocol. This measures how well each DR method preserves label-relevant structure in $M$ dimensions.

2. $k$**NN classification in latent space.** Evaluate $k$NN accuracy using the compressed embeddings under the same stratified cross-validation protocol to assess neighborhood quality.

3. **Label-based separability metrics.** Compute silhouette and Davies-Bouldin scores on each latent representation using ground-truth class labels as the partition.

4. **Statistical comparison across datasets.** Aggregate per-dataset metrics and apply nonparametric tests, using a Friedman test (Friedman, 1937) across methods followed by one-sided Wilcoxon signed-rank comparisons with NSC-pSP as the reference (see Tables D.1, D.2, D.3).

**Evaluation protocol.**    We report only quantitative DR-style evaluations under a fixed aggressive budget of $M{=}32$. For each method (NSC variants, PCA, AE, RP, UMAP, PaCMAP), we first compute a 32-dimensional latent representation in $\mathbb{R}^{32}$, and then assess (i) linear-probe accuracy via logistic regression, (ii) $k$NN accuracy in latent space, and (iii) label-based separability via silhouette and Davies-Bouldin scores, using the same stratified cross-validation protocol across methods (Tables D.1-D.2). Overall, these experiments treat NSC as a first-class dimensionality reduction layer: it produces an explicit $\mathbb{R}^M$ representation that can be consumed directly by TabPFN-style predictor under strict token budgets, while still supporting standard DR comparisons to PCA/RP/AE/UMAP/PaCMAP via probe and clustering metrics. In HDLSS settings, NSC couples GO-LR's MinLA-motivated ordering with subunit-wise pooling to achieve (i) aggressive compression ($M \ll m$) and (ii) a locality-preserving structured embedding. Empirically, NSC is typically competitive with PCA, AE, UMAP, and PaCMAP and consistently outperforms RP on real HDLSS datasets; on the block-structured synthetic model, the NSC family yields the best mean results across the evaluated metrics (Table D.1) and shows significantly stronger predictive, neighborhood, and separability structure in several comparisons (Table D.2), highlighting a regime where its locality bias matches the data-generating process.

**Synthetic block-model validation and takeaways.**    To connect the empirical trends to our theoretical picture, we evaluate NSC as an explicit DR layer on a synthetic HDLSS block model aligned with its inductive bias: $m$ features are generated in $B{=}40$ latent-factor blocks (shared block signal + noise), class information is injected via mean shifts on a small subset of blocks, and then feature indices are randomly permuted to destroy contiguity in the raw space. Under an aggressive budget ($M{=}32$) and over 10 independent repetitions (Table D.1), NSC-style embeddings remain strongly discriminative while preserving neighborhood/cluster structure competitively against unstructured and nonlinear DR: nonparametric tests confirm significant differences across methods and show consistent gains in latent-space geometry (e.g., higher $k$NN accuracy versus PCA/RP and improved separability via silhouette/DB; Table D.2). Overall, these results reinforce the DR interpretation of NSC: GO-LR constructs a locality-revealing axis, and NSC compresses it via subunit-wise pooling into interpretable meta-features whose coordinates summarize coherent redundant neighborhoods, contrasting with the global mixing of PCA/RP, many AE encoders, and sample-space manifold embeddings such as UMAP/PaCMAP. Thus, its primary advantage under strong compression is retaining local geometry and cluster structure while staying competitive in discriminative accuracy; among variants, NSC-pSP is the most consistently competitive on the block model (Table D.3), while on real HDLSS benchmarks at the same $M{=}32$ budget it is broadly comparable to PCA/AE/UMAP/PaCMAP with smaller, dataset-dependent differences.

**GO-LR+NSC vs. PCA.**    Global PCA (Hotelling, 1933) $\rightarrow$ TabPFN-2.5 (Grinsztajn et al., 2025) is a natural control for testing whether the gains of GOTabPFN come merely from dimensionality reduction. We therefore compare GO-LR+NSC+TabPFN-2.5 against Global PCA+TabPFN-2.5 while keeping the same frozen TabPFN-2.5 predictor and changing only the front-end compression interface. As shown in Table D.4, GO-LR+NSC outperforms Global PCA on all 8 HDLSS datasets, suggesting that the improvement is not explained by generic global compression alone, but by locality-aware ordering and structured neighborhood compression.

**GO-LR+NSC vs. Lasso-selected features.**    We compare GO-LR+NSC against Lasso-selected features under the same frozen TabPFN-2.5 (Grinsztajn et al., 2025) predictor and identical $5 \times 5$ CV protocol. Specifically, we evaluate

Lasso+TabPFN-2.5 (LT) across sparsity levels $C \in \{0.01, 0.02, 0.05\}$. As shown in Table D.5, GOTabPFN outperforms all LT variants on most HDLSS datasets, while LT only slightly exceeds GOTabPFN on SMK and AML under the weakest sparsity setting ($C = 0.05$). This suggests that locality-aware feature ordering and structured neighborhood compression provide more robust gains than sparsity-based feature selection alone in HDLSS settings.

## E. Why Feature Ordering? Local Neighborhoods Enable Structure-Aware Compression

A common critique is that tabular features form an unordered set, so learning a column order may appear arbitrary. In this work, ordering is not introduced to impose a fictional semantics (e.g., time), but to construct an algorithmic coordinate system: a 1D axis on which local neighborhoods become meaningful via a structure-revealing layout objective from seriation / graph layout (rather than positional meaning) (Arabie & Hubert, 1992; Hahsler et al., 2008; Díaz et al., 2002; Atkins et al., 1998). This is essential for our pipeline because NSC is an explicitly local operator it pools contiguous segments into meta-features (Sec. 3.2). Without an ordering that places related features nearby, contiguity-based pooling becomes arbitrary aggregation and can destroy predictive signal. Therefore, feature ordering is primarily justified as a neighborhood construction mechanism that enables structured compression and stable tokenization under tight budgets.

### E.1. Ordering Constructs Coherent Neighborhoods That NSC Can Pool

NSC partitions the ordered feature axis into $M$ contiguous segments and pools each segment into a token (Sec. 3.2). This implicitly assumes that adjacency along the axis corresponds to statistical relatedness. GO-LR is designed exactly to enforce this locality: it approximately minimizes a MinLA/seriation-style dispersion objective that penalizes placing strongly related feature pairs far apart, aligning index locality with a similarity graph (Díaz et al., 2002; Atkins et al., 1998; Garey et al., 1974). As a result, features that are close under the global dissimilarity structure become near neighbors in index, so NSC pooling operates on coherent neighborhoods and yields compressed tokens that preserve informative local structure. In contrast, if features are left in raw or arbitrary order, contiguous segments mix unrelated variables and pooling becomes a lossy averaging operation. This is precisely the failure mode we can worry about: ordering only matters if downstream modules use contiguity. Since NSC explicitly uses contiguity, ordering becomes a necessary part of the representation interface.

### E.2. Ordering Improves Compression by Reducing Cross-Segment Boundary Cuts

The role of ordering can be formalized through the segmentation-induced boundary cut. Let $\bar{W} \in \mathbb{R}^{m \times m}$ denote the global dissimilarity matrix used to define neighborhoods (Sec. 3.2), and let $\Pi^*$ be the learned ordering. Consider a segmentation into $M$ contiguous segments $\{\mathcal{S}_t\}_{t=1}^{M}$ along the ordered axis. Define the cross-segment boundary cost by Eq. 58. Intuitively, Cut measures how often highly related features are split across segment boundaries; the definition parallels cut objectives used in graph-based segmentation/layout (Shi & Malik, 2000; Díaz et al., 2002). A smaller cut indicates that each segment captures a coherent neighborhood, so pooled tokens retain structure. Since GO-LR reduces dispersion, it typically also reduces boundary cuts for reasonable segmentations; random or raw orders inflate boundary cuts, making local pooling ineffective.

$$\mathrm{Cut}(\Pi^*, \{\mathcal{S}_t\}) = \sum_{t=1}^{M-1} \sum_{i \in \mathcal{S}_t} \sum_{j \in \mathcal{S}_{t+1}} \bar{W}_{\Pi^*(i), \Pi^*(j)} \tag{58}$$

### E.3. What Ordering Is Not: We Do Not Make Tabular Data "Sequential"

We do not claim that tabular columns possess an intrinsic order like words or pixels. Rather, we learn an order as a layout that makes neighborhood-based operators (pooling, segmentation, local filters) well-defined. This mirrors classic seriation/linear arrangement goals: the objective is not positional semantics, but an ordering that makes locality meaningful for downstream computation (Arabie & Hubert, 1992; Hahsler et al., 2008; Atkins et al., 1998; Díaz et al., 2002). In our case, ordering is valuable specifically because NSC is local along the constructed axis.

### E.4. Empirical Diagnostics for Neighborhood Preservation (Order-Only)

We report order-only diagnostics that quantify whether an ordering induces meaningful local neighborhoods independently of any classifier or tokenizer. All diagnostics are computed on the 8 HDLSS datasets using standardized features. Since explicitly materializing a dense pairwise dissimilarity/Gram matrix scales as $\mathcal{O}(m^2)$ in memory (and time), forming

$\bar{W} \in \mathbb{R}^{m \times m}$ quickly becomes impractical at HDLSS dimensions (e.g., gene-expression arrays with $\sim 10^4$-$10^4.5$ genes and GWAS with $\geq 10^5$ SNP markers). (Si et al., 2017; Dangond, 2000; Maguire et al., 2018), we use a lightweight proxy $\bar{W}_{ij} \triangleq 1 - |\text{corr}(x_i, x_j)|$, estimated from the standardized data (and for adjacent deltas from a small row subset for stability/speed) where lower is better, indicating that neighbors along the axis are more mutually similar. Across the 8 datasets, $\Pi^*$ achieves significantly lower adjacency dissimilarity than random permutations (paired by dataset; Wilcoxon signed-rank $p = 0.00390625$; Fig. E.1d,f).

**Local adjacency coherence (path-length objective).** Given a feature ordering $\Pi^*$ and a dissimilarity matrix $\bar{W}$, we define the adjacent dissimilarity $\delta_t = \bar{W}_{\Pi^*(t), \Pi^*(t+1)}$. We define the adjacency coherence as the mean adjacent dissimilarity along the ordering, which is the Hamiltonian path (TSP-path) length objective used in seriation, up to normalization (Hahsler et al., 2008). We measure $\text{AdjCoh}(\Pi^*)$ by Eq. 59.

$$\text{AdjCoh}(\Pi^*) = \frac{1}{m-1} \sum_{t=1}^{m-1} \delta_t = \frac{1}{m-1} \sum_{t=1}^{m-1} \bar{W}_{\Pi^*(t), \Pi^*(t+1)} \tag{59}$$

**Neighborhood hit-rate for top-$k$ neighbors.** For each feature $i$, let $\mathcal{N}_k(i)$ be its top-$k$ nearest neighbors under $\bar{W}$. For an ordering $\Pi^*$, define the window neighborhood $\mathcal{W}_h(i) = \{j : |\text{pos}_{\Pi^*}(j) - \text{pos}_{\Pi^*}(i)| \leq h\}$. We compute $\text{HitRate}_{k,h}(\Pi^*)$ by Eq. 60 where higher is better. To avoid $O(m^2)$ complexity, we compute $\mathcal{N}_k(i)$ on a feature subsample (default 2048 features) and average over features and (when applicable) multiple random seeds. $\Pi^*$ yields consistently higher hit-rate than random orderings (Wilcoxon signed-rank $p = 0.0078125$ for a representative $(k, h) = (10, 16)$; Fig. E.1c,e,f), and the advantage persists over multiple $(k, h)$ choices (Fig. E.1e) (Venna & Kaski, 2001).

$$\text{HitRate}_{k,h}(\Pi^*) = \frac{1}{m} \sum_{i=1}^{m} \frac{|\mathcal{N}_k(i) \cap \mathcal{W}_h(i)|}{k} \tag{60}$$

**Segmentation boundary cut.** To test alignment between the ordering and contiguity-based pooling, we evaluate a boundary-cut proxy under common segmentation rules (uniform, equal-mass, largest-jump) with $M=32$ and $l_{\min}=8$ (matching our NSC configuration). Given segments $\{\mathcal{S}_t\}$ along $\Pi^*$, we summarize the average dissimilarity at segment boundaries via the adjacent deltas(Eq. 61) where $\mathcal{B}$ are boundary indices between consecutive segments. Lower cut indicates that segment boundaries fall on weaker connections, i.e., stronger within-segment neighborhood coherence. Across datasets, $\Pi^*$ yields favorable boundary alignment relative to random permutations (Fig. E.1b) (Shi & Malik, 2000).

$$\text{Cut}(\Pi^*, \{\mathcal{S}_t\}) \approx \frac{1}{|\mathcal{B}|} \sum_{b \in \mathcal{B}} \delta_{b-1} \tag{61}$$

### E.5. Ablations That Isolate Why Ordering Helps NSC

To separate the effect of ordering from the effect of compression/tokenization, we evaluate NSC under controlled ordering perturbations while keeping the tokenizer/compressor fixed (same $M$, segmentation rule, pooling, and tuned hyperparameters). We use five random seeds for permutation-based controls.

**Ordered vs. un-ordered.** We compare the same NSC configuration under: (i) GO-LR order $\Pi^*$, (ii) the raw/original column order, (iii) random permutations (averaged over seeds). This directly tests whether NSC benefits from neighborhood structure rather than from compression alone. In parallel, the order-only diagnostics (AdjCoh/HitRate/Cut) show that $\Pi^*$ is systematically more neighborhood-preserving than random, providing a mechanistic explanation for the observed gains (Fig. E.1c,d,f).

**Destroy global layout while partially preserving locality (block shuffle).** Starting from $\Pi^*$, we partition indices into contiguous blocks of size $b$ and randomly permute the blocks. This preserves within-block neighborhoods but disrupts long-range arrangement. If NSC relies primarily on local neighborhoods, performance/diagnostics should improve as $b$ increases. Consistent with this, normalized hit-rate increases monotonically with block size: $0.740 \pm 0.184$ ($b=8$), $0.889 \pm 0.177$ ($b=16$), $0.968 \pm 0.097$ ($b=32$), $1.011 \pm 0.095$ ($b=64$), relative to the corresponding $\Pi^*$ value per dataset (Fig. E.1a,f). This supports the locality hypothesis: preserving larger local neighborhoods recovers the $\Pi^*$ advantage.

**Keep order fixed, break contiguity (round-robin segments).** We keep $\Pi^*$ but break contiguity by assigning features to segments in a round-robin manner and then concatenating segments. This retains the same set and global ordering statistics, but destroys the contiguous neighborhood pooling assumption. The resulting drop in hit-rate (and corresponding degradation in order-sensitive behavior) isolates contiguity as the operative mechanism (Fig. E.1c,d).

**Optional representation probes.** When needed, we complement the order-only metrics with lightweight probes on the produced tokens (e.g., linear probe) under the ablations above. These probes are used only to verify that improved neighborhood preservation translates into higher-quality representations, while the primary claim remains anchored in the classifier-free diagnostics.

**Summary.** Across datasets, the learned ordering $\Pi^*$ consistently preserves local neighborhoods better than baselines: the hit-rate is highest under $\Pi^*$, typically followed by the raw order, while randomization and explicitly breaking contiguity degrade neighborhood agreement (Fig. E.1(c)); this is mirrored by adjacency coherence, where $\Pi^*$ attains the lowest (best) AdjCoh and random orderings are worst (Fig. E.1(d)). The robustness heatmap further shows that $\Pi^*$ yields uniformly positive gains over random across all tested $(k, h)$, with larger improvements for wider locality windows (larger $h$) and smaller $k$ (Fig. E.1(e)). For segmentation alignment, $\Pi^*$ tends to reduce boundary cut under "equal_mass" (and is roughly neutral under "largest_jump"), whereas "uniform" can be inconsistent and often flips the advantage (negative median $\Delta\mathrm{Cut}$), suggesting uniform boundaries may not match the induced contiguous structure (Fig. E.1(b)). Finally, the block-shuffle experiment shows a clear scale effect: when features are shuffled within small blocks, the normalized hit-rate drops substantially, but it recovers toward $\Pi^*$ as block size increases (e.g., rising from $\approx 0.67$ at $b{=}8$ to $\approx 0.90$ at $b{=}64$), indicating that locality is largely preserved within coarse blocks and mainly disrupted by fine-grained shuffles (Fig. E.1(a)).

### E.6. Beyond NSC: When Ordering Can Improve Accuracy

While our primary motivation is NSC's contiguity-based pooling, the same locality principle applies to any order-sensitive learner that introduces architectural locality over feature tokens (e.g., local attention windows, relative position bias, or convolutional mixing) (Vaswani et al., 2017; Child et al., 2019; Huang et al., 2020; Gorishniy et al., 2021; Somepalli et al., 2022). In such models, the feature order acts as a computational layout: by placing statistically related features nearby, the model concentrates informative interactions into small neighborhoods, which can reduce sample complexity and improve generalization in low-sample regimes.

**Experimental evidence (accuracy changes only when the backbone uses locality).** We run controlled ordering experiments on two biomedical $n < m$ datasets (AI-d_case5 (Ohlsson et al., 2020) and ADNI_AD123 (Petersen et al., 2010)) using the same backbone and training protocol, changing only the column order applied consistently to train/val/test. As an order-sensitive backbone, we use a local-window Transformer whose attention is restricted to a fixed neighborhood around each feature token (Beltagy et al., 2020), making performance dependent on index-locality. We evaluate multiple ordering strategies: our GO-LR order $\Pi^*$, a TabSeq-style ordering (Habib et al., 2024), the raw column order, random permutations (averaged over seeds followed by TabICL (Jingang et al., 2025)), a light version of ROTATOR (Wang et al., 2025) and controlled perturbations that partially preserve locality (block-shuffle) or explicitly destroy contiguity while keeping the same global order statistics (round-robin "break contiguity"). Figure E.2 (top) and Table E.1 show that ordering yields non-trivial AUC changes for the local-window Transformer, and GO-LR produces the strongest gains among the tested ordering methods on these datasets. We use a fixed training protocol with a single train/val/test split and fixed hyperparameters (no dataset-specific tuning or HPO); all results are produced with a fixed seed (SEED=42), except the random-permutation baseline which is averaged over 5 permutation seeds.

**Permutation-invariant sanity check.** To verify that gains are not artifacts of reindexing, we repeat the same experiment using a permutation-invariant control model (set encoder / invariant pooling) where consistent reordering should not systematically affect performance (Zaheer et al., 2017). As expected, Figure E.2 (middle) shows near-zero deltas across orderings, supporting the interpretation that improvements arise from locality-aware computation rather than from accidental leakage or inconsistent preprocessing. This perspective is also compatible with permutation-ensemble approaches such as TabICL, which averages predictions over multiple random feature permutations to approximate invariance (Jingang et al., 2025).

**Mechanistic link: better neighborhood preservation ⇒ better accuracy.** Finally, we connect accuracy changes to order-only locality diagnostics. Figure E.2 (bottom) shows that orderings with higher neighborhood hit-rate ($HitRate_{k,h}$) tend to yield higher AUC under the local-window backbone, supporting the hypothesis that ordering helps by increasing the density of meaningful local interactions. In other words, ordering can improve accuracy whenever the downstream architecture uses locality; when the architecture is invariant, ordering should not matter such as tree-based models or set-based models (Zaheer et al., 2017).

**Relation to prior ordering methods.** This finding aligns with prior work that explicitly learns or uses feature layouts to benefit order-sensitive tabular learners (e.g., TabSeq (Habib et al., 2024) ordering heuristics and ordering strategies in recent LLM-based tabular pipelines such as ROTATOR-LLM (Wang et al., 2025)).

**Summary.** Figure E.2 provides a causal/mechanistic check that ordering only matters when the backbone uses locality. In the order-sensitive local-window Transformer, GO-LR ($\Pi^*$) yields the most consistent positive $\Delta$AUC versus random across both datasets, while disrupting locality either by random permutations, breaking contiguity, or (to a lesser extent) block shuffling reduces or erases these gains; in contrast, the permutation-invariant control shows $\Delta$AUC values clustered near zero with no systematic advantage for any ordering, indicating that improvements are not an artifact of reindexing features but arise from the model's locality bias (Fig. E.2a). The mechanism plot further supports this explanation: for the local model, downstream AUC increases with neighborhood preservation ($HitRate_{k,h}$), with higher-performing orderings (e.g., GO-LR/$\Pi^*$) occupying the high-HitRate/high-AUC region, whereas random or contiguity-breaking variants sit at lower HitRate and correspondingly lower AUC (Fig. E.2b). Together, these results suggest that learned orderings improve accuracy or overall classification performance beyond NSC-based compression specifically by aligning informative feature neighborhoods with the local attention window, and that when locality is removed (invariant control), ordering ceases to provide systematic benefit (Fig. E.2).

# F. Feature Ordering - When to Use? Through the Lens of Locality

**When Feature Ordering Matters.** Deep Sets (Zaheer et al., 2017) is designed to be permutation-invariant for genuinely unordered inputs (e.g., point clouds, MIL, and chemoinformatics). In contrast, we study high-dimensional tabular settings where the chosen column layout can materially affect learning. A well-chosen permutation can reduce redundancy, expose latent dependencies among features, and ultimately improve predictive performance or help in contiguity based-process like NSC where locality matters. This is particularly pertinent for high-dimensional biological measurements (e.g., gene expression), EEG and other sensor data, remote sensing and climate datasets, and multimodal or heavily engineered feature tables domains that often exhibit sparsity, redundancy, and hidden structure, and are thus natural targets for sequence-dependent models. DynaTab (Habib et al., 2026b) initiated a systematic study of when feature ordering is useful in high-dimensional tabular learning, primarily from the perspective of ordering sensitivity and sequence-dependent modeling. We extend this view through the lens of locality: feature ordering is useful not only because sequence-sensitive backbones depend on token order, but also because a good permutation can create contiguous neighborhoods of statistically related features, enabling locality-based operators such as NSC to compress related features into informative meta-features.

**Dataset Categorization Rules.** There is no universally agreed upon numerical cutoff that uniquely determines when a dataset should be labeled HDLSS. In much of the HDLSS literature, the term is used broadly for regimes where the ambient dimension (number of variables) is far larger than the sample size, and is often formalized through HDLSS asymptotics (Aoshima et al., 2018) in which the dimension grows while the sample size is fixed (or grows much more slowly) (Hall et al., 2005; Jung & Marron, 2009). Consequently, the simple rule "$m > n$" is a useful heuristic but too coarse to capture the practical spectrum of high dimensionality. To make this notion more operational, we follow DynaTab's (Habib et al., 2026b) empirical regime stratification and use the feature-to-sample ratio $\rho = m/n$, which helps distinguish qualitatively different regimes beyond a binary HDLSS vs. non-HDLSS split. Let $n$ denote the number of samples, $m$ the number of features, and $\rho = \frac{m}{n}$ the feature-to-sample ratio. We assign each dataset to one of five regimes using the following empirical $\rho$ thresholds:

**HDLSS:** $m > 1000,\ n < 1000,\ \rho > 2.$
**HDHSS:** $m > 1000,\ n > 10^4,\ 0.005 < \rho \le 2.$
**LDHSS:** $m \le 100,\ n > 10^4,\ \rho \le 0.01.$
**LDLSS:** $m \le 100,\ n \le 1000,\ \rho \le 0.05.$
**MixedRegime:** otherwise.

### F.1. Intrinsic Dimensionality Factor as a Proxy for Locality Exploitability

Our primary justification for feature ordering in this paper is locality: ordering constructs an algorithmic 1D axis on which contiguity corresponds to statistical relatedness, making neighborhood-based operators (e.g., NSC's contiguous pooling) well-defined (Sec. 3.2, Appendix E). This motivates a complementary question: when should we expect ordering to provide tangible benefit? We connect to this locality view via the IDF. Let $m$ be the ambient number of features and let $\hat{d}$ denote an estimate of the dataset's intrinsic dimensionality (e.g., the number of principal components required to reach a fixed cumulative variance threshold, or an effective-rank estimate). In other words, the minimal number of features capturing core variability (Chen et al., 2022) to its total feature count. Following DynaTab (Habib et al., 2026b), we use Eq. 62 to compute IDF. Intuitively, IDF measures how compact the data are relative to the ambient dimension. A small IDF indicates substantial redundancy/low effective rank, which we hypothesize corresponds to stronger, more compressible correlation structure and thus a greater ability to induce local neighborhoods via a 1D layout.

$$\text{IDF} \;=\; \frac{\hat{d}}{m} \tag{62}$$

**Complexity score (IDF-normalized compactness).**   Following DynaTab (Habib et al., 2026b), we summarize dataset compactness and "ordering opportunity" by the IDF-normalized score in Eq. 63, where $\text{CumVar}(\hat{d})$ is the cumulative variance explained at $\hat{d}$ components and $p$ is a tunable sensitivity parameter. Larger values indicate that a small intrinsic subspace captures substantial variance, suggesting higher redundancy and greater potential for ordering-based locality.

$$\text{ComplexityScore} \;=\; \frac{\text{CumVar}(\hat{d})}{\text{IDF}^p} \tag{63}$$

### F.2. Feature Ordering Effectiveness and Success Probability

Following DynaTab (Habib et al., 2026b), we use the Feature Ordering Effectiveness (FOE) as a composite indicator of ordering benefit, given in Eq. 64, where $\kappa$ is a dataset-specific scaling factor and AUC denotes the area under the IDF-variance curve, estimated by trapezoidal integration over discrete IDF-variance pairs. We choose $\kappa$ by minimizing the deviation from a target value (set to 1) via Eq. 65. Setting $p{=}2$ introduces quadratic sensitivity (Hinton & Salakhutdinov, 2006), amplifying penalties when variance grows slowly with intrinsic dimension. AUC is estimated using the trapezoidal rule (Hanley & McNeil, 1982) for efficient integration of discrete IDF–variance pairs. While $\kappa$ and AUC vary by dataset, FOE preserves an inverse dependence on IDF by Eq. 66. We also report a simple success-probability proxy by Eq. 67. As $\hat{d} \to m$, $p_{\text{succ}}$ decreases, reflecting limited room for ordering to expose structure beyond what is already "fully spread" across features.

$$\text{FOE} \;=\; \frac{\kappa}{(\text{AUC} \cdot \text{IDF})^p} \tag{64}$$

$$\text{Loss}(\kappa) \;=\; \left( \frac{\kappa}{(\text{AUC})^p} - 1 \right)^2 \tag{65}$$

$$\text{FOE} \;\propto\; \frac{1}{\text{IDF}} \quad \text{(for fixed } \kappa \text{ and AUC)} \tag{66}$$

$$p_{\text{succ}} \;=\; 1 - \text{IDF} \;=\; 1 - \frac{\hat{d}}{m} \tag{67}$$

### F.3. Linking IDF/FOE to Locality: Testable Predictions

The locality view yields a mechanistic interpretation of IDF/FOE: ordering helps when the dataset admits a linear layout that concentrates strong relations into short-range neighborhoods. We formalize this using order-only locality diagnostics (Appendix E.4). Let $\Pi^*$ be the learned GO-LR ordering and let $\Pi^{(r)}$ denote random permutations.

**Locality gains relative to random orderings.** We define three locality gains by Eqs. 68, 69, 70.

$$\Delta\text{AdjCoh} = \mathbb{E}_r\Big[\text{AdjCoh}(\Pi^{(r)})\Big] - \text{AdjCoh}(\Pi^*) \tag{68}$$

$$\Delta\text{HitRate}_{K,h} = \text{HitRate}_{K,h}(\Pi^*) - \mathbb{E}_r\Big[\text{HitRate}_{K,h}(\Pi^{(r)})\Big] \tag{69}$$

$$\Delta\text{Cut} = \mathbb{E}_r\Big[\text{Cut}(\Pi^{(r)})\Big] - \text{Cut}(\Pi^*) \tag{70}$$

Positive values indicate that $\Pi^*$ induces stronger locality than random orderings: lower adjacent dissimilarity (better AdjCoh), higher neighborhood recovery (better HitRate), and lower cross-segment boundary cost (better Cut).

**Locality Exploitability Score (LES).** Optionally, we aggregate these into a single dataset-level score by z-normalizing each gain across datasets and averaging by Eq. 71 where LES measures how much local neighborhood structure a dataset allows ordering to unlock.

$$\text{LES} = \frac{1}{3}\Big(\text{zscore}(\Delta\text{AdjCoh}) + \text{zscore}(\Delta\text{HitRate}_{K,h}) + \text{zscore}(\Delta\text{Cut})\Big) \tag{71}$$

For a benchmark suite with multiple datasets, the $\text{zscore}(\cdot)$ terms in Eq. 71 are computed across the evaluated dataset collection. For single-dataset diagnostics, cross-dataset z-normalization is not defined. We therefore report the raw finite-diagnostic aggregate

$$\text{LES}_{\text{single}} = \frac{1}{|\mathcal{D}_{\text{fin}}|}\sum_{d\in\mathcal{D}_{\text{fin}}} d, \qquad \mathcal{D}_{\text{fin}} = \{\Delta\text{AdjCoh}, \Delta\text{HitRate}_{K,h}, \Delta\text{Cut}\} \cap \mathbb{R}_{\text{finite}} \tag{72}$$

Here, $\mathcal{D}_{\text{fin}}$ contains only finite locality diagnostics, so unavailable quantities such as $\Delta\text{HitRate}_{K,h}$ or $\Delta\text{Cut}$ for very small feature dimensions are omitted from the average. Thus, Eq. 71 is benchmark-relative, while Eq. 72 provides a dataset-level locality summary when only one dataset is evaluated.

**Predictions.** Under the locality hypothesis, IDF/FOE provide opportunity indicators for locality gains, rather than deterministic guarantees. We summarize this diagnostic expectation as in Eq. 73.

$$\text{IDF} \downarrow, \ \text{FOE} \uparrow, \ p_{\text{succ}} \uparrow \implies \quad \text{higher expected opportunity for locality gains} \tag{73}$$

Importantly, Eq. 73 is diagnostic rather than deterministic: IDF/FOE indicate when ordering is worth trying, while LES measures whether the learned GO-LR ordering actually realizes locality gains over random orderings.

For single-dataset diagnostics, the same opportunity-based interpretation can be applied to $\text{LES}_{\text{single}}$ from Eq. 72, but it should be treated as an empirical diagnostic rather than a guaranteed monotonic relationship. Intuitively, small IDF suggests that variance concentrates in a low-dimensional subspace, which may co-occur with stronger feature redundancy or community structure in the similarity graph. When such structure is present, GO-LR can align related features into contiguous neighborhoods that NSC can pool effectively.

### F.4. Experimental Validation Protocol (Locality as the Bridge)

We validate the bridge "When $\Rightarrow$ Why" by testing whether IDF/FOE predict order-only locality gains on the same dataset suite used throughout the paper.

**Correlation tests (dataset-level).** Across datasets, we compute Spearman correlations between IDF (or $p_{\text{succ}}$ / FOE) and each locality gain by Eq. 74.

$$\rho(\text{IDF}, \Delta\text{HitRate}_{K,h}), \ \rho(\text{IDF}, \Delta\text{AdjCoh}), \ \rho(\text{IDF}, \Delta\text{Cut}), \ \rho(\text{FOE}, \text{LES}) \tag{74}$$

We expect negative correlations for IDF (smaller IDF $\Rightarrow$ larger gains) and positive correlations for FOE and $p_{\text{succ}}$.

**Link to downstream NSC behavior.** To directly connect locality gains to NSC's contiguity-based pooling, we also test whether datasets with larger LES obtain larger ordering-induced improvements under NSC by Eq. 75.

$$\rho(\text{LES}, \Delta\text{Perf}_{\text{NSC}}), \qquad \Delta\text{Perf}_{\text{NSC}} = \text{Perf}(\text{NSC} + \Pi^*) - \mathbb{E}_r\Big[\text{Perf}(\text{NSC} + \Pi^{(r)})\Big] \tag{75}$$

This closes the chain:

low IDF / high FOE $\rightsquigarrow$ higher opportunity for locality gains $\xrightarrow{\text{validated by LES}}$ effective contiguity-based pooling (NSC)

**Summary.** We use locality as the operational criterion for deciding "when" feature ordering should help. Concretely, we first compute an opportunity proxy from intrinsic dimensionality: we estimate $\hat{d}$ from the PCA cumulative-variance curve at a fixed threshold, then we define $\text{IDF} = \hat{d}/m$, and form FOE by combining IDF with the IDF-variance curve area AUC (with $\kappa$ optimized via Eq. 65). Intuitively, HDLSS/HDHSS datasets typically exhibit very small IDF and hence large FOE (Table F.1, top ranks), indicating strong redundancy/low effective rank and thus substantial room for an ordering algorithm to expose coherent local neighborhoods. In contrast, low-dimensional or near-full-rank datasets tend to have $\text{IDF} \approx 1$ (and $P_{\text{success}} = 1 - \text{IDF} \approx 0$), suggesting limited remaining structure for ordering to uncover; in such cases, ordering is expected to be less critical unless the locality diagnostics indicate otherwise. Additionally, Table F.2 shows that orlraws10P has the strongest expected benefit from ordering, with the lowest IDF and highest FOE among the additional cross-domain datasets. Cell Cycle also exhibits a relatively high FOE despite being MixedRegime, suggesting substantial compressible structure. In contrast, RELATHE, BASEHOCK, and PCMAC have lower FOE scores, indicating that feature ordering is expected to provide more limited gains for these datasets. Second, beyond this screen, we quantify whether the opportunity is realized by the ordering algorithm: we learn a GO-LR ordering $\Pi^*$ and measure order-only locality gains against random permutations using $\Delta\text{AdjCoh}$, $\Delta\text{HitRate}_{K,h}$, and $\Delta\text{Cut}$, whose z-normalized average defines LES (Eqs. 68-71). The scatter plots (Fig. F.1) show that IDF/FOE are best interpreted as capacity measures: they separate regimes where ordering is plausibly useful (low IDF/high FOE, often HDLSS) from regimes where it is unlikely to matter (high IDF, typically low-dimensional), while LES diagnoses whether GO-LR successfully linearizes the dataset's similarity structure into short-range neighborhoods that contiguity-based operators (e.g., NSC segmentation/pooling) can exploit. In summary, ordering is most relevant in HDLSS-like regimes with low IDF or high FOE, and it is most likely to help when GO-LR also produces positive locality improvements over random orderings, reflected by higher LES.; conversely, for low-dimensional/high-IDF datasets, ordering is generally not required. Practically, we recommend a two-stage test: use low IDF / high FOE to flag datasets where ordering may help, and use positive locality gains (high LES) to predict when ordering will actually benefit architectures that rely on contiguity or local neighborhoods along the input sequence e.g., local-window attention/Transformer variants, state-space sequence models (e.g., Mamba-style SSMs), recurrent models (LSTM/GRU), and sequence-based LLM backbones since these models implicitly assume that nearby tokens/features should interact more strongly than distant ones. In our pipeline, NSC is not a backbone but a compression / dimensionality-reduction operator whose contiguous segmentation and pooling explicitly depends on meaningful neighborhoods; thus, when GO-LR induces stronger locality than random orderings (positive $\Delta\text{AdjCoh}$, $\Delta\text{HitRate}_{K,h}$, $\Delta\text{Cut}$ and higher LES), NSC-style compression, and potentially other locality-sensitive sequence models, are expected to benefit.

## G. Detailed Comparative Results

Table G.1 reports the full 5×5 cross-validation results (mean accuracy with subscripted standard deviation) for all 8 HDLSS benchmarks and the complete set of 50+ baselines. Our method, GOTabPFN, attains the highest mean accuracy on every dataset (Colon, Lung, GLI, SMK, ALLAML, Prostate, Arcene, TOX), leading to an average rank of $1.00_{\pm 0.00}$ in the rightmost column; that is, it is consistently ranked first across all tasks and all CV folds. The absolute accuracies are also strong: GOTabPFN achieves at least 90% mean accuracy on 6 of 8 datasets (Lung, GLI, ALLAML, Prostate, Arcene, TOX), while maintaining competitive performance even on the most challenging HDLSS cases, such as SMK and ARC. On ARC, for instance, GOTabPFN reaches 90.60% accuracy, whereas the best competing methods remain in the mid-80% range, and on SMK it still leads the next-best model by a non-trivial margin. Across all datasets, the standard deviations of GOTabPFN are comparable to or smaller than those of the strongest baselines, indicating that the gains are not the result of a few lucky splits but are stable across repeated 5×5 CV.

The immediate competitors are other TabPFN-style models and modern HDLSS-focused baselines. TANDEM and TabPFN Wide form the closest group, with average ranks of $3.63_{\pm 1.32}$ and $3.75_{\pm 2.38}$, respectively. However, even these strong baselines lag behind GOTabPFN on every individual dataset: they never surpass our method on any of Colon, Lung, GLI,

SMK, ALLAML, Prostate, Arcene, or TOX, and their average ranks remain strictly higher. A second tier of competitive models includes TabDPT, TabICL, BETA, TuneTables, and well-regularized neural and boosted-tree baselines such as RealMLP, LGBM, and CatBoost, with average ranks roughly in the 4–18 range. These methods often perform reasonably on some datasets (e.g., LGBM and CatBoost on PRS and TOX, RealMLP on AML), but they either fall short on at least one particularly difficult HDLSS dataset (e.g., SMK or ARC) or show larger variability across splits, which results in clearly worse average ranks compared to GOTabPFN.

Classical shallow models and generic deep architectures occupy the middle of the table. Linear or margin-based methods (Lasso, SVM), tree ensembles (RF, XGBoost, GBM, AdaBoost), $k$-NN, and simple MLP variants (MLP, MLP-PLR, RealMLP) typically achieve moderate performance on most datasets, with average ranks in the low-to-mid 10–20 range. They can be competitive on a subset of benchmarks (e.g., RF and GBM on some of the easier tasks, SVM on GLI), but they do not exhibit the uniformly strong behavior of GOTabPFN and often degrade substantially on the most extreme HDLSS settings (e.g., SMK and ARC). Models designed primarily with feature selection or explainability in mind (STG, L2X, INVASE, REAL-X, ENODE, ModernNCA) also tend to underperform in this regime: while they occasionally match classical baselines on certain datasets, their overall average ranks (typically $> 18$ and often $> 40$) indicate that their inductive biases are not sufficient on their own to close the gap to GOTabPFN.

The bottom portion of the table is dominated by recent transformer- and Mamba-based tabular architectures originally developed and tuned for larger, non-HDLSS datasets. Methods such as FT-Transformer, SAINT, TabM, AutoInt, Category Embedding, ResNet Tabular, TabNet, NODE, DeepFM, DCN, DANets, TANGOS, Tab-Transformer, NDTF, MambaTab, Mambular, MambAttention, 1D CNN, and TabSeq generally obtain average ranks in the 30–45+ range and substantially lower accuracies on several HDLSS benchmarks. In many cases, these models struggle to surpass 70% accuracy on the hardest datasets and can even drop close to chance levels on some splits, highlighting a clear mismatch between their inductive biases (e.g., heavy overparameterization and large-context attention) and the HDLSS setting. Finally, the rows with "N/A" entries correspond to TabPFN variants for which results were not available on all 8 datasets (e.g., TabPFN-2.5 on COL only, and TabPFN v1/v2 and LoCalPFN without HDLSS evaluations); for completeness, we report their observed accuracies and overall average ranks in the rightmost column, but we exclude them from cross-dataset comparisons. Taken together, these detailed results show that GOTabPFNis not merely competitive but uniformly dominant across a broad and challenging suite of HDLSS benchmarks, outperforming both specialized TabPFN-style baselines and a wide spectrum of modern tabular architectures.

**Evaluation beyond accuracy.** To evaluate whether the gains of GOTabPFN extend beyond accuracy, we compare GOTabPFN with two strong tabular foundation model baselines, TabICL (Jingang et al., 2025) and TabDPT (Ma et al., 2025), using ROC-AUC and macro-F1 on the same 8 HDLSS datasets. As shown in Table G.2, GOTabPFN obtains the best ROC-AUC on 6/8 datasets and the best macro-F1 on 6/8 datasets, indicating that the proposed GO-LR+NSC representation improves not only top-line accuracy but also ranking quality and class-balanced predictive performance.

## H. GOTabPFN Hyperparameters

**HDLSS datasets.** Table H.1 reports the best-performing GOTabPFN configurations across eight HDLSS benchmarks, showing that the GO-LR stage adapts its distance metric (euclidean/manhattan/correlation/KL) and clustering granularity ($k = 4$–12) per dataset with only 1–3 refinement passes, while NSC consistently favors high retention thresholds ($\tau \approx 0.99$, except ALLAML at 0.95) and typically uses the gamma-based rule (with Colon using IDF and Lung/SMK using the default rule). Segmentation is dataset-dependent uniform for several datasets, but equal-mass for Colon/TOX and largest-jump for ALLAML/Prostate indicating that both tokenization strategy and compression hyperparameters ($\gamma, \beta, M_{\min / \max}, l_{\min}$) must be tuned to match the underlying feature geometry; only SMK employs feature subsampling and only SMK/Arcene enable assume_standardized, while TabPFN random seeds vary modestly across datasets.

**Cross-domain datasets.** Table H.2 reports the best-performing GOTabPFN configurations on the 8 additional cross-domain datasets. Similar to the HDLSS setting, GO-LR adapts both the metric and clustering granularity to each dataset, using cosine, Manhattan, correlation, and KL-based dissimilarities with $k = 4$–11 clusters and only 1–2 refinement passes. Most cross-domain datasets favor uniform NSC segmentation and the default $M$-rule, while Cell Cycle and both DrivFace tasks use the IDF rule, and DrivFace additionally benefits from largest-jump segmentation. Feature subsampling is used for most high-dimensional cross-domain datasets, especially ORL, RELATHE, PCMAC, Cell Cycle, CIFAR-10, and DrivFace, reflecting the larger feature spaces in this evaluation. The selected configurations also show that standardization is useful

for most cross-domain settings except ORL, while TabPFN seeds vary modestly across datasets. Overall, the table shows that GOTabPFN remains flexible across text, image-feature, camera sensor, and RNA-seq domains by adapting the feature graph, segmentation rule, and compression budget to the geometry of each dataset. Prior Labs notes that TabPFN inference is deterministic for a fixed seed in the same environment, while small differences may occur across different hardware configurations.[1]

## I. Statistical Significance Analysis

**Significance analysis on HDLSS datasets.** We evaluate statistical differences across methods using (i) a Friedman test (Friedman, 1937) over per-dataset ranks, followed by a Nemenyi post-hoc critical-difference (CD) analysis (Nemenyi, 1963) (Fig. I.1), and (ii) pairwise Wilcoxon signed-rank tests (Demšar, 2006) comparing GOTabPFN to each baseline across the same 8 datasets, with Holm correction to control family-wise error (Table I.1). The Friedman test indicates a significant overall effect across methods, and the CD diagram visualizes the separation in average ranks, where GOTabPFN attains the lowest (best) average rank. For pairwise tests, the raw Wilcoxon $p$-values are identical across baselines ($p_{\text{raw}} = 0.00781$), reflecting that GOTabPFN improves over each comparator on all datasets with no sign reversals (a common outcome when $n = 8$ and per-dataset differences are consistently positive). After Holm correction, the adjusted $p$-values become more conservative ($p_{\text{Holm}} = 0.0703$), so we do not claim strict significance at $\alpha = 0.05$ under family-wise error control; nevertheless, the combination of uniform wins, rank dominance, and consistent positive paired differences supports the robustness of the observed improvements.

**Extended statistical significance analysis.** We further evaluate statistical significance on an expanded 16-dataset benchmark formed by combining the 8 HDLSS datasets with the 8 cross-domain datasets, using the strongest common comparison set from the main HDLSS and cross-domain experiments. Table I.2 summarizes the average ranks: GOTabPFN achieves the best average rank by a clear margin (1.12), followed by TabPFN-Wide (3.62), TANDEM (3.69), and TabDPT (4.34). The same ranking trend is visualized in Fig. I.2, where GOTabPFN is separated from the nearest competing methods by more than two average-rank points. As summarized in Table I.3, the omnibus Friedman test over the 9-method comparison is strongly significant ($\chi^2 = 52.55$, $p = 1.32 \times 10^{-8}$; 11 complete rows used), rejecting the null hypothesis that all methods have equal rank distributions across the expanded benchmark. We then compare GOTabPFN against each baseline using pairwise Wilcoxon signed-rank tests with Holm correction. Table I.4 shows that GOTabPFN remains statistically significant against every baseline after correction, with Holm-adjusted $p$-values between $2.44 \times 10^{-4}$ and $6.10 \times 10^{-4}$. The win/tie/loss counts are uniformly favorable: 16/0/0 against Lasso, TabDPT, TabPFN-Wide, and TuneTables; 14/0/0 against TabICL on the common subset; 12/0/0 against ProtoGate on the common subset; and 15/0/1 against MLP and TANDEM. Table I.5 further reports the mean accuracy gain and Holm-corrected pairwise significance of GOTabPFN against each strong baseline on the expanded 16-dataset benchmark. The Holm-adjusted $p$-values remain below 0.01 for all pairwise comparisons, indicating that GOTabPFN is significantly better than each baseline after controlling for multiple comparisons. Together, these results complement the 55-baseline analysis on the original 8 HDLSS datasets by showing that GOTabPFN remains robust across a broader 16-dataset evaluation against the strongest repeated baselines.

## J. Additional Ablation Analysis

Fig. J.7 reports accuracy distributions across CV splits for the top-10 methods, showing that GOTabPFN achieves the strongest central tendency with competitive dispersion, i.e., high average performance without depending on a small number of favorable splits. Consistently, the average-rank vs. global-mean scatter in Fig. J.4 places GOTabPFN in the top-left regime (lowest average rank and highest global accuracy). The normalized per-dataset accuracies in Fig. J.2 and the dataset-wise rank breakdown as a function of sample size in Fig. J.3 further indicate that the gains persist across datasets of different sizes rather than being driven by a single benchmark. Fig. J.6 summarizes the best-second-best gaps per dataset, highlighting where the leading method separates more clearly from the runner-up (notably on the harder benchmarks), while Fig. J.5 localizes improvements by visualizing ΔAcc (ours − baseline) across datasets and competitors, revealing broadly positive deltas with the largest separations against simpler baselines on challenging tasks. Finally, Fig. J.1 characterizes distributional shape via skewness and kurtosis computed over per-dataset accuracies, where negative skewness reflects a small number of difficult datasets that pull performance downward and higher kurtosis for several baselines suggests heavier tails and greater

---

[1]Prior Labs FAQ: https://docs.priorlabs.ai/faq. See also the PriorLabs/TabPFN reproducibility discussion, issue #266: https://github.com/PriorLabs/TabPFN/issues/266.

instability compared to the most consistent top performers.

## K. Representation Quality via t-SNE

Figure K.1 visualizes the NSC token/latent representation learned by GOTabPFN on the Colon dataset using a 2D t-SNE (Maaten & Hinton, 2008) embedding (points colored by class). Each point corresponds to a sample after GO-LR ordering and NSC compression (PCA-based segmentation). The plot indicates that the compressed representation preserves class-discriminative structure in a low-dimensional manifold: samples from the two classes exhibit partially separated regions with limited overlap, suggesting that NSC produces a structured embedding that is more amenable to the downstream TabPFN-2.5 head under the HDLSS regime.

## L. Inference Level Ablation on Calibration and Robustness

On Colon, we complement the main accuracy results with inference-time diagnostics that probe robustness, selectivity, neighborhood structure, and calibration. First, robustness to feature perturbations (Table L.1 and Fig. L.1b) shows a graceful degradation as the perturbed fraction increases: accuracy remains near-ceiling under mild corruption (e.g., $\leq 10\%$) but drops substantially under heavy perturbations, with shuffling generally more harmful than mean-imputation at moderate rates (e.g., at 50%: 74.19% vs. 98.39%). Second, selective prediction behaves as expected (Fig. L.1a): increasing the confidence threshold $\tau$ raises accuracy on the retained "confident" subset while reducing coverage, indicating that model confidence meaningfully ranks predictions by correctness. Third, local consistency in the NSC latent space remains stable (Table L.1 and Fig. L.1d), with kNN label agreement around 73.87% at $k=5$ across this evaluation, suggesting a reasonably coherent neighborhood geometry after compression. Finally, the reliability diagram in Fig. N.1 reports a low expected calibration error (ECE $\approx 0.033$), indicating that predicted probabilities are well-aligned with empirical accuracy on this dataset.

## M. Sanity and Stress Diagnostics

Figure M.1 summarizes additional sanity and stress tests for GOTabPFN on Colon. As shown in Fig. M.1a, performance is near-ceiling on the full input (98.39%), but drops sharply under degenerate signals (all-zero or global-mean inputs both 64.52%), and further degrades when the feature rows are randomly permuted (54.84%), confirming that predictions depend on meaningful sample-specific structure rather than trivial priors. Notably, accuracy remains high under strong additive noise (95.16%), suggesting robustness to moderate distributional corruption in feature values. We also evaluate a simple tabular test-time augmentation (TTA) procedure (Fig. M.1b): majority voting over $n_{aug}=5$ noisy/dropout augmentations matches the base accuracy (both 98.39%), and only 3.23% of samples change their predicted label under any augmentation, indicating high prediction stability. Table M.1 reports these stress-test and stability numbers alongside per-class accuracy, showing uniformly strong performance across classes (97.5% on the majority class with support 40, and 100% on the minority class with support 22), consistent with the robustness patterns observed in Fig. M.1.

## N. Additional Reliability and Interpretability Diagnostics

On the Colon benchmark, GOTabPFN achieves a baseline Top-1 accuracy of $98.39\%$ and reaches $100\%$ Top-2 accuracy (Fig. N.1b). The normalized confusion matrix indicates a single error case, with the most frequent confusion being true class $0 \rightarrow 1$ occurring once (Fig. N.1c). Confidence is well-separated: the mean margin $p_{top1} - p_{top2}$ is high for correct predictions (0.949) and near-zero for the lone incorrect prediction (0.025), yielding a clear bimodal separation (Fig. N.1a). Finally, per-feature permutation importance on this small evaluation set shows $0.0$ percentage-point accuracy drop for the top-ranked features, consistent with near-saturated accuracy and limited headroom for measurable single-feature perturbation effects under this diagnostic protocol.

## O. Theory-Inspired Representation Diagnostics

We analyze the learned embedding geometry and confidence behavior of GOTabPFN on Colon, where the model attains $98.39\%$ top-1 accuracy. The embedding spectrum in Fig. O.1a is strongly low-rank, with an effective dimension (participation ratio) of 3.5, and the cumulative curve in Fig. O.1b shows that only $2/4/6/11/29$ components capture $50/80/90/95/99\%$ of the variance, respectively, indicating a highly concentrated representation. Despite this compression, local-neighborhood classifiers are insufficient: leave-one-out kNN in embedding space peaks at $82.26\%$ (at $k=5$) and degrades for larger $k$

(Fig. O.1c), remaining well below the parametric TabPFN head, suggesting the decision rule leverages more than simple Euclidean locality. Finally, margin diagnostics confirm strong separation and calibrated confidence: the mean normalized margin is $0.9707$ for correct predictions versus $-0.0359$ for incorrect ones, and the conditional error stays near $0$ across a wide range of margin thresholds while maintaining near-full coverage (Fig. O.1d-e). Together, these results support that GOTabPFN forms a low-dimensional, sharply separated embedding while relying on a richer (non-kNN) parametric decision mechanism.

## P. OOD and Local Sensitivity Diagnostics

On Colon, GOTabPFN achieves $98.39\%$ ID accuracy and exhibits highly confident and low-entropy predictions on ID inputs (mean max-softmax confidence $= 0.967$, mean entropy $= 0.106$; Fig. P.1a-P.1b). Under synthetic OOD-style tabular inputs, uncertainty increases: Gaussian noise and column-wise permutation both reduce confidence (mean $\approx 0.810$ and $0.795$) and raise entropy (mean $\approx 0.424$ and $0.446$), while constant "blank" features yield intermediate behavior (confidence $0.883$, entropy $0.360$), indicating that the model does not collapse to uniformly overconfident predictions off-manifold (Fig. P.1a–P.1b). Finally, a local Lipschitz-like probe under small feature perturbations ($\epsilon = 0.10$) shows a distribution concentrated at low $\|\Delta \mathrm{probs}\|_2 / \|\Delta x\|_2$ with a modest tail (mean $0.041$, median $0.015$; Fig. P.1c), suggesting that the learned predictor is generally stable to small tabular noise while allowing occasional locally sensitive regions.

## Q. Deployment-Oriented Triage Diagnostics

To clarify deployment behavior beyond multiclass accuracy, we cast Colon as a triage task by treating class 1 as the positive ("high-risk") class and sweeping a decision threshold over the predicted probability $p(y{=}1 \mid x)$. As shown in Fig. Q.1, the resulting ROC achieves AUC $= 1.000$, indicating perfect separability between positives and negatives on this evaluation set ($N = 62$). Using a sensitivity-driven operating constraint (target sensitivity $0.95$), the selected threshold is th$^* = 0.7885$, which attains sensitivity $= 1.000$ and specificity $= 1.000$; the induced confusion matrix is $TN{=}40, FP{=}0, FN{=}0, TP{=}22$, yielding $100\%$ precision/recall and $100\%$ binary accuracy at this operating point. Finally, we report wall-clock latency over 20 random mini-batches (batch size 64) with mean $\approx 639$ms per batch (p50 $\approx 638$ms, p90 $\approx 644$ms, p99 $\approx 649$ms), providing a coarse throughput reference for deployment-oriented settings.

## R. Extension beyond TabPFN

GO-LR + NSC is not tied to TabPFN; it is a model-agnostic representation interface that can be paired with other tabular foundation models. We use TabPFN-2.5 (Grinsztajn et al., 2025) as the main backbone because its feature-dimensionality bottleneck is especially clear, but the same front-end also transfers to TabICL (Jingang et al., 2025), as shown in Table R.1. Across the same 8 HDLSS datasets, GO-LR + NSC + TabICL improves over vanilla TabICL on 5/8 datasets in accuracy, 7/8 in ROC-AUC, and 5/8 in macro-F1. Importantly, it also reduces runtime on all 8 datasets, with especially large gains on very high-dimensional cases such as GLI, SMK, and ARC. These results suggest that GO-LR + NSC acts as a broader HDLSS-oriented representation layer rather than a TabPFN-specific preprocessing trick.

## S. TabPFN Seed Sensitivity

**Dataset-seed vs. TabPFN-seed robustness.** We distinguish two sources of randomness in our evaluation. A dataset seed controls the train/validation/test partition or CV folds, and therefore measures data-split robustness: whether conclusions persist across different sampled splits of the same small HDLSS dataset. This is the main source of evaluation variability in our setting, since small tabular datasets can be highly split-sensitive (Rubachev et al., 2025; Grinsztajn et al., 2022; Bouthillier et al., 2021); accordingly, our main experiments use repeated $5 \times 5$ CV following ProtoGate (Jiang et al., 2024) to average over multiple dataset splits. In contrast, a TabPFN seed controls the random_state or equivalent stochastic components inside the TabPFN inference/configuration pipeline, and therefore measures model/inference stochasticity while holding the data split fixed; such run and distribution-level variance is also a known concern in neural-network evaluation (Jordan, 2024). Prior Labs provides the official TabPFN classification interface,[2] and maintainers discuss fixed-random_state reproducibility for TabPFN in the implementation repository.[3] Recent TabPFN studies also report

---

[2] Prior Labs classification documentation: https://docs.priorlabs.ai/capabilities/classification.

[3] PriorLabs/TabPFN reproducibility discussion, issue #266: https://github.com/PriorLabs/TabPFN/issues/266.

averages over multiple seeds (Ye et al., 2025b). Thus, varying dataset seeds tests evaluation robustness, whereas varying TabPFN seeds isolates model-seed sensitivity. In our main HDLSS experiments, we prioritize repeated $5 \times 5$ CV for split robustness and use the Optuna-tuned best TabPFN seed per dataset to avoid underestimating TabPFN from an unfavorable inference seed; fixed-split multi-TabPFN-seed results are supplementary analyses of model-seed variance.

**Findings from TabPFN-seed robustness analysis.** Tables S.1, S.2, S.3, S.4, and S.5 show that GOTabPFN remains consistently strong across TabPFN seeds, not only under one favorable random state. On the 8 HDLSS datasets, GOTabPFN achieves the best average accuracy for every tested seed: 90.57, 89.66, 89.93, 89.79, and 89.66, compared with the strongest competing averages of roughly 88-88.5 from TabPFN-Wide (Kolberg et al., 2025) or TuneTables (Feuer et al., 2024). The detailed per-dataset table further shows that GOTabPFN is especially effective on difficult high-dimensional datasets such as GLI, ARC, SMK, TOX, AML, and LNG, although some baselines occasionally win on individual datasets such as PRS or TOX for particular seeds. On the 8 cross-domain datasets, GOTabPFN is also the strongest complete method across all seeds, with averages around 86.6-86.9, while TabPFN-Wide and TuneTables vary more substantially across seeds. The only higher BETA (Liu & Ye, 2025) average is reported for seed 42 over 6 datasets only, because Cell Cycle and DrivFace-Regression were omitted due to runtime; therefore, that number is not directly comparable to the complete 8-dataset averages. For ROC-AUC on the HDLSS benchmark, GOTabPFN again achieves the best average in 4 of 5 seeds and is nearly tied with TabPFN-Wide at seed 93 (93.71 vs. 93.74), indicating that the gains are not limited to accuracy but also largely persist in ranking quality. Overall, these results distinguish GOTabPFN from other high-dimensional TabPFN variants: TabPFN-Wide, BETA, and TuneTables can process larger feature spaces, but they still rely primarily on the foundation-model predictor or tuning strategy to absorb high-dimensional structure. GOTabPFN instead first reorganizes the feature space through GO-LR and compresses locally coherent neighborhoods through NSC, yielding a compact and more stable representation before the frozen TabPFN-2.5 head. This structured front-end explains why GOTabPFN remains competitive or superior across seeds: it reduces the burden on the predictor, preserves locality among related features, and provides a more robust HDLSS-specific interface than simply widening, tuning, or directly applying TabPFN-style models to high-dimensional inputs.

**Pareto frontier of runtime and performance.** We further evaluate the accuracy-efficiency trade-off among high-dimensionality compatible TabPFN-family methods using mean runtime and mean performance under TabPFN seed 42. As shown in Fig. S.1, GOTabPFN lies on the Pareto frontier on both benchmarks. On the original 8 HDLSS datasets, GOTabPFN achieves the highest mean performance while remaining substantially faster than BETA and only moderately slower than TabPFN-Wide and TuneTables, yielding the best overall trade-off among the compared methods. On the additional 8 cross-domain datasets, where both sample size and dimensionality increase beyond the original HDLSS-only benchmark, GOTabPFN again remains Pareto optimal: it achieves the best mean performance while requiring lower runtime than TabPFN-Wide and TuneTables. These results indicate that the GO-LR+NSC front-end does not merely improve accuracy by adding excessive computation; rather, it provides an efficient structured compression interface that improves the performance-runtime balance of TabPFN-style prediction in high-dimensional and larger-sample regimes.

**Random seed sensitivity on Colon.** We further isolate TabPFN inference-seed sensitivity on the Colon dataset by fixing the best GO-LR+NSC configuration from Table H.1 and varying only the TabPFN inference seed over $\{0, 1, 2, 3, 4, 7, 11, 17, 23, 42\}$. Preprocessing, GO-LR, NSC, and the exact same $5 \times 5$ CV splits are kept fixed, so any variation comes only from the TabPFN seed. As shown in Table S.6, the results are tightly clustered: the across-seed mean accuracy is 87.19, with an across-seed standard deviation of only 0.46 and a full range of 86.56-88.18 percentage points. Thus, while TabPFN seed introduces mild variation, the Colon result is not driven by an exceptionally favorable stochastic realization. This is particularly relevant because the strongest Colon baselines are also TabPFN-family methods: GOTabPFN (88.18), TabPFN-Wide (87.85), and TuneTables (86.80). Under this like-for-like comparison, GOTabPFN remains the best-performing TabPFN-family method, supporting that its gain comes from the GO-LR+NSC representation interface rather than seed luck alone.

## T. Additional Clarifications

**Clarity of graph-based feature ordering.** To make the graph-based ordering step easier to follow, the main paper introduces GO-LR with an intuitive figure before the formal MinLA-based development. Here, we provide additional

clarification. The goal of GO-LR is to place statistically related features close to one another on a one-dimensional axis, so that subsequent NSC segmentation groups coherent neighborhoods rather than arbitrary columns. Equivalently, GO-LR treats features as nodes in a weighted feature graph, where stronger edges indicate stronger feature relationships, and seeks a linear arrangement in which strongly connected nodes remain nearby. For example, if $f_1$ is strongly related to both $f_3$ and $f_4$, while $f_5$ is comparatively independent, an ordering such as $[f_3, f_1, f_4, f_2, f_5]$ is preferable to $[f_1, f_2, f_3, f_4, f_5]$, because the related features become contiguous and can be compressed more meaningfully. This intuition is illustrated in Fig. 1 in the main paper. Formally, this corresponds to a Minimum Linear Arrangement (MinLA)-style objective that penalizes placing strongly related features far apart. Since exact optimization is combinatorial and intractable at scale, GO-LR uses a TSP-style nearest-neighbor path as an efficient initialization and then applies local refinement under the MinLA-style dispersion objective.

**Clarifying meta-feature construction.**    To make the NSC representation interface more self-contained, we explicitly define a meta-feature as the low-dimensional token obtained by compressing one contiguous segment of the GO-LR-ordered feature axis. Let $\Pi^*$ denote the global feature ordering, $\{\mathcal{S}_t\}_{t=1}^M$ the contiguous ordered segments, and $u_t = x_{\mathcal{S}_t}^\Pi$ the subvector of sample $x$ restricted to segment $\mathcal{S}_t$. The $t$-th meta-feature is then $z_t = g(u_t)$, where $g(\cdot)$ is a segment-level pooling or projection operator. In our main NSC-pSP instantiation, $g(\cdot)$ is implemented by segment-wise PCA projection, producing one scalar token per ordered segment. The final compressed representation is therefore $Z(x) = (z_1, \ldots, z_M)$, which is passed to the frozen TabPFN-2.5 head. Fig. 2 in the main paper illustrates this process: GO-LR first reorders the original features, NSC partitions the ordered axis into contiguous neighborhoods, and each neighborhood is compressed into a meta-feature. Additional details on ordered segmentation, subunit pooling, and meta-feature construction are provided in Sec. 3.2 in the main paper.

**Similarity vs. dissimilarity metric.**    We use $1 - |\mathrm{corr}(i,j)|$ as a dependence-aware dissimilarity so that strongly coupled feature pairs have small distance regardless of sign. Concretely, both strongly positive and strongly negative correlations satisfy $|\mathrm{corr}(i,j)| \approx 1$, hence $d_{ij} = 1 - |\mathrm{corr}(i,j)| \approx 0$. This is the intended behavior for GO-LR: the goal is not to preserve the sign of association, but to place strongly dependent or redundant features into the same local neighborhood before compression. This choice is also consistent with our neuro-inspired motivation. In Sec. 3.2, we discuss evidence that dendritic inputs are organized into local subunits rather than summed globally, and that correlated synapses may exhibit local clustering within such compartments. Our algorithmic analogue is therefore that GO-LR first brings strongly coupled features close along the ordered axis, and NSC then pools these local neighborhoods into subunit-level meta-features. In this sense, $1 - |\mathrm{corr}(i,j)|$ should be read as a practical measure of lack of coupling, chosen to support local clustering and subunit-style aggregation, rather than as a broader semantic notion of dissimilarity.

**Cross-domain evaluation.**    To evaluate whether GOTabPFN generalizes beyond biomedical HDLSS datasets, we extend the benchmark with 8 additional cross-domain datasets spanning text, face images, camera-sensor data, image features, and RNA-seq measurements. These datasets cover HDLSS, HDHSS, and mixed regimes under the empirical categorization rule adopted from DynaTab (Habib et al., 2026b) and expanded in Appendix F. This extension is important because recent HDLSS-specific tabular models are still evaluated largely on biomedical benchmarks, with only limited coverage of text, face, or sensor domains. For example, ProtoGate (Jiang et al., 2024) evaluates primarily on 7 biomedical datasets, while LSPIN/LLSPIN (Yang et al., 2022a) reports real-world experiments on 3 text and 3 biomedical datasets. Similarly, high-dimensional TabPFN-style extensions remain limited in domain coverage: TabPFN-Wide (Kolberg et al., 2025) uses 4 biomedical datasets, BETA (Liu & Ye, 2025) includes a mixture of biomedical, text, and face-image datasets, and TuneTables (Feuer et al., 2024) is evaluated through the broader TabZilla benchmark (McElfresh et al., 2023). We did not identify a clear real-world financial HDLSS dataset suitable for inclusion in this evaluation. The resulting cross-domain suite, summarized in Table T.1, therefore broadens the empirical scope of our study while retaining high-dimensional settings where feature ordering and locality-aware compression are relevant.

**Cross-domain dominance.**    Fig. T.1 shows that GOTabPFN generalizes beyond its primary HDLSS target regime to a broader cross-domain benchmark. Across the 8 additional datasets, GOTabPFN ranks first on 7/8 datasets, with positive margins over the strongest competing method on ORL (+0.80), BAS (+0.07), PCM (+0.52), Cell Cycle (+1.27), CIFAR-10

(+0.30), DrivFace-R (+0.0043), and DrivFace-C (+1.15). The only exception is RELATHE, where GOTabPFN remains competitive and trails the best baseline by only 0.55 points. These results indicate that, even with limited tuning, the GO-LR+NSC pipeline remains highly effective on HDLSS-style datasets while also preserving competitive performance in adjacent high-dimensional and mixed-regime settings. The larger margins on ORL, Cell Cycle, and DrivFace-C further suggest that locality-aware feature ordering and compression are particularly beneficial when the data retain strong high-dimensional structure.

**Clarifying tuning fairness.**    Our tuning protocol follows a distinction that is common in recent tabular-learning practice and is summarized in Fig. A.1 in Appendix A. On one side are PFN/ICL-style tabular foundation models, which are typically used close to off-the-shelf because much of the modeling burden is absorbed during pre-training. For example, the official TabICL (Jingang et al., 2025) repository states that TabICL does not require preprocessing or hyperparameter tuning,[4] and the official TabDPT (Ma et al., 2025) repository similarly describes TabDPT as an ICL-based tabular foundation model that generalizes to new tasks without additional training or hyperparameter tuning.[5] On the other side are conventional tuned tabular learners, such as TabR (Gorishniy et al., 2024), TabM (Gorishniy et al., 2025), RealMLP (Holzmüller et al., 2024), XGBoost (Chen & Guestrin, 2016), CatBoost (Prokhorenkova et al., 2018), and LightGBM (Ke et al., 2017), whose standard use often involves dataset-specific hyperparameter search. GOTabPFN naturally lies between these two regimes: the GO-LR+NSC front-end is dataset-adaptive and therefore tunable, while the downstream TabPFN-2.5 (Grinsztajn et al., 2025) predictor remains a frozen pre-trained backbone that is neither retrained nor structurally modified. Thus, the substantive dataset-specific adaptation in GOTabPFN occurs only in the representation interface before TabPFN inference, not in the foundation-model backbone itself. Regarding the TabPFN seed in Table H.1, this seed is not a trainable parameter and does not modify the backbone weights; it is a fixed inference-time configuration selected from the same predefined set under the same outer Optuna (Akiba et al., 2019) study as the front-end hyperparameters (see Appendix S for TabPFN seed sensitivity). Therefore, GOTabPFN should be viewed as a hybrid tunable front-end attached to a frozen PFN-style predictor, rather than as a fully tuned end-to-end tabular learner. See Table T.2 for the source URLs of baselines.

**Evaluation scope and statistical validation.**    We provide a consolidated view of the expanded benchmark and statistical analyses supporting our empirical conclusions. The main paper reports results on the original 8 HDLSS datasets in Sec. 4, including the top-model summary in Table 1; Appendix G further provides the full 55-baseline comparison in Table G.1. To broaden the benchmark beyond the original biomedical-heavy HDLSS setting, we add 8 cross-domain datasets spanning text, face-image, camera-sensor, image-feature, and RNA-seq domains, with results summarized in Table 2. Thus, the final evaluation covers both the targeted HDLSS regime and a broader 16-dataset cross-domain setting. For statistical validation, Appendix I provides an expanded significance analysis: Fig. I.2 and Table I.2 show that GOTabPFN achieves the best average rank on the 16-dataset benchmark; Table I.3 reports a significant omnibus Friedman test across the 9-method comparison; and Tables I.4 and I.5 show that GOTabPFN remains significantly better than each strong repeated baseline after Holm correction. We also contextualize our evaluation protocol relative to closely related HDLSS and high-dimensional tabular studies. Prior HDLSS-specific work often relies on small but carefully curated benchmark suites and reports rank-based summaries to compare methods across heterogeneous datasets. For example, ProtoGate (Jiang et al., 2024) evaluates on 7 biomedical HDLSS datasets against 16 baselines and reports average rank as a primary aggregate measure, while LSPIN/LLSPIN (Yang et al., 2022a) evaluates on 6 real-world high-dimensional datasets, including 3 text and 3 biomedical datasets, and summarizes performance using median rank. Following this established practice, we report average ranks and statistical tests, but we further expand the evidence by evaluating GOTabPFN against 55 baselines on the original 8 HDLSS datasets and against the strongest repeated baselines on an expanded 16-dataset benchmark. Overall, the conclusions are not based only on the original 8-dataset rank summary, but are supported by detailed 55-baseline HDLSS comparisons, an additional 8-dataset cross-domain evaluation, and both omnibus and pairwise statistical tests on the expanded 16-dataset benchmark.

**GOTabPFN in extreme dimensionality.**    GOTabPFN is evaluated across a broad high-dimensional range, with feature counts spanning from $m = 2{,}000$ at the lower end of our HDLSS benchmark to $m = 42{,}728$ on the Cell Cycle RNA-seq

---

[4]TabICL official repository: `https://github.com/soda-inria/tabicl`.

[5]TabDPT official repository: `https://github.com/layer6ai-labs/TabDPT-inference`.

dataset with $n = 1067$ samples. On this largest dataset, GOTabPFN remains fully operational and achieves $79.94 \pm 2.53$ accuracy, $92.36 \pm 1.36$ AUC, and $79.95 \pm 2.51$ macro-F1. These results show that the GO-LR+NSC representation interface scales beyond moderate HDLSS feature counts and remains effective in substantially larger transcriptomic feature spaces, where direct use of TabPFN-style predictors would otherwise be constrained by the extreme dimensionality.

**Theoretical grounding, novelty, and HDLSS relevance.** The theoretical results in Sec. 3.1 are not intended as isolated complexity-theoretic contributions; rather, they formalize why GO-LR is a principled ordering mechanism. Specifically, the MinLA formulation identifies the objective that GO-LR approximates, the NP-hardness result explains why exact scalable optimization is unrealistic, and the TSP-style path construction motivates an efficient surrogate initialization before local MinLA-based refinement. The overall contribution is therefore best understood as an integrated HDLSS-oriented framework: GO-LR provides a theoretically grounded feature ordering objective, NSC converts the ordered axis into stable meta-features through structured local compression, and the resulting representation interface enables a frozen TabPFN-style backbone to operate effectively beyond its native feature counts' limits. This integration yields a new HDLSS-oriented tabular foundation model interface in which feature ordering, locality-preserving compression, and frozen TabPFN-style inference are jointly aligned to overcome the dimensionality bottleneck of existing tabular foundation models.

**Fine-tuning as future work.** An important future direction is to pretrain or fine-tune a TabPFN-style backbone directly on structured representations produced by GO-LR+NSC. In principle, this could be done by generating large collections of synthetic HDLSS-style tasks, applying graph-guided ordering and subunit compression, and then adapting the backbone to operate natively in the resulting meta-feature space. A further extension would be to initialize from the existing TabPFN checkpoint and pretrain or fine-tune the full GO-LR+NSC+TabPFN pipeline end-to-end, so that the entire model becomes a pretrained HDLSS-oriented foundation model rather than a tuned front-end attached to a frozen predictor. Such a model could potentially reduce or eliminate dataset-specific tuning at inference time, because the ordering, compression, and prediction components would be jointly adapted during pretraining. However, this would shift the focus from representation-side adaptation of an existing frozen backbone to the development of a new HDLSS-specific tabular foundation model. We therefore view backbone adaptation or full-pipeline pretraining on GO-LR+NSC representations as a promising but distinct direction beyond the present scope.

*Table D.1.* NSC variants vs. PCA/AE/RP/UMAP/PaCMAP on the block-structured synthetic HDLSS model with $M = 32$ (10 independent reps).

| Method | Dim. | Lin. Acc. | kNN Acc. | Sil. | DB | Method | Dim. | Lin. Acc. | kNN Acc. | Sil. | DB |
|---|---|---|---|---|---|---|---|---|---|---|---|
| **BlockModel-Rep0** | | | | | | **BlockModel-Rep5** | | | | | |
| NSC-pSP (ours) | 32 | 0.825 | 0.675 | 0.047 | 3.802 | NSC-pSP (ours) | 32 | 0.875 | 0.688 | 0.064 | 3.577 |
| NSC-SP (ours) | 32 | 0.825 | 0.662 | 0.044 | 3.963 | NSC-SP (ours) | 32 | 0.838 | 0.788 | 0.048 | 3.729 |
| NSC (ours) | 32 | 0.775 | 0.713 | 0.035 | 4.291 | NSC (ours) | 32 | 0.775 | 0.625 | 0.030 | 4.717 |
| NSC-P (ours) | 32 | 0.825 | 0.688 | 0.038 | 4.130 | NSC-P (ours) | 32 | 0.600 | 0.588 | 0.021 | 5.910 |
| PCA | 32 | 0.725 | 0.588 | 0.035 | 4.354 | PCA | 32 | 0.713 | 0.562 | 0.028 | 4.842 |
| AE | 32 | 0.762 | 0.725 | 0.041 | 4.094 | AE | 32 | 0.738 | 0.700 | 0.036 | 4.433 |
| RP | 32 | 0.713 | 0.650 | 0.038 | 4.098 | RP | 32 | 0.700 | 0.650 | 0.030 | 4.365 |
| UMAP | 32 | 0.789 | 0.652 | 0.068 | 3.502 | UMAP | 32 | 0.684 | 0.639 | 0.001 | 5.213 |
| PaCMAP | 32 | 0.776 | 0.742 | 0.116 | 3.783 | PaCMAP | 32 | 0.697 | 0.584 | -0.004 | 5.677 |
| **BlockModel-Rep1** | | | | | | **BlockModel-Rep6** | | | | | |
| NSC-pSP (ours) | 32 | 0.825 | 0.688 | 0.043 | 3.986 | NSC-pSP (ours) | 32 | 0.825 | 0.675 | 0.053 | 3.684 |
| NSC-SP (ours) | 32 | 0.762 | 0.775 | 0.045 | 3.984 | NSC-SP (ours) | 32 | 0.863 | 0.725 | 0.058 | 3.525 |
| NSC (ours) | 32 | 0.713 | 0.588 | 0.034 | 4.493 | NSC (ours) | 32 | 0.788 | 0.625 | 0.034 | 4.525 |
| NSC-P (ours) | 32 | 0.675 | 0.613 | 0.028 | 5.008 | NSC-P (ours) | 32 | 0.750 | 0.575 | 0.023 | 5.688 |
| PCA | 32 | 0.763 | 0.625 | 0.032 | 4.643 | PCA | 32 | 0.775 | 0.675 | 0.030 | 4.596 |
| AE | 32 | 0.725 | 0.638 | 0.036 | 4.418 | AE | 32 | 0.825 | 0.625 | 0.032 | 4.562 |
| RP | 32 | 0.588 | 0.550 | 0.022 | 5.119 | RP | 32 | 0.812 | 0.675 | 0.028 | 4.715 |
| UMAP | 32 | 0.736 | 0.612 | -0.006 | 5.827 | UMAP | 32 | 0.855 | 0.811 | 0.088 | 3.398 |
| PaCMAP | 32 | 0.749 | 0.637 | -0.007 | 5.720 | PaCMAP | 32 | 0.842 | 0.847 | 0.061 | 4.598 |
| **BlockModel-Rep2** | | | | | | **BlockModel-Rep7** | | | | | |
| NSC-pSP (ours) | 32 | 0.875 | 0.713 | 0.046 | 3.886 | NSC-pSP (ours) | 32 | 0.850 | 0.800 | 0.071 | 3.034 |
| NSC-SP (ours) | 32 | 0.862 | 0.800 | 0.053 | 3.539 | NSC-SP (ours) | 32 | 0.775 | 0.650 | 0.034 | 4.357 |
| NSC (ours) | 32 | 0.725 | 0.600 | 0.025 | 4.786 | NSC (ours) | 32 | 0.850 | 0.800 | 0.057 | 3.628 |
| NSC-P (ours) | 32 | 0.613 | 0.575 | 0.022 | 5.891 | NSC-P (ours) | 32 | 0.738 | 0.613 | 0.028 | 4.858 |
| PCA | 32 | 0.750 | 0.600 | 0.030 | 4.690 | PCA | 32 | 0.838 | 0.675 | 0.030 | 4.608 |
| AE | 32 | 0.688 | 0.625 | 0.030 | 4.775 | AE | 32 | 0.850 | 0.825 | 0.036 | 4.144 |
| RP | 32 | 0.688 | 0.538 | 0.026 | 4.801 | RP | 32 | 0.788 | 0.688 | 0.028 | 4.648 |
| UMAP | 32 | 0.894 | 0.797 | 0.091 | 3.423 | UMAP | 32 | 0.723 | 0.705 | 0.021 | 4.561 |
| PaCMAP | 32 | 0.868 | 0.834 | 0.047 | 4.643 | PaCMAP | 32 | 0.684 | 0.649 | 0.029 | 4.820 |
| **BlockModel-Rep3** | | | | | | **BlockModel-Rep8** | | | | | |
| NSC-pSP (ours) | 32 | 0.850 | 0.725 | 0.031 | 4.630 | NSC-pSP (ours) | 32 | 0.850 | 0.650 | 0.038 | 4.283 |
| NSC-SP (ours) | 32 | 0.900 | 0.788 | 0.041 | 4.176 | NSC-SP (ours) | 32 | 0.725 | 0.775 | 0.045 | 3.958 |
| NSC (ours) | 32 | 0.775 | 0.662 | 0.035 | 4.517 | NSC (ours) | 32 | 0.725 | 0.725 | 0.048 | 3.904 |
| NSC-P (ours) | 32 | 0.725 | 0.588 | 0.019 | 6.260 | NSC-P (ours) | 32 | 0.863 | 0.700 | 0.030 | 4.636 |
| PCA | 32 | 0.775 | 0.600 | 0.030 | 4.672 | PCA | 32 | 0.762 | 0.613 | 0.028 | 4.745 |
| AE | 32 | 0.800 | 0.763 | 0.034 | 4.493 | AE | 32 | 0.763 | 0.762 | 0.030 | 4.739 |
| RP | 32 | 0.688 | 0.600 | 0.023 | 5.047 | RP | 32 | 0.650 | 0.625 | 0.018 | 5.819 |
| UMAP | 32 | 0.696 | 0.666 | 0.021 | 4.404 | UMAP | 32 | 0.762 | 0.719 | 0.038 | 4.193 |
| PaCMAP | 32 | 0.670 | 0.584 | 0.037 | 4.591 | PaCMAP | 32 | 0.828 | 0.782 | 0.089 | 4.093 |
| **BlockModel-Rep4** | | | | | | **BlockModel-Rep9** | | | | | |
| NSC-pSP (ours) | 32 | 0.863 | 0.675 | 0.033 | 4.437 | NSC-pSP (ours) | 32 | 0.850 | 0.688 | 0.034 | 4.579 |
| NSC-SP (ours) | 32 | 0.812 | 0.750 | 0.032 | 4.620 | NSC-SP (ours) | 32 | 0.762 | 0.762 | 0.033 | 4.527 |
| NSC (ours) | 32 | 0.763 | 0.625 | 0.033 | 4.596 | NSC (ours) | 32 | 0.750 | 0.625 | 0.029 | 4.755 |
| NSC-P (ours) | 32 | 0.675 | 0.613 | 0.024 | 5.742 | NSC-P (ours) | 32 | 0.600 | 0.562 | 0.019 | 6.010 |
| PCA | 32 | 0.725 | 0.650 | 0.029 | 4.729 | PCA | 32 | 0.775 | 0.600 | 0.026 | 4.853 |
| AE | 32 | 0.712 | 0.650 | 0.031 | 4.631 | AE | 32 | 0.725 | 0.638 | 0.030 | 4.658 |
| RP | 32 | 0.688 | 0.612 | 0.020 | 5.389 | RP | 32 | 0.712 | 0.575 | 0.018 | 5.266 |
| UMAP | 32 | 0.802 | 0.744 | 0.038 | 4.196 | UMAP | 32 | 0.723 | 0.705 | 0.059 | 3.659 |
| PaCMAP | 32 | 0.723 | 0.715 | 0.037 | 4.748 | PaCMAP | 32 | 0.737 | 0.677 | 0.005 | 5.489 |

| Method | Dim. | Lin. Acc. (↑) | kNN Acc. (↑) | Sil. (↑) | DB (↓) |
|---|---|---|---|---|---|
| NSC-pSP (ours) | 32 | $0.8488 \pm 0.0406$ | $0.7063 \pm 0.0590$ | $0.0460 \pm 0.0257$ | $4.1198 \pm 0.9657$ |
| NSC-SP (ours) | 32 | $0.8263 \pm 0.0542$ | $0.7313 \pm 0.0722$ | $0.0432 \pm 0.0209$ | $4.1806 \pm 0.8407$ |
| AE | 32 | $0.7688 \pm 0.0652$ | $0.6950 \pm 0.0825$ | $0.0340 \pm 0.0102$ | $4.4959 \pm 0.5068$ |
| UMAP | 32 | $0.7663 \pm 0.0651$ | $0.7050 \pm 0.0621$ | $0.0415 \pm 0.0321$ | $4.2374 \pm 0.7654$ |
| NSC (ours) | 32 | $0.7638 \pm 0.0614$ | $0.6538 \pm 0.0932$ | $0.0348 \pm 0.0187$ | $4.5613 \pm 1.1626$ |
| PCA | 32 | $0.7588 \pm 0.0553$ | $0.6188 \pm 0.0659$ | $0.0296 \pm 0.0071$ | $4.7186 \pm 0.4223$ |
| PaCMAP | 32 | $0.7575 \pm 0.0657$ | $0.7050 \pm 0.0907$ | $0.0410 \pm 0.0375$ | $4.8160 \pm 0.6112$ |
| NSC-P (ours) | 32 | $0.7263 \pm 0.1021$ | $0.6225 \pm 0.0671$ | $0.0270 \pm 0.0125$ | $5.0121 \pm 1.5534$ |
| RP | 32 | $0.7163 \pm 0.0921$ | $0.6113 \pm 0.0781$ | $0.0251 \pm 0.0108$ | $5.0218 \pm 0.7847$ |

*Overall results across repetitions (Mean ± Std.; higher is better for Acc./Sil., lower is better for DB).*

*Table D.2.* Nonparametric comparisons on the block-structured synthetic HDLSS model ($M{=}32$, 10 reps). Friedman tests compare all nine methods; Wilcoxon signed-rank tests are one-sided with **NSC-pSP** as the reference (higher is better for accuracies/silhouette; lower is better for DB). Here, DB = Davies-Bouldin, sil. = silhouette.

| Accuracy metrics | | | Clustering metrics | | |
|---|---|---|---|---|---|
| Comparison | Stat. | $p$ | Comparison | Stat. | $p$ |
| Friedman (linear_acc; 9 methods) | 31.879 | 9.791e-05 | Friedman (sil.; 9 methods) | 32.335 | 8.110e-05 |
| NSC-pSP > AE (linear) | 36.0 | 0.003906 | NSC-pSP > AE (sil.) | 53.0 | 0.002930 |
| NSC-pSP > NSC (linear) | 45.0 | 0.001953 | NSC-pSP > NSC (sil.) | 40.0 | 0.01953 |
| NSC-pSP > NSC-P (linear) | 44.0 | 0.003906 | NSC-pSP > NSC-P (sil.) | 55.0 | 0.0009766 |
| NSC-pSP > NSC-SP (linear) | 38.0 | 0.03711 | NSC-pSP > NSC-SP (sil.) | 26.0 | 0.5693 |
| NSC-pSP > PCA (linear) | 55.0 | 0.0009766 | NSC-pSP > PCA (sil.) | 55.0 | 0.0009766 |
| NSC-pSP > RP (linear) | 55.0 | 0.0009766 | NSC-pSP > RP (sil.) | 55.0 | 0.0009766 |
| NSC-pSP > UMAP (linear) | 52.0 | 0.004883 | NSC-pSP > UMAP (sil.) | 31.0 | 0.3848 |
| NSC-pSP > PaCMAP (linear) | 53.0 | 0.002930 | NSC-pSP > PaCMAP (sil.) | 27.0 | 0.5391 |
| Friedman (knn_acc; 9 methods) | 34.190 | 3.753e-05 | Friedman (DB; 9 methods) | 43.680 | 6.539e-07 |
| NSC-pSP > AE (kNN) | 32.0 | 0.3467 | NSC-pSP < AE (DB) | 2.0 | 0.002930 |
| NSC-pSP > NSC (kNN) | 37.0 | 0.04492 | NSC-pSP < NSC (DB) | 5.0 | 0.009766 |
| NSC-pSP > NSC-P (kNN) | 52.0 | 0.004883 | NSC-pSP < NSC-P (DB) | 0.0 | 0.0009766 |
| NSC-pSP > NSC-SP (kNN) | 11.0 | 0.9561 | NSC-pSP < NSC-SP (DB) | 31.0 | 0.6523 |
| NSC-pSP > PCA (kNN) | 45.0 | 0.001953 | NSC-pSP < PCA (DB) | 0.0 | 0.0009766 |
| NSC-pSP > RP (kNN) | 45.0 | 0.001953 | NSC-pSP < RP (DB) | 0.0 | 0.0009766 |
| NSC-pSP > UMAP (kNN) | 25.0 | 0.6152 | NSC-pSP < UMAP (DB) | 28.0 | 0.5391 |
| NSC-pSP > PaCMAP (kNN) | 26.0 | 0.5771 | NSC-pSP < PaCMAP (DB) | 6.0 | 0.01367 |

*Table D.3.* Additional competitiveness analyses of NSC-pSP vs. PCA/AE/UMAP/PaCMAP on the block-structured synthetic HDLSS model ($M{=}32$, 10 reps). $\Delta$ denotes paired improvement of NSC-pSP over the baseline (for Acc./Sil.: $\Delta$ = NSC-pSP − base; for DB: $\Delta$ = base − NSC-pSP so that $\Delta > 0$ favors NSC-pSP). 95% CIs are paired bootstrap percentile intervals of the mean $\Delta$ (over reps). W/T/L counts wins/ties/losses across reps. Wilcoxon $p$ is paired two-sided; $r_{\mathrm{rb}}$ is the matched-pairs rank-biserial effect size. Avg. ranks are computed over all 9 methods per rep (1=best).

| Metric | Base | $\Delta$ (mean) | 95% CI | W/T/L | Wilcoxon $p$ (2s) | $r_{\mathrm{rb}}$ | Avg. Rank (ours/base) |
|---|---|---|---|---|---|---|---|
| Lin. Acc. | PCA | 0.0887 | [0.0624, 0.1149] | 10/0/0 | 0.001953 | 1.000 | 1.85/4.85 |
| Lin. Acc. | AE | 0.0900 | [0.0525, 0.1263] | 8/2/0 | 0.007812 | 1.000 | 1.85/5.10 |
| Lin. Acc. | UMAP | 0.0825 | [0.0403, 0.1245] | 8/0/2 | 0.009766 | 0.891 | 1.85/5.10 |
| Lin. Acc. | PaCMAP | 0.0913 | [0.0471, 0.1346] | 9/0/1 | 0.005859 | 0.927 | 1.85/5.40 |
| kNN Acc. | PCA | 0.0789 | [0.0513, 0.1044] | 9/1/0 | 0.003906 | 1.000 | 3.85/6.85 |
| kNN Acc. | AE | 0.0026 | [-0.0337, 0.0376] | 5/0/5 | 0.695312 | 0.164 | 3.85/3.60 |
| kNN Acc. | UMAP | -0.0073 | [-0.0534, 0.0384] | 5/0/5 | 0.845703 | -0.091 | 3.85/4.10 |
| kNN Acc. | PaCMAP | -0.0073 | [-0.0747, 0.0609] | 5/0/5 | 0.921875 | -0.055 | 3.85/4.00 |
| Sil. | PCA | 0.0162 | [0.0091, 0.0244] | 10/0/0 | 0.001953 | 1.000 | 2.95/6.40 |
| Sil. | AE | 0.0124 | [0.0056, 0.0199] | 9/0/1 | 0.005859 | 0.927 | 2.95/4.60 |
| Sil. | UMAP | 0.0045 | [-0.0171, 0.0274] | 5/0/5 | 0.769531 | 0.127 | 2.95/4.40 |
| Sil. | PaCMAP | 0.0050 | [-0.0210, 0.0302] | 4/0/6 | 1.000000 | -0.018 | 2.95/4.40 |
| DB | PCA | 0.6834 | [0.4172, 0.9733] | 10/0/0 | 0.001953 | 1.000 | 2.90/6.40 |
| DB | AE | 0.5049 | [0.2642, 0.7506] | 9/0/1 | 0.005859 | 0.927 | 2.90/4.50 |
| DB | UMAP | 0.2476 | [-0.3096, 0.8761] | 3/0/7 | 1.000000 | -0.018 | 2.90/3.20 |
| DB | PaCMAP | 0.8262 | [0.3524, 1.3164] | 7/0/3 | 0.027344 | 0.782 | 2.90/6.00 |

*Table D.4.* **GO-LR+NSC vs. PCA.** Accuracy comparison using the same frozen TabPFN-2.5 predictor, where only the front-end compression method differs. Values are mean accuracy with subscripted standard deviation over $5 \times 5$ CV.

| Model / DB | COL | LNG | GLI | SMK | AML | PRS | ARC | TOX |
|---|---|---|---|---|---|---|---|---|
| GO-LR+NSC+TabPFN-2.5 | $88.18_{\pm10.05}$ | $97.44_{\pm2.32}$ | $93.82_{\pm5.81}$ | $74.23_{\pm5.17}$ | $97.54_{\pm3.86}$ | $93.37_{\pm4.48}$ | $90.60_{\pm3.97}$ | $93.33_{\pm4.74}$ |
| Global PCA+TabPFN-2.5 | $78.51_{\pm10.13}$ | $96.16_{\pm2.85}$ | $86.35_{\pm6.73}$ | $69.71_{\pm7.70}$ | $95.52_{\pm4.90}$ | $89.21_{\pm5.23}$ | $81.90_{\pm5.37}$ | $89.47_{\pm6.45}$ |

*Table D.5.* **GO-LR+NSC vs. Lasso-selected features.** Accuracy comparison using the same frozen TabPFN-2.5 predictor. LT denotes Lasso-selected features + TabPFN-2.5, evaluated at different sparsity levels $C$. Values are mean accuracy with subscripted standard deviation over $5 \times 5$ CV.

| Method / DB | COL | LNG | GLI | SMK | AML | PRS | ARC | TOX |
|---|---|---|---|---|---|---|---|---|
| GOTabPFN | $88.18_{\pm10.05}$ | $97.44_{\pm2.32}$ | $93.82_{\pm5.81}$ | $74.23_{\pm5.17}$ | $97.54_{\pm3.86}$ | $93.37_{\pm4.48}$ | $90.60_{\pm3.97}$ | $93.33_{\pm4.74}$ |
| LT ($C = 0.01$) | $85.59_{\pm9.80}$ | $96.76_{\pm2.63}$ | $91.76_{\pm5.88}$ | $68.53_{\pm7.54}$ | $96.10_{\pm4.52}$ | $93.33_{\pm5.06}$ | $89.60_{\pm3.73}$ | $90.06_{\pm5.22}$ |
| LT ($C = 0.02$) | $85.59_{\pm9.80}$ | $96.75_{\pm2.44}$ | $91.76_{\pm5.88}$ | $68.53_{\pm7.54}$ | $96.10_{\pm4.52}$ | $93.33_{\pm5.06}$ | $89.60_{\pm3.73}$ | $92.28_{\pm5.79}$ |
| LT ($C = 0.05$) | $85.59_{\pm10.05}$ | $97.15_{\pm2.33}$ | $92.71_{\pm5.44}$ | $75.14_{\pm6.10}$ | $98.04_{\pm3.21}$ | $92.35_{\pm5.08}$ | $90.30_{\pm4.23}$ | $92.74_{\pm3.90}$ |

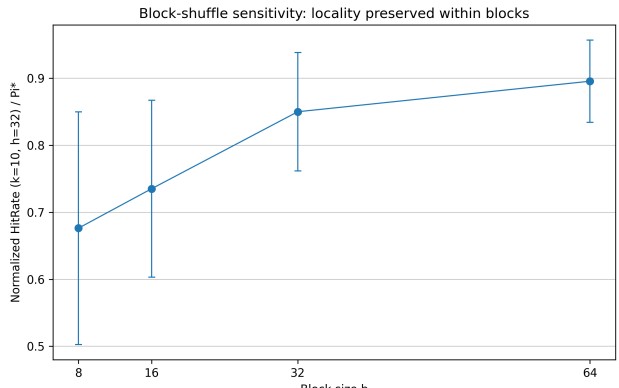

*(a)* **Block-shuffle sensitivity.** Normalized $\text{HitRate}_{k,h}$ (relative to $\Pi^*$) vs. block size $b$.

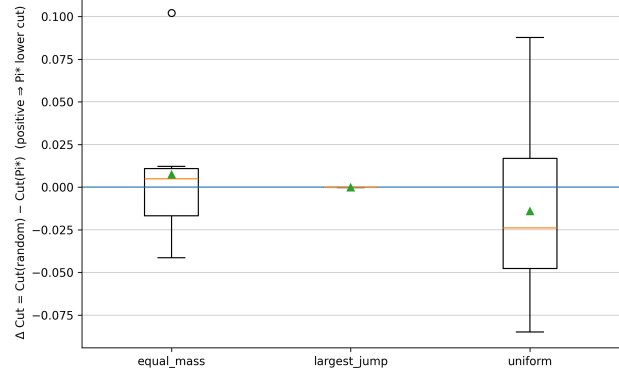

*(b)* **Boundary-cut advantage.** $\Delta\text{Cut} = \text{Cut}(\text{random}) - \text{Cut}(\Pi^*)$ across datasets (positive $\Rightarrow \Pi^*$ lower cut).

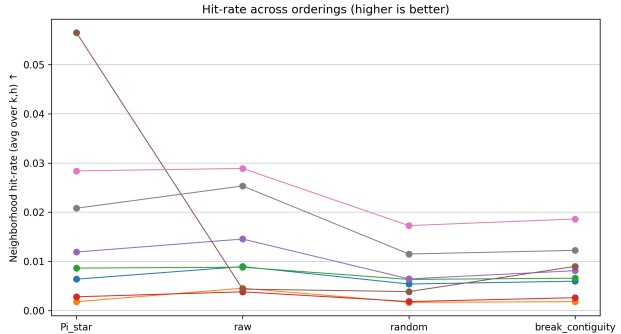

*(c)* **Hit-rate across order families.** $\text{HitRate}_{k,h}$ is highest under $\Pi^*$ and drops when contiguity is broken.

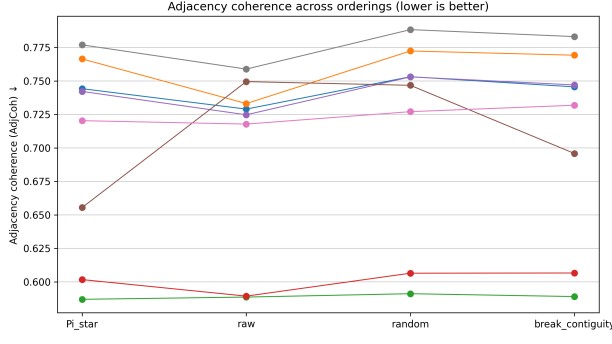

*(d)* **Adjacency coherence across families.** $\text{AdjCoh}$ is lowest (best) under $\Pi^*$ and worse under random.

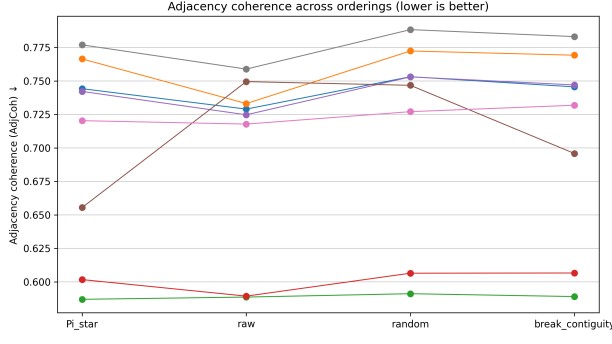

| Test / Statistic | Value |
|---|---|
| Wilcoxon $p$ (AdjCoh: $\Pi^* < $ random) | 0.00390625 |
| Wilcoxon $p$ (HitRate: $\Pi^* > $ random) | 0.0078125 |

| Block size $b$ | Norm. HitRate (mean$\pm$std) |
|---|---|
| 8 | $0.740448 \pm 0.184317$ |
| 16 | $0.888879 \pm 0.176857$ |
| 32 | $0.967536 \pm 0.096630$ |
| 64 | $1.010504 \pm 0.094868$ |

*(e)* **Robust neighborhood gains.** Mean $\Delta\text{HitRate}_{k,h} = \text{HitRate}(\Pi^*) - \text{HitRate}(\text{random})$ over $(k, h)$.

*(f)* **Aggregated stats.** Wilcoxon tests ($n$=8 datasets) and block-shuffle summary (normalized to $\Pi^*$).

*Figure E.1.* **Order-only neighborhood diagnostics and controls.** GO-LR ordering $\Pi^*$ improves local coherence (AdjCoh), increases neighborhood recovery (HitRate), yields favorable segmentation alignment (Cut), and degrades predictably under block-shuffle and contiguity-breaking controls.

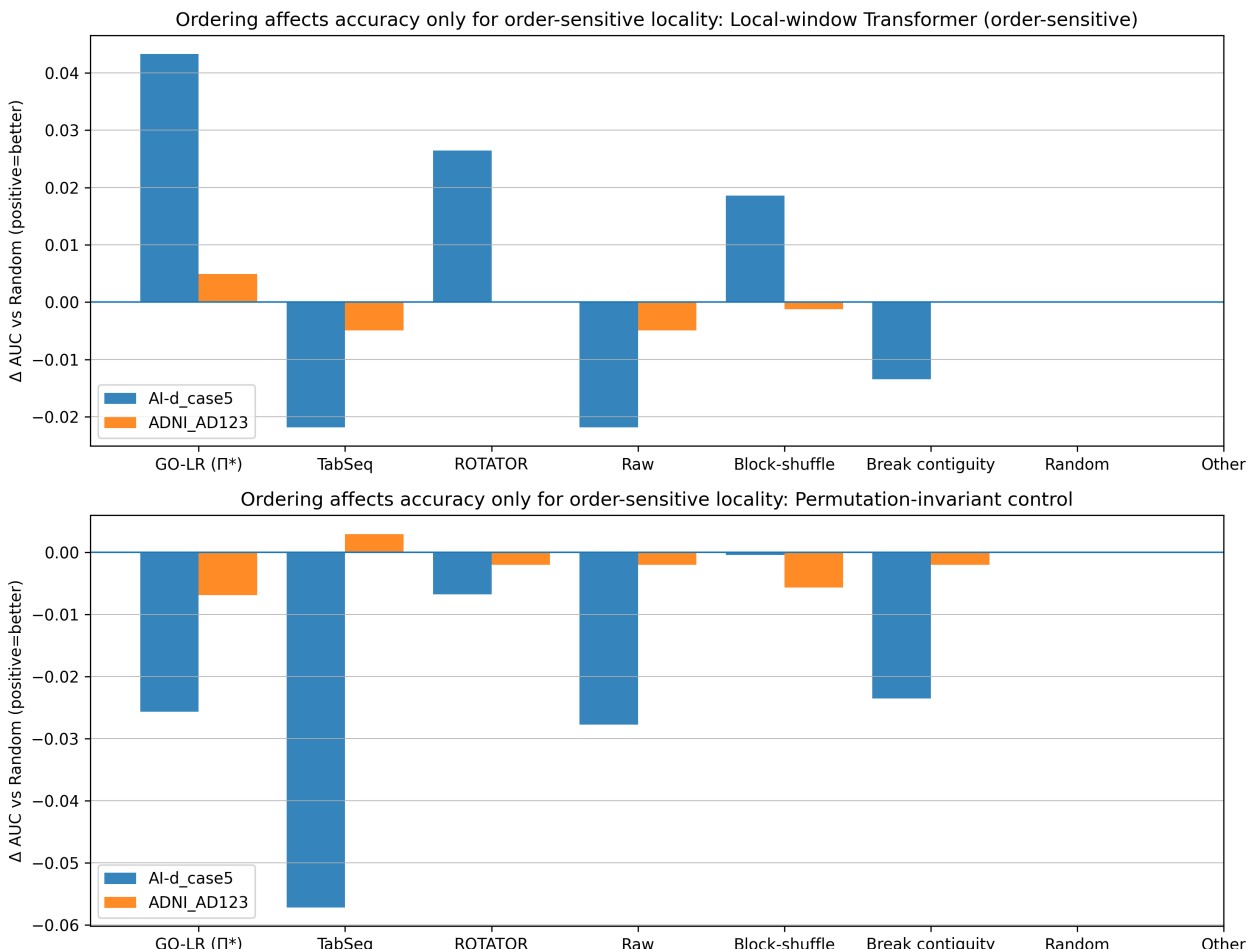

*(a)* **Causal check.** $\Delta$AUC vs. random permutations for an order-sensitive local-window Transformer (top) and a permutation-invariant control (bottom). Ordering changes performance only when the backbone uses locality.

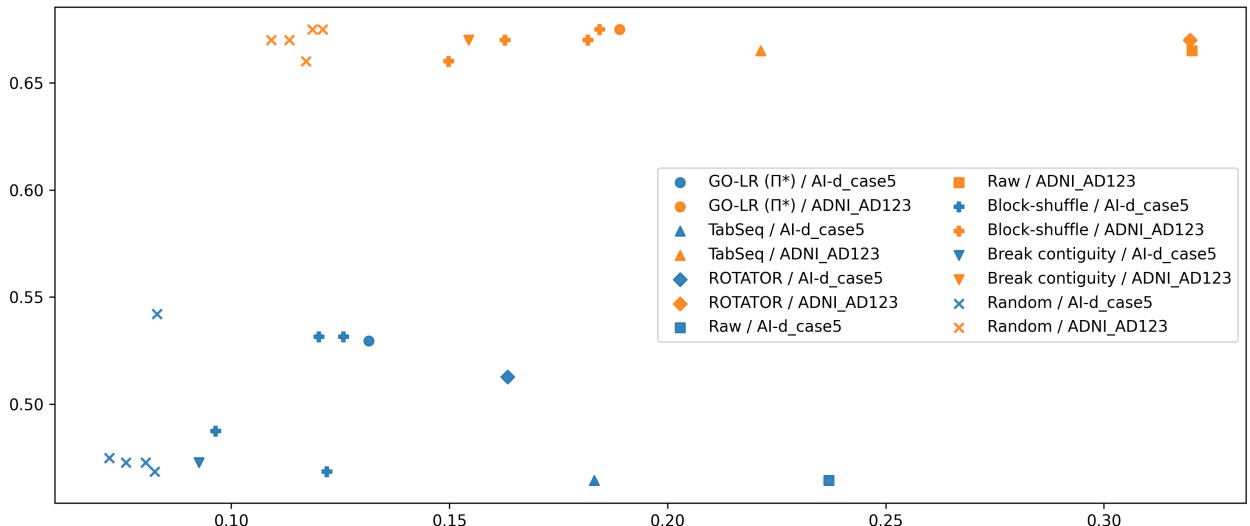

*(b)* **Mechanism.** Neighborhood preservation (HitRate$_{k,h}$) correlates with downstream AUC for the local model, supporting the locality hypothesis.

*Figure E.2.* **Ordering can improve accuracy beyond NSC.** Learned orderings matter for architectures that introduce locality over feature tokens; invariant controls do not exhibit systematic gains.

*Table E.1.* Ordering improves AUC for an order-sensitive local-window Transformer, but not for a permutation-invariant control, on two $n < m$ datasets. Values are mean±std where multiple runs exist (e.g., random permutations); parentheses show $\Delta$AUC relative to the random baseline within each dataset (local model).

| Ordering | Local window Transformer (AUC) | | Permutation-invariant control (AUC) | |
|---|---|---|---|---|
| | AI-d_case5 | ADNI_AD123 | AI-d_case5 | ADNI_AD123 |
| GO-LR ($\Pi^*$) | 0.529 (+0.043) | 0.675 (+0.005) | 0.466 | 0.665 |
| TabSeq | 0.464 (-0.022) | 0.665 (-0.005) | 0.435 | 0.675 |
| ROTATOR | 0.513 (+0.026) | 0.670 (+0.000) | 0.485 | 0.670 |
| Raw | 0.464 (-0.022) | 0.665 (-0.005) | 0.464 | 0.670 |
| Block-shuffle ($b = 8$) | 0.487 (+0.001) | 0.660 (-0.010) | 0.508 | 0.660 |
| Block-shuffle ($b = 16$) | 0.532 (+0.045) | 0.670 (+0.000) | 0.513 | 0.670 |
| Block-shuffle ($b = 32$) | 0.468 (-0.018) | 0.675 (+0.005) | 0.471 | 0.665 |
| Block-shuffle ($b = 64$) | 0.532 (+0.045) | 0.670 (+0.000) | 0.475 | 0.670 |
| Break contiguity | 0.473 (-0.013) | 0.670 (+0.000) | 0.468 | 0.670 |
| Random (baseline) | 0.486±0.031 | 0.670±0.006 | 0.492±0.032 | 0.672±0.003 |

*Table F.1.* When to use ordering through locality: FOE-sorted datasets with IDF/FOE/$P_{\text{success}}$ and order-only locality gains/LES (not used for sorting). Here, AUC denotes the area under the cumulative explained-variance–IDF curve, computed via trapezoidal integration over discrete pairs $\left(\text{IDF}_k = k/n_{\text{total}}, \; \text{CVar}(k)\right)$. Here, HDLSS = High-Dimensional Low-Sample Size, HDHSS = High-Dimensional High-Sample Size, LDLSS = Low-Dimensional Low-Sample Size, LDHSS = Low-Dimensional High-Sample Size.

| Rank | Dataset | Cat. | IDF↓ | FOE↑ | $P_{\text{success}}$ ↑ | $\Delta$AdjCoh↑ | $\Delta$HitRate↑ | $\Delta$Cut↑ | LES↑ | AUC |
|---|---|---|---|---|---|---|---|---|---|---|
| 1 | GLI-85 | HDLSS | 3.770e-03 | 7.037e+04 | 0.996 | 5.942e-03 | 1.465e-04 | 0.0105 | -0.457 | 1.027e-03 |
| 2 | SMK_CAN_187 | HDLSS | 9.153e-03 | 1.194e+04 | 0.991 | 4.770e-03 | 6.445e-04 | 0.102 | 0.222 | 4.629e-03 |
| 3 | ALLAML | HDLSS | 9.959e-03 | 1.008e+04 | 0.99 | 8.937e-03 | 1.387e-03 | -0.0169 | -0.639 | 2.681e-03 |
| 4 | DeepLesion+ | HDHSS | 0.011 | 8.19e+03 | 0.989 | 7.085e-03 | 9.307e-03 | 0.000 | -0.481 | 5.144e-03 |
| 5 | Prostate-GE | HDLSS | 0.0164 | 3.71e+03 | 0.984 | 4.189e-03 | 5.176e-04 | -0.0168 | -0.667 | 8.838e-03 |
| 6 | Arcene | HDLSS | 0.0197 | 2.58e+03 | 0.98 | 0.0912 | 0.0368 | 0.0122 | 0.185 | 6.477e-03 |
| 7 | TOX171 | HDLSS | 0.0292 | 1.17e+03 | 0.971 | 0.0109 | 5.498e-03 | -0.0413 | -0.789 | 0.0116 |
| 8 | Colon | HDLSS | 0.03 | 1.11e+03 | 0.97 | 6.735e-03 | 9.160e-03 | 4.156e-03 | -0.452 | 0.0124 |
| 9 | Lung | HDLSS | 0.0598 | 280 | 0.94 | 0.0114 | 7.168e-03 | 5.703e-03 | -0.428 | 0.0251 |
| 10 | DrivFace | HDLSS | 0.0692 | 209 | 0.931 | 0.112 | 0.058 | -5.279e-03 | 0.274 | 0.0597 |
| 11 | MiniBooNE | LDHSS | 0.18 | 30.9 | 0.82 | 0.0413 | -0.0208 | 0.0712 | 0.0648 | 0.136 |
| 12 | EEG-FE | MixedRegime | 0.264 | 14.4 | 0.736 | 0.0827 | 0.0933 | -7.165e-03 | 0.296 | 0.231 |
| 13 | MOF | MixedRegime | 0.296 | 11.4 | 0.704 | 0.0492 | 0.0653 | 0.0755 | 0.593 | 0.137 |
| 14 | EEG-PD | MixedRegime | 0.357 | 7.85 | 0.643 | 0.202 | 0.244 | 0.134 | 2.76 | 0.301 |
| 15 | AI-D (Case 5) | MixedRegime | 0.415 | 5.81 | 0.585 | 0.0293 | 0.068 | 0.0606 | 0.394 | 0.348 |
| 16 | ADNI (AD123) | MixedRegime | 0.418 | 5.72 | 0.582 | 0.11 | 0.158 | 0.111 | 1.66 | 0.28 |
| 17 | CNAE9 | MixedRegime | 0.662 | 2.28 | 0.338 | 6.173e-04 | -2.290e-03 | 9.856e-07 | -0.575 | 0.269 |
| 18 | MNIST+ | HDHSS | 0.735 | 1.85 | 0.265 | 5.163e-03 | 6.973e-03 | 0.0192 | -0.36 | 0.635 |
| 19 | WDBC | MixedRegime | 0.742 | 1.82 | 0.258 | 0.197 | nan | nan | 2.52 | 0.47 |
| 20 | HAM10000 | HDHSS | 0.757 | 1.74 | 0.243 | 9.772e-03 | 0.0114 | 0.0278 | -0.249 | 0.646 |
| 21 | Fashion MNIST+ | HDHSS | 0.772 | 1.68 | 0.228 | 3.848e-03 | 6.934e-03 | 0.0238 | -0.333 | 0.663 |
| 22 | Dog vs Cat+ | HDHSS | 0.825 | 1.47 | 0.175 | 0.0164 | 0.0131 | -0.0158 | -0.531 | 0.702 |
| 23 | CIFAR-10+ | HDHSS | 0.846 | 1.4 | 0.154 | 0.012 | 0.0162 | -9.114e-03 | -0.487 | 0.71 |
| 24 | Glass | LDLSS | 0.889 | 1.27 | 0.111 | 0.0542 | nan | nan | 0.326 | 0.219 |
| 25 | Cargo | MixedRegime | 0.897 | 1.24 | 0.103 | 0.0956 | 0.156 | 3.076e-03 | 0.773 | 0.405 |
| 26 | Forest Cover | LDHSS | 0.926 | 1.17 | 0.0741 | 0.0309 | 4.444e-03 | 0.0605 | 0.065 | 0.197 |
| 27 | Iris | LDLSS | 1 | 1 | 0.000 | -0.0893 | nan | nan | -1.87 | 0.493 |
| 28 | Higgs | LDHSS | 1 | 1 | 0.000 | 0.064 | nan | nan | 0.477 | 0.275 |
| 29 | BUPA Liver | LDLSS | 1 | 1 | 0.000 | -8.495e-03 | nan | nan | -0.634 | 0.163 |
| 30 | Adult | LDHSS | 1 | 1 | 0.000 | -0.113 | nan | nan | -2.24 | 0.138 |
| 31 | Pima Indian | LDLSS | 1 | 1 | 0.000 | -0.0553 | nan | nan | -1.35 | 0.122 |
| 32 | Water Potability | MixedRegime | 1 | 1 | 0.000 | 0.0325 | nan | nan | -5.418e-03 | 0.106 |
| 33 | Poker Hand | LDHSS | 1 | 1 | 0.000 | 0.0243 | nan | nan | -0.131 | 0.0955 |
| 34 | Hayes-Roth | LDLSS | 1 | 0.000 | 0.000 | 0.131 | nan | nan | 1.51 | 0.000 |
| 35 | Monks-1 | LDLSS | 1 | 0.000 | 0.000 | -0.0388 | nan | nan | -1.1 | 0.000 |

*Table F.2.* **Ordering-locality diagnostics for additional cross-domain datasets.** Datasets are sorted by FOE score. Categories follow the empirical regime rule using $n$, $m$, and $\rho = m/n$. AUC is computed under the cumulative explained variance-IDF curve via trapezoidal integration. LES is standardized within this five-dataset subset.

| Rank | Dataset | Cat. | IDF↓ | FOE↑ | $P_{\text{success}}$ ↑ | $\Delta$AdjCoh↑ | $\Delta$HitRate↑ | $\Delta$Cut↑ | LES↑ | AUC |
|---|---|---|---|---|---|---|---|---|---|---|
| 1 | orlraws10P | HDLSS | 9.317e-03 | 1.152e+04 | 0.991 | 3.487e-03 | 1.641e-03 | -0.0223 | -0.170 | 5.118e-03 |
| 2 | Cell Cycle | MixedRegime | 0.0244 | 1.681e+03 | 0.976 | 8.670e-03 | 1.758e-04 | 7.770e-04 | 0.659 | 4.773e-03 |
| 3 | RELATHE | MixedRegime | 0.292 | 11.77 | 0.709 | -1.728e-03 | -5.078e-04 | -4.570e-03 | -0.664 | 0.148 |
| 4 | BASEHOCK | MixedRegime | 0.362 | 7.649 | 0.638 | 2.262e-04 | 1.270e-04 | 5.039e-03 | 0.0345 | 0.185 |
| 5 | PCMAC | MixedRegime | 0.513 | 3.796 | 0.487 | -3.030e-04 | 2.520e-03 | -0.0111 | 0.141 | 0.261 |

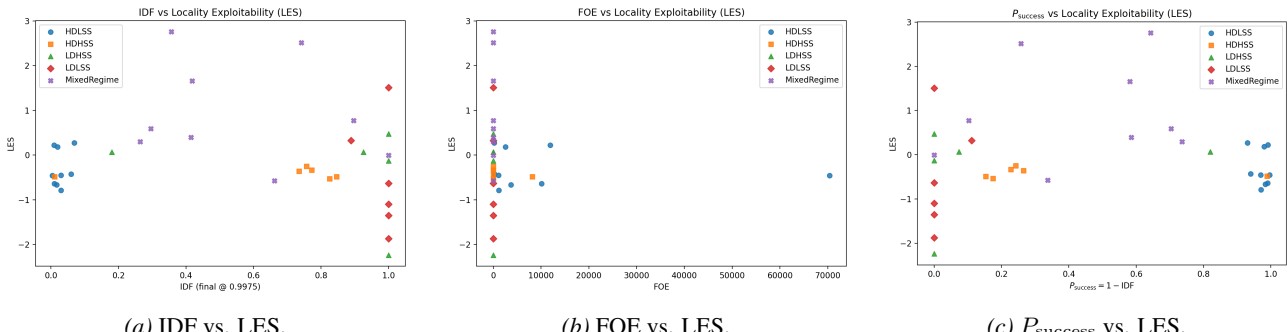

*(a)* IDF vs. LES.           *(b)* FOE vs. LES.           *(c)* $P_{\text{success}}$ vs. LES.

*Figure F.1.* Locality Exploitability Score (LES) against intrinsic-dimension/compression proxies.

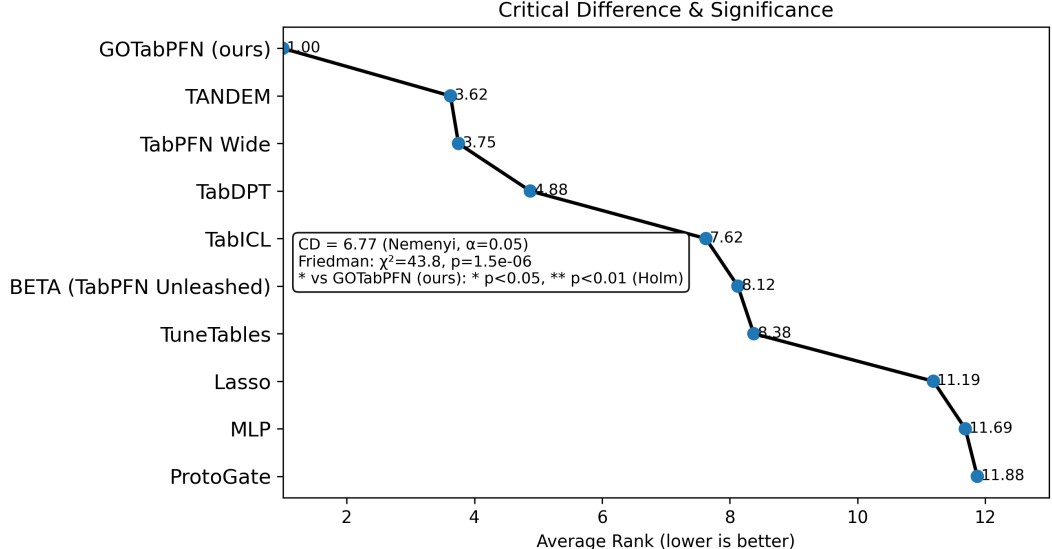

*Figure I.1.* **Average-rank comparison on the 8 HDLSS datasets.** Lower rank is better. Friedman/Nemenyi analysis shows GOTabPFN as the best-ranked method on the original HDLSS benchmark.

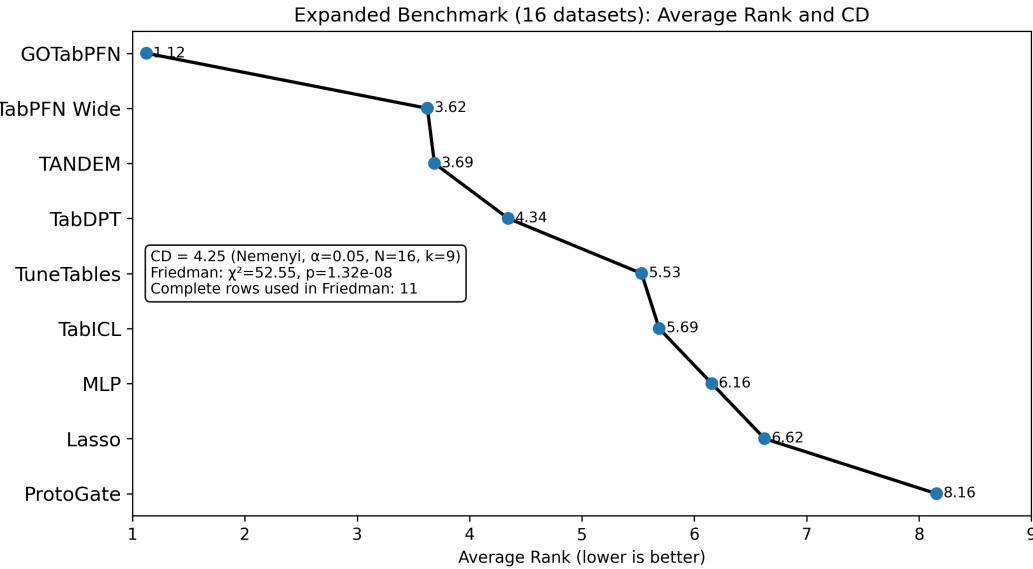

*Figure I.2.* **Average-rank comparison on the expanded 16-dataset benchmark.** Lower rank is better. GOTabPFN achieves the best average rank (1.12), followed by TabPFN-Wide (3.62) and TANDEM (3.69), with a significant Friedman test ($\chi^2 = 52.55$, $p = 1.32 \times 10^{-8}$).

*Table G.1.* Performance of the models on 8 HDLSS datasets (mean accuracy with subscripted standard deviation over 5×5 CV). Dataset abbreviations: COL = Colon, LNG = Lung, GLI = GLI-85, SMK = SMK_CAN_187, AML = ALLAML, PRS = Prostate-GE, ARC = Arcene, TOX = TOX-171. Model abbreviations: GOTabPFN$_{ours}$ = our method, TWide = TabPFN Wide, TTables = TuneTables, BETA = TabPFN Unleashed, PGate = ProtoGate, TRNN = TabulaRNN, RF = Random Forest, NB = Naive Bayes, DT = Decision Tree, MambAtt = MambAttention, FT-T = FT-Transformer, CatEmbed = CategoryEmbedding, ResNetT = ResNetTabular, Tab-T = TabTransformer.

| MODEL | COL | LNG | GLI | SMK | AML | PRS | ARC | TOX | AVG. RANK |
|---|---|---|---|---|---|---|---|---|---|
| GOTABPFN$_{OURS}$ | **88.18**$_{\pm10.05}$ | **97.44**$_{\pm2.32}$ | **93.82**$_{\pm5.81}$ | **74.23**$_{\pm5.17}$ | **97.54**$_{\pm3.86}$ | **93.37**$_{\pm4.48}$ | **90.60**$_{\pm3.97}$ | **93.33**$_{\pm4.74}$ | 1.00$_{\pm0.00}$ |
| TANDEM | 86.15$_{\pm7.75}$ | 96.46$_{\pm2.88}$ | 91.53$_{\pm6.02}$ | 72.72$_{\pm5.69}$ | 95.81$_{\pm5.53}$ | 91.55$_{\pm4.32}$ | 86.90$_{\pm6.34}$ | 93.08$_{\pm2.61}$ | 3.63$_{\pm1.32}$ |
| TWIDE | 87.85$_{\pm7.28}$ | 96.55$_{\pm2.15}$ | 88.47$_{\pm5.75}$ | 68.78$_{\pm8.60}$ | 97.16$_{\pm4.10}$ | 93.10$_{\pm5.92}$ | 88.00$_{\pm5.20}$ | 89.35$_{\pm4.95}$ | 3.75$_{\pm2.38}$ |
| TABDPT | 86.26$_{\pm7.27}$ | 96.05$_{\pm2.57}$ | 87.76$_{\pm5.86}$ | 71.99$_{\pm7.32}$ | 96.32$_{\pm4.15}$ | 90.94$_{\pm5.72}$ | 82.10$_{\pm6.48}$ | 93.25$_{\pm3.44}$ | 4.88$_{\pm1.69}$ |
| TABICL | 84.62$_{\pm10.52}$ | 96.36$_{\pm2.61}$ | 87.06$_{\pm6.23}$ | 68.73$_{\pm7.28}$ | 95.52$_{\pm5.59}$ | 90.17$_{\pm5.93}$ | 82.60$_{\pm6.14}$ | 88.78$_{\pm5.92}$ | 7.63$_{\pm2.29}$ |
| BETA | 84.73$_{\pm9.36}$ | 94.38$_{\pm3.34}$ | 86.21$_{\pm8.91}$ | 70.21$_{\pm5.61}$ | 95.67$_{\pm7.54}$ | 87.53$_{\pm4.67}$ | 86.45$_{\pm5.92}$ | 90.38$_{\pm6.42}$ | 8.13$_{\pm4.31}$ |
| TTABLES | 86.80$_{\pm2.14}$ | 94.37$_{\pm2.35}$ | 89.66$_{\pm3.12}$ | 70.28$_{\pm6.46}$ | 95.80$_{\pm2.14}$ | 93.31$_{\pm2.83}$ | 81.40$_{\pm3.66}$ | 77.96$_{\pm2.55}$ | 8.38$_{\pm7.70}$ |
| LASSO | 79.40$_{\pm10.18}$ | 94.47$_{\pm4.39}$ | 85.88$_{\pm4.71}$ | 61.19$_{\pm13.72}$ | 87.24$_{\pm3.39}$ | 91.18$_{\pm6.39}$ | 81.00$_{\pm3.39}$ | 91.86$_{\pm6.03}$ | 11.13$_{\pm5.06}$ |
| MLP | 83.95$_{\pm9.80}$ | 96.47$_{\pm2.69}$ | 85.41$_{\pm8.00}$ | 59.05$_{\pm7.44}$ | 89.98$_{\pm9.17}$ | 89.20$_{\pm6.07}$ | 78.40$_{\pm4.05}$ | 92.48$_{\pm4.28}$ | 11.63$_{\pm5.45}$ |
| PGATE | 83.95$_{\pm9.82}$ | 93.44$_{\pm6.37}$ | 82.48$_{\pm5.68}$ | 60.16$_{\pm5.10}$ | 86.12$_{\pm3.34}$ | 90.58$_{\pm5.72}$ | 81.50$_{\pm5.10}$ | 92.34$_{\pm5.67}$ | 12.06$_{\pm4.77}$ |
| TRNN | 84.20$_{\pm6.50}$ | 90.50$_{\pm4.80}$ | 79.68$_{\pm6.68}$ | 60.02$_{\pm3.18}$ | 88.92$_{\pm2.02}$ | 90.50$_{\pm6.00}$ | 81.50$_{\pm5.10}$ | 85.80$_{\pm4.70}$ | 14.81$_{\pm6.03}$ |
| REALMLP | 76.28$_{\pm13.16}$ | 93.39$_{\pm3.81}$ | 82.59$_{\pm13.31}$ | 64.93$_{\pm9.86}$ | 96.65$_{\pm5.78}$ | 85.17$_{\pm11.94}$ | 79.70$_{\pm6.01}$ | 88.67$_{\pm5.32}$ | 15.13$_{\pm7.36}$ |
| LGBM | 76.60$_{\pm11.67}$ | 93.42$_{\pm5.91}$ | 85.88$_{\pm11.53}$ | 58.85$_{\pm10.14}$ | 85.81$_{\pm5.67}$ | 91.38$_{\pm5.71}$ | 80.50$_{\pm5.79}$ | 81.98$_{\pm6.25}$ | 16.00$_{\pm6.08}$ |
| CATBOOST | 72.65$_{\pm10.12}$ | 91.57$_{\pm5.74}$ | 84.71$_{\pm12.11}$ | 58.28$_{\pm12.16}$ | 91.71$_{\pm8.22}$ | 90.24$_{\pm6.87}$ | 81.00$_{\pm2.00}$ | 81.95$_{\pm7.47}$ | 17.38$_{\pm6.38}$ |
| STG | 79.55$_{\pm10.53}$ | 93.30$_{\pm6.28}$ | 82.48$_{\pm4.56}$ | 57.25$_{\pm6.82}$ | 86.08$_{\pm5.60}$ | 89.38$_{\pm5.85}$ | 74.40$_{\pm6.90}$ | 87.95$_{\pm5.01}$ | 18.25$_{\pm4.92}$ |
| RF | 80.05$_{\pm10.37}$ | 91.73$_{\pm6.61}$ | 85.88$_{\pm8.80}$ | 58.29$_{\pm10.61}$ | 85.71$_{\pm5.71}$ | 90.38$_{\pm7.31}$ | 74.00$_{\pm1.22}$ | 79.78$_{\pm7.10}$ | 18.50$_{\pm5.57}$ |
| REAL-X | 76.75$_{\pm12.21}$ | 93.27$_{\pm4.32}$ | 83.24$_{\pm5.56}$ | 56.48$_{\pm4.90}$ | 84.16$_{\pm5.68}$ | 86.75$_{\pm6.68}$ | 77.30$_{\pm6.10}$ | 90.79$_{\pm4.75}$ | 19.50$_{\pm6.18}$ |
| LLSPIN | 79.35$_{\pm7.74}$ | 70.10$_{\pm12.31}$ | 84.42$_{\pm7.12}$ | 61.16$_{\pm7.92}$ | 88.12$_{\pm1.26}$ | 88.71$_{\pm5.98}$ | 80.80$_{\pm4.90}$ | 81.67$_{\pm9.01}$ | 19.50$_{\pm8.50}$ |
| LSPIN | 81.30$_{\pm7.97}$ | 76.92$_{\pm9.38}$ | 83.48$_{\pm6.62}$ | 58.92$_{\pm6.78}$ | 84.46$_{\pm3.36}$ | 87.75$_{\pm6.74}$ | 78.60$_{\pm5.80}$ | 83.47$_{\pm8.59}$ | 19.88$_{\pm5.16}$ |
| TABR | 80.75$_{\pm8.40}$ | 86.70$_{\pm6.40}$ | 81.42$_{\pm6.64}$ | 58.46$_{\pm6.68}$ | 80.84$_{\pm2.24}$ | 84.50$_{\pm8.00}$ | 75.85$_{\pm6.50}$ | 86.50$_{\pm5.00}$ | 21.56$_{\pm4.15}$ |
| KNN | 71.65$_{\pm12.03}$ | 91.06$_{\pm5.41}$ | 83.53$_{\pm5.76}$ | 52.05$_{\pm13.13}$ | 79.05$_{\pm8.02}$ | 78.78$_{\pm9.20}$ | 82.50$_{\pm5.00}$ | 83.86$_{\pm7.07}$ | 22.75$_{\pm9.36}$ |
| MLP-PLR | 71.41$_{\pm11.32}$ | 77.32$_{\pm12.81}$ | 80.94$_{\pm9.50}$ | 60.79$_{\pm8.28}$ | 92.84$_{\pm7.68}$ | 86.89$_{\pm8.69}$ | 66.30$_{\pm12.52}$ | 81.85$_{\pm8.32}$ | 22.88$_{\pm8.33}$ |
| ADABOOST | 78.97$_{\pm10.96}$ | 78.32$_{\pm1.93}$ | 85.88$_{\pm7.97}$ | 58.28$_{\pm9.55}$ | 84.38$_{\pm8.32}$ | 89.19$_{\pm4.94}$ | 75.50$_{\pm5.34}$ | 57.85$_{\pm9.01}$ | 23.38$_{\pm8.62}$ |
| XGBOOST | 72.60$_{\pm12.59}$ | 86.61$_{\pm8.72}$ | 77.06$_{\pm7.80}$ | 55.59$_{\pm6.34}$ | 90.38$_{\pm9.39}$ | 82.55$_{\pm10.22}$ | 81.50$_{\pm4.06}$ | 70.13$_{\pm7.85}$ | 24.38$_{\pm8.51}$ |
| GBM | 77.31$_{\pm14.43}$ | 91.59$_{\pm2.52}$ | 80.00$_{\pm8.80}$ | 58.31$_{\pm9.72}$ | 82.10$_{\pm6.67}$ | 82.24$_{\pm5.34}$ | 82.00$_{\pm3.32}$ | 53.85$_{\pm15.31}$ | 24.50$_{\pm9.80}$ |
| SVM | 70.75$_{\pm13.93}$ | 72.77$_{\pm8.33}$ | 85.88$_{\pm2.88}$ | 61.09$_{\pm11.78}$ | 83.14$_{\pm13.37}$ | 85.75$_{\pm6.63}$ | 77.00$_{\pm2.92}$ | 66.75$_{\pm7.86}$ | 24.50$_{\pm10.49}$ |
| INVASE | 75.40$_{\pm10.10}$ | 91.22$_{\pm6.16}$ | 80.14$_{\pm4.56}$ | 36.42$_{\pm4.46}$ | 78.90$_{\pm2.26}$ | 88.00$_{\pm6.50}$ | 71.20$_{\pm7.80}$ | 79.94$_{\pm6.60}$ | 26.63$_{\pm7.76}$ |
| NB | 83.85$_{\pm10.56}$ | 84.24$_{\pm3.99}$ | 82.35$_{\pm3.72}$ | 59.32$_{\pm14.95}$ | 88.57$_{\pm2.86}$ | 60.86$_{\pm14.63}$ | 53.50$_{\pm8.31}$ | 58.49$_{\pm8.14}$ | 27.13$_{\pm12.07}$ |
| DT | 83.85$_{\pm9.15}$ | 85.16$_{\pm5.58}$ | 78.82$_{\pm9.56}$ | 56.63$_{\pm6.94}$ | 75.14$_{\pm10.18}$ | 82.33$_{\pm5.13}$ | 73.50$_{\pm5.39}$ | 45.68$_{\pm8.70}$ | 28.63$_{\pm9.01}$ |
| MAMBATAB | 80.75$_{\pm8.40}$ | 87.85$_{\pm6.00}$ | 80.16$_{\pm6.64}$ | 54.98$_{\pm9.20}$ | 68.16$_{\pm4.90}$ | 58.52$_{\pm15.60}$ | 64.00$_{\pm9.60}$ | 82.30$_{\pm6.00}$ | 28.81$_{\pm9.28}$ |
| TABSEQ | 72.00$_{\pm11.24}$ | 86.81$_{\pm3.98}$ | 75.29$_{\pm9.98}$ | 65.16$_{\pm7.50}$ | 77.28$_{\pm14.49}$ | 65.24$_{\pm10.76}$ | 65.30$_{\pm6.45}$ | 47.95$_{\pm7.27}$ | 30.50$_{\pm10.64}$ |
| MAMBULAR | 83.55$_{\pm7.10}$ | 89.90$_{\pm5.10}$ | 46.92$_{\pm4.56}$ | 32.90$_{\pm15.12}$ | 60.80$_{\pm5.77}$ | 81.12$_{\pm8.02}$ | 69.65$_{\pm8.20}$ | 84.95$_{\pm5.20}$ | 30.50$_{\pm12.46}$ |
| MAMBATT | 81.90$_{\pm8.00}$ | 78.00$_{\pm9.10}$ | 76.18$_{\pm7.78}$ | 52.18$_{\pm4.46}$ | 70.16$_{\pm6.28}$ | 61.99$_{\pm14.45}$ | 49.90$_{\pm12.90}$ | 80.90$_{\pm6.40}$ | 31.75$_{\pm8.82}$ |
| FT-T | 69.25$_{\pm12.50}$ | 67.30$_{\pm12.20}$ | 52.46$_{\pm8.92}$ | 56.45$_{\pm9.68}$ | 56.12$_{\pm12.10}$ | 85.00$_{\pm7.00}$ | 52.30$_{\pm12.30}$ | 79.45$_{\pm6.90}$ | 35.63$_{\pm7.45}$ |
| SAINT | 67.60$_{\pm13.10}$ | 78.00$_{\pm9.10}$ | 78.56$_{\pm9.36}$ | 50.34$_{\pm12.16}$ | 52.92$_{\pm14.15}$ | 61.99$_{\pm14.45}$ | 57.10$_{\pm11.20}$ | 75.10$_{\pm8.20}$ | 35.63$_{\pm5.75}$ |
| TABM | 65.80$_{\pm13.70}$ | 74.70$_{\pm10.10}$ | 60.46$_{\pm4.42}$ | 42.90$_{\pm6.36}$ | 66.67$_{\pm3.34}$ | 75.03$_{\pm10.08}$ | 66.10$_{\pm9.10}$ | 70.20$_{\pm9.60}$ | 35.94$_{\pm3.57}$ |
| AUTOINT | 65.80$_{\pm13.70}$ | 74.70$_{\pm10.10}$ | 48.44$_{\pm5.65}$ | 49.68$_{\pm9.80}$ | 58.34$_{\pm9.78}$ | 83.47$_{\pm7.53}$ | 67.90$_{\pm8.70}$ | 66.80$_{\pm10.60}$ | 36.00$_{\pm5.27}$ |
| CATEMBED | 60.50$_{\pm15.50}$ | 72.95$_{\pm10.60}$ | 69.14$_{\pm9.46}$ | 59.16$_{\pm6.68}$ | 78.12$_{\pm6.90}$ | 45.01$_{\pm19.50}$ | 54.80$_{\pm11.80}$ | 57.80$_{\pm13.30}$ | 36.38$_{\pm9.50}$ |
| RESNETT | 64.10$_{\pm14.30}$ | 74.70$_{\pm10.10}$ | 66.67$_{\pm12.16}$ | 52.16$_{\pm4.46}$ | 75.10$_{\pm3.30}$ | 48.43$_{\pm18.70}$ | 42.60$_{\pm14.70}$ | 63.30$_{\pm11.70}$ | 38.88$_{\pm5.67}$ |
| 1D CNN | 58.70$_{\pm16.10}$ | 63.50$_{\pm13.30}$ | 56.92$_{\pm5.62}$ | 40.92$_{\pm10.42}$ | 60.42$_{\pm9.92}$ | 70.00$_{\pm12.00}$ | 54.80$_{\pm11.80}$ | 71.85$_{\pm9.10}$ | 39.19$_{\pm4.35}$ |
| NODE | 54.95$_{\pm17.40}$ | 55.10$_{\pm15.70}$ | 64.92$_{\pm12.34}$ | 46.58$_{\pm10.45}$ | 76.92$_{\pm10.08}$ | 58.52$_{\pm15.60}$ | 59.40$_{\pm10.70}$ | 65.10$_{\pm11.10}$ | 39.94$_{\pm5.36}$ |
| TABNET | 56.75$_{\pm15.20}$ | 80.14$_{\pm12.23}$ | 55.29$_{\pm10.26}$ | 48.67$_{\pm2.17}$ | 63.89$_{\pm4.17}$ | 66.55$_{\pm15.33}$ | 50.00$_{\pm7.55}$ | 41.68$_{\pm9.03}$ | 40.38$_{\pm6.02}$ |
| L2X | 57.60$_{\pm13.48}$ | 50.02$_{\pm14.26}$ | 56.92$_{\pm3.58}$ | 45.68$_{\pm9.78}$ | 76.28$_{\pm2.36}$ | 61.78$_{\pm13.69}$ | 72.95$_{\pm7.30}$ | 31.72$_{\pm9.11}$ | 40.75$_{\pm7.50}$ |
| TROMPT | 58.70$_{\pm16.10}$ | 61.55$_{\pm13.90}$ | 43.96$_{\pm12.16}$ | 46.65$_{\pm12.12}$ | 46.12$_{\pm4.46}$ | 69.08$_{\pm12.20}$ | 40.10$_{\pm15.30}$ | 66.80$_{\pm10.60}$ | 42.38$_{\pm4.90}$ |
| DCN | 48.30$_{\pm19.20}$ | 57.25$_{\pm15.10}$ | 42.86$_{\pm8.84}$ | 34.92$_{\pm14.10}$ | 43.78$_{\pm16.10}$ | 85.02$_{\pm7.05}$ | 61.75$_{\pm10.10}$ | 51.90$_{\pm15.00}$ | 43.31$_{\pm8.41}$ |
| DANETS | 50.60$_{\pm18.60}$ | 76.35$_{\pm9.60}$ | 35.90$_{\pm6.78}$ | 29.16$_{\pm16.10}$ | 56.98$_{\pm4.48}$ | 78.58$_{\pm9.10}$ | 30.30$_{\pm17.80}$ | 59.60$_{\pm12.80}$ | 43.63$_{\pm7.03}$ |
| TANGOS | 38.20$_{\pm21.70}$ | 48.10$_{\pm17.50}$ | 64.84$_{\pm10.42}$ | 32.68$_{\pm6.80}$ | 50.12$_{\pm8.86}$ | 65.47$_{\pm13.30}$ | 27.90$_{\pm18.40}$ | 68.50$_{\pm10.10}$ | 44.94$_{\pm7.20}$ |
| DEEPFM | 56.90$_{\pm16.80}$ | 59.40$_{\pm14.50}$ | 43.28$_{\pm7.82}$ | 36.12$_{\pm12.26}$ | 48.16$_{\pm12.56}$ | 45.01$_{\pm19.50}$ | 64.00$_{\pm9.60}$ | 55.90$_{\pm13.90}$ | 45.00$_{\pm4.70}$ |
| ENODE | 52.80$_{\pm18.00}$ | 50.45$_{\pm16.90}$ | 58.92$_{\pm13.12}$ | 26.12$_{\pm5.68}$ | 71.16$_{\pm4.46}$ | 52.05$_{\pm17.95}$ | 35.20$_{\pm16.50}$ | 45.40$_{\pm16.80}$ | 45.75$_{\pm5.33}$ |
| MODERNNCA | 40.85$_{\pm21.10}$ | 43.10$_{\pm18.80}$ | 27.68$_{\pm14.10}$ | 31.16$_{\pm11.67}$ | 33.47$_{\pm10.24}$ | 71.95$_{\pm11.12}$ | 32.80$_{\pm17.10}$ | 68.50$_{\pm10.10}$ | 46.69$_{\pm7.53}$ |
| TABPFN-2.5 | 86.85$_{9.16}$ | N/A | N/A | N/A | N/A | N/A | N/A | N/A | 46.75$_{16.54}$ |
| TAB-T | 46.44$_{\pm16.84}$ | 21.01$_{\pm12.53}$ | 47.77$_{\pm17.71}$ | 50.26$_{\pm8.56}$ | 53.70$_{\pm15.05}$ | 51.01$_{\pm10.64}$ | 48.20$_{\pm7.57}$ | 23.66$_{\pm7.84}$ | 46.88$_{\pm5.04}$ |
| NDTF | 48.30$_{\pm19.20}$ | 63.50$_{\pm13.30}$ | 26.14$_{\pm16.10}$ | 36.18$_{\pm4.48}$ | 43.14$_{\pm6.94}$ | 54.98$_{\pm16.80}$ | 37.70$_{\pm15.90}$ | 43.10$_{\pm17.30}$ | 48.00$_{\pm2.93}$ |
| TABPFN v2 | N/A | N/A | N/A | N/A | N/A | N/A | N/A | N/A | 53.13$_{\pm0.33}$ |
| LOCALPFN | N/A | N/A | N/A | N/A | N/A | N/A | N/A | N/A | 53.13$_{\pm0.33}$ |
| TABPFN v1 | N/A | N/A | N/A | N/A | N/A | N/A | N/A | N/A | 53.13$_{\pm0.33}$ |

*Table G.2.* **Evaluation beyond accuracy on 8 HDLSS datasets.** ROC-AUC and macro-F1 comparisons under $5 \times 5$ CV. Values are mean with subscripted standard deviation. Bold denotes the best result per dataset/metric.

| Metric | Model | COL | LNG | AML | TOX | PRS | ARC | SMK | GLI |
|---|---|---|---|---|---|---|---|---|---|
| ROC-AUC | GOTabPFN | **91.40**$_{\pm 10.12}$ | **99.57**$_{\pm 0.60}$ | **99.36**$_{\pm 1.70}$ | 99.23$_{\pm 0.63}$ | **94.27**$_{\pm 5.87}$ | **95.77**$_{\pm 2.35}$ | **80.55**$_{\pm 5.67}$ | 90.97$_{\pm 7.09}$ |
| | TabICL | 89.37$_{\pm 10.24}$ | 99.39$_{\pm 0.46}$ | 98.27$_{\pm 4.05}$ | 98.62$_{\pm 1.15}$ | 93.92$_{\pm 5.56}$ | 91.48$_{\pm 3.84}$ | 73.27$_{\pm 8.14}$ | 91.41$_{\pm 7.25}$ |
| | TabDPT | 90.80$_{\pm 7.28}$ | 99.27$_{\pm 0.97}$ | 99.28$_{\pm 0.62}$ | **99.93**$_{\pm 0.14}$ | 94.07$_{\pm 3.50}$ | 91.33$_{\pm 4.51}$ | 76.34$_{\pm 7.90}$ | **93.33**$_{\pm 7.76}$ |
| Macro-F1 | GOTabPFN | **87.14**$_{\pm 10.93}$ | **95.73**$_{\pm 5.37}$ | **96.17**$_{\pm 5.97}$ | 92.91$_{\pm 4.01}$ | **90.32**$_{\pm 7.63}$ | **88.92**$_{\pm 4.92}$ | **75.46**$_{\pm 5.12}$ | **93.54**$_{\pm 4.84}$ |
| | TabICL | 78.84$_{\pm 14.16}$ | 93.72$_{\pm 7.09}$ | 92.90$_{\pm 9.29}$ | 88.89$_{\pm 5.91}$ | 90.13$_{\pm 5.92}$ | 79.58$_{\pm 6.53}$ | 71.04$_{\pm 6.85}$ | 90.91$_{\pm 4.62}$ |
| | TabDPT | 84.17$_{\pm 10.91}$ | 94.71$_{\pm 4.91}$ | 95.45$_{\pm 4.89}$ | **93.62**$_{\pm 2.37}$ | 90.27$_{\pm 6.42}$ | 82.08$_{\pm 6.64}$ | 70.07$_{\pm 7.99}$ | 85.25$_{\pm 9.20}$ |

*Table H.1.* Best GOTabPFN hyperparameters for 8 HDLSS datasets (COL = Colon, LNG = Lung, GLI = GLI-85, SMK = SMK_CAN_187, AML = ALLAML, PRS = Prostate-GE, ARC = Arcene, TOX = TOX-171). GO-LR metric: eucl.=euclidean, manh.=manhattan, corr.=correlation, KL=kl_divergence. NSC seg: unif.=uniform, eq-mass=equal_mass, lrg-jump=largest_jump. NSC rule: gam.=gamma, def.=default. Std? indicates `assume_standardized`.

| PARAM | COL | LNG | GLI | SMK | AML | PRS | ARC | TOX |
|---|---|---|---|---|---|---|---|---|
| GO-LR METRIC | EUCL. | MANH. | CORR. | CORR. | KL | MANH. | EUCL. | KL |
| GO-LR $k$ (CLUSTERS) | 10 | 11 | 7 | 9 | 5 | 12 | 4 | 10 |
| GO-LR REFINE PASSES | 3 | 1 | 3 | 1 | 2 | 2 | 1 | 3 |
| GO-LR DIR SELECT | T | T | F | F | T | T | F | T |
| GO-LR FEAT-SUB | – | – | – | 3000 | – | – | – | – |
| NSC SEG | EQ-MASS | UNIF. | UNIF. | UNIF. | LRG-JUMP | LRG-JUMP | UNIF. | EQ-MASS |
| NSC RULE | IDF | DEF. | GAM. | DEF. | GAM. | GAM. | GAM. | GAM. |
| NSC $\tau$ | 0.99 | 0.99 | 0.99 | 0.99 | 0.95 | 0.99 | 0.99 | 0.99 |
| NSC $\gamma$ | 1.76 | 2.38 | 2.92 | 2.90 | 2.40 | 2.91 | 2.26 | 2.86 |
| NSC $\beta$ | 0.22 | 0.79 | 0.17 | 0.08 | 0.32 | 0.84 | 0.19 | 0.65 |
| NSC $M_{\min}$ | 64 | 64 | 64 | 48 | 64 | 48 | 16 | 32 |
| NSC $M_{\max}$ | 384 | 384 | 640 | 384 | 512 | 256 | 640 | 384 |
| NSC $l_{\min}$ | 16 | 12 | 12 | 16 | 16 | 8 | 16 | 16 |
| STD? (ASSUME STD.) | F | F | F | T | F | F | T | F |
| TABPFN SEED | 42 | 42 | 42 | 2 | 3 | 3 | 4 | 4 |

*Table H.2.* **Best GOTabPFN hyperparameters for 8 cross-domain datasets** (ORL = orlraws10P, BAS = BASEHOCK, REL = RELATHE, PCM = PCMAC, CCY = Cell Cycle, CIF = CIFAR-10, DF-R = DrivFace-Regression, DF-C = DrivFace-Classification). GO-LR metric: cos.=cosine, manh.=manhattan, corr.=correlation, KL=kl_divergence. NSC seg: unif.=uniform, lrg-jump=largest_jump. NSC rule: def.=default. Std? indicates `assume_standardized`.

| PARAM | ORL | BAS | REL | PCM | CCY | CIF | DF-R | DF-C |
|---|---|---|---|---|---|---|---|---|
| GO-LR METRIC | COS. | KL | MANH. | COS. | CORR. | MANH. | MANH. | MANH. |
| GO-LR $k$ (CLUSTERS) | 5 | 10 | 6 | 4 | 4 | 11 | 5 | 5 |
| GO-LR REFINE PASSES | 1 | 2 | 2 | 2 | 1 | 2 | 1 | 1 |
| GO-LR DIR SELECT | F | F | T | T | F | F | F | F |
| GO-LR FEAT-SUB | 3000 | – | 2000 | 2000 | 3000 | 2000 | 2000 | 2000 |
| NSC SEG | UNIF. | UNIF. | UNIF. | UNIF. | UNIF. | UNIF. | LRG-JUMP | LRG-JUMP |
| NSC RULE | DEF. | DEF. | DEF. | DEF. | IDF | DEF. | IDF | IDF |
| NSC $\tau$ | 0.99 | 0.99 | 0.95 | 0.95 | 0.95 | 0.95 | 0.99 | 0.99 |
| NSC $\gamma$ | 2.05 | 1.98 | 2.12 | 1.07 | 2.26 | 1.60 | 2.65 | 2.65 |
| NSC $\beta$ | 0.39 | 0.16 | 0.06 | 0.50 | 0.58 | 0.21 | 0.04 | 0.04 |
| NSC $M_{\min}$ | 32 | 48 | 16 | 48 | 48 | 32 | 16 | 16 |
| NSC $M_{\max}$ | 384 | 640 | 384 | 384 | 384 | 384 | 256 | 256 |
| NSC $l_{\min}$ | 12 | 12 | 12 | 12 | 12 | 12 | 12 | 12 |
| STD? (ASSUME STD.) | F | T | T | T | T | T | T | T |
| TABPFN SEED | 42 | 3 | 1 | 4 | 4 | 3 | 3 | 3 |

*Table I.1.* Pairwise significance of GOTabPFN against the top baselines across the 8 HDLSS datasets. $\Delta$Acc denotes the mean accuracy improvement of GOTabPFN over each baseline (percentage points) averaged across datasets. We report Wilcoxon signed-rank $p$-values ($p_{\text{raw}}$) and Holm-corrected $p$-values ($p_{\text{Holm}}$) for multiple comparisons; Sig. indicates significance after Holm correction ($\alpha = 0.05$).

| Baseline | $\Delta$Acc (pp) | $n_{\text{ds}}$ | $p_{\text{raw}}$ | $p_{\text{Holm}}$ | Sig. |
|---|---|---|---|---|---|
| TANDEM | 1.79 | 8 | 0.00781 | 0.0703 | n.s. |
| TabPFN Wide | 2.41 | 8 | 0.00781 | 0.0703 | n.s. |
| TabDPT | 2.98 | 8 | 0.00781 | 0.0703 | n.s. |
| TabICL | 4.33 | 8 | 0.00781 | 0.0703 | n.s. |
| BETA (TabPFN Unleashed) | 4.12 | 8 | 0.00781 | 0.0703 | n.s. |
| TuneTables | 4.87 | 8 | 0.00781 | 0.0703 | n.s. |
| Lasso | 7.04 | 8 | 0.00781 | 0.0703 | n.s. |
| MLP | 6.70 | 8 | 0.00781 | 0.0703 | n.s. |
| ProtoGate | 7.24 | 8 | 0.00781 | 0.0703 | n.s. |

*Table I.2.* **Average-rank summary on the expanded 16-dataset benchmark.** Lower rank is better. The Friedman test over the 9-method comparison is significant ($\chi^2 = 52.55$, $p = 1.32 \times 10^{-8}$; 11 complete rows).

| Model | Avg. Rank | Std. Rank |
|---|---|---|
| **GOTabPFN** | **1.12** | 0.50 |
| TabPFN-Wide | 3.62 | 1.75 |
| TANDEM | 3.69 | 1.62 |
| TabDPT | 4.34 | 1.49 |
| TuneTables | 5.53 | 2.29 |
| TabICL | 5.69 | 2.02 |
| MLP | 6.16 | 2.43 |
| Lasso | 6.62 | 1.59 |
| ProtoGate | 8.16 | 1.23 |

*Table I.3.* **Omnibus Friedman test on the expanded 16-dataset benchmark.** The test evaluates whether the 9 methods have equal rank distributions across datasets; 11 complete rows were used because Friedman requires all compared methods to be present.

| # Datasets | # Methods | Complete Rows | Friedman $\chi^2$ | $p$-value |
|---|---|---|---|---|
| 16 | 9 | 11 | 52.55 | $1.32 \times 10^{-8}$ |

*Table I.4.* **Pairwise significance of GOTabPFN on the expanded 16-dataset benchmark.** $\Delta$Acc denotes the mean accuracy improvement of GOTabPFN over each baseline in percentage points, averaged over the datasets used for that comparison. W/T/L counts wins/ties/losses in favor of GOTabPFN. We report raw Wilcoxon signed-rank $p$-values and Holm-corrected $p$-values across the 8 pairwise comparisons.

| Baseline | $\Delta$Acc (pp) | $n_{\text{ds}}$ | W/T/L | $p_{\text{raw}}$ | $p_{\text{Holm}}$ | Sig. |
|---|---|---|---|---|---|---|
| TANDEM | 1.33 | 16 | 15/0/1 | 0.000305 | 0.000610 | Yes |
| TabPFN-Wide | 1.76 | 16 | 16/0/0 | 0.000031 | 0.000244 | Yes |
| TabDPT | 2.20 | 16 | 16/0/0 | 0.000031 | 0.000244 | Yes |
| TabICL | 2.81 | 14 | 14/0/0 | 0.000122 | 0.000488 | Yes |
| TuneTables | 4.36 | 16 | 16/0/0 | 0.000031 | 0.000244 | Yes |
| MLP | 4.65 | 16 | 15/0/1 | 0.000153 | 0.000488 | Yes |
| Lasso | 4.78 | 16 | 16/0/0 | 0.000031 | 0.000244 | Yes |
| ProtoGate | 11.90 | 12 | 12/0/0 | 0.000488 | 0.000610 | Yes |

*Table I.5.* **Pairwise significance on the expanded 16-dataset benchmark.** $\Delta$Acc denotes the mean accuracy improvement of GOTabPFN over each baseline in percentage points. We report the number of datasets used ($n_{\text{ds}}$), raw Wilcoxon signed-rank $p$-values, and Holm-corrected $p$-values across the 8 pairwise comparisons. All comparisons remain significant after Holm correction.

| Baseline | $\Delta$Acc (pp) | $n_{\text{ds}}$ | $p_{\text{raw}}$ | $p_{\text{Holm}}$ | Sig. |
|---|---|---|---|---|---|
| TANDEM | 1.33 | 16 | $3.05 \times 10^{-4}$ | $6.10 \times 10^{-4}$ | < 0.01 |
| TabPFN-Wide | 1.76 | 16 | $3.05 \times 10^{-5}$ | $2.44 \times 10^{-4}$ | < 0.01 |
| TabDPT | 2.20 | 16 | $3.05 \times 10^{-5}$ | $2.44 \times 10^{-4}$ | < 0.01 |
| TabICL | 2.81 | 14 | $1.22 \times 10^{-4}$ | $4.88 \times 10^{-4}$ | < 0.01 |
| TuneTables | 4.36 | 16 | $3.05 \times 10^{-5}$ | $2.44 \times 10^{-4}$ | < 0.01 |
| MLP | 4.65 | 16 | $1.53 \times 10^{-4}$ | $4.88 \times 10^{-4}$ | < 0.01 |
| Lasso | 4.78 | 16 | $3.05 \times 10^{-5}$ | $2.44 \times 10^{-4}$ | < 0.01 |
| ProtoGate | 11.90 | 12 | $4.88 \times 10^{-4}$ | $6.10 \times 10^{-4}$ | < 0.01 |

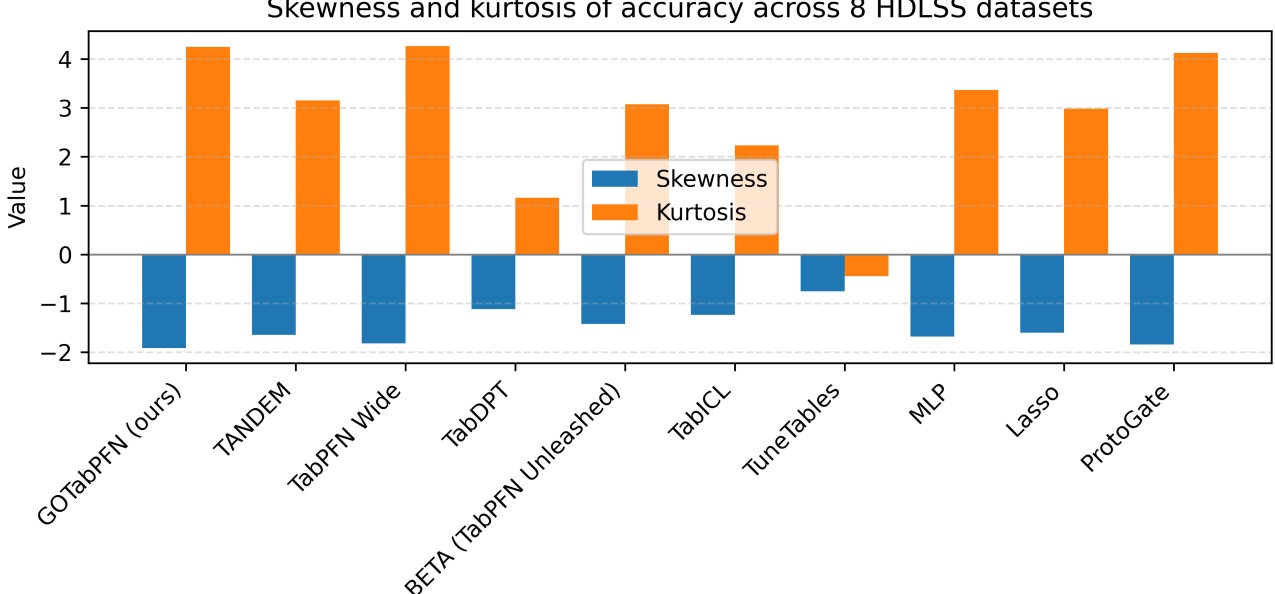

*Figure J.1.* Skewness/kurtosis for the top-10 methods on the 8 HDLSS benchmarks.

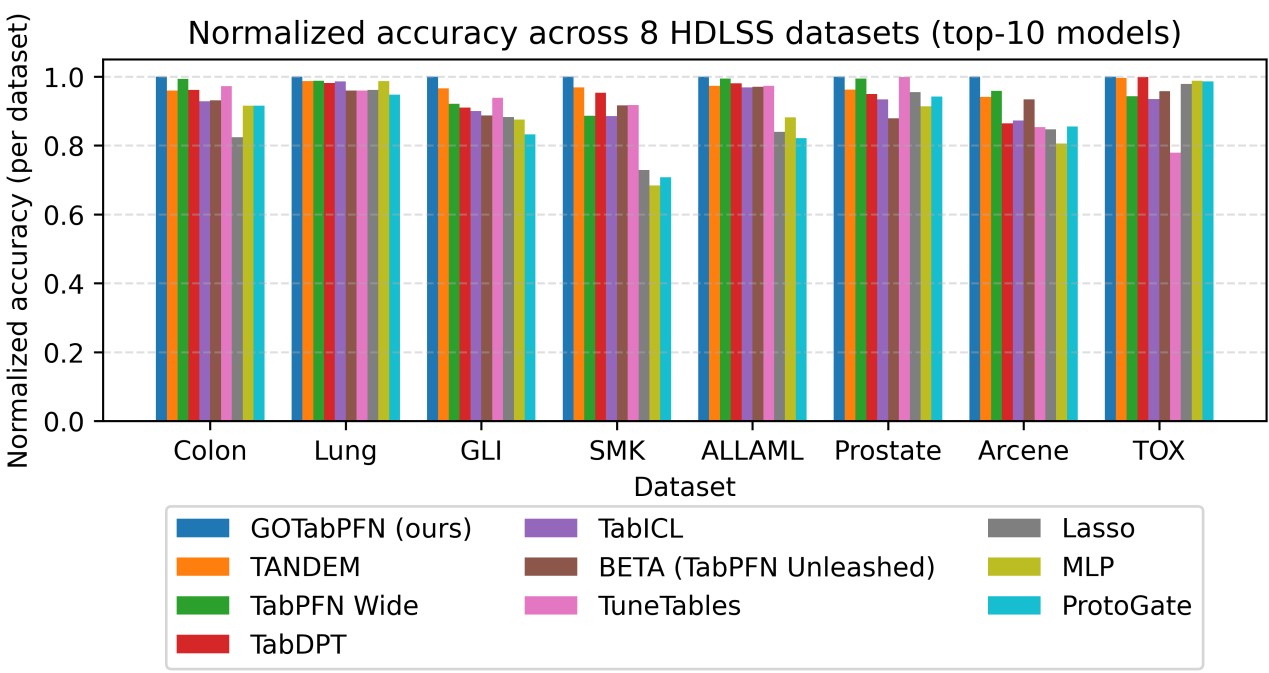

*Figure J.2.* Normalized accuracy for the top-10 methods on the 8 HDLSS benchmarks.

*Table L.1.* GOTabPFN (Colon) inference-time robustness to feature perturbations and latent-space neighborhood consistency. Tab Shuffle randomly permutes a fraction of feature columns across samples; Tab Drop replaces a fraction with the global feature mean. Higher is better for accuracy and kNN agreement.

| Fraction perturbed | Tab Shuffle (%) | Tab Drop/mean (%) | kNN-Agree@5 (%) |
|---|---|---|---|
| 0.00 | 98.39 | 98.39 | 73.87 |
| 0.10 | 96.77 | 100.00 | 73.87 |
| 0.25 | 88.71 | 93.55 | 73.87 |
| 0.50 | 74.19 | 98.39 | 73.87 |
| 0.75 | 62.90 | 61.29 | 73.87 |
| 1.00 | 58.06 | 64.52 | 73.87 |

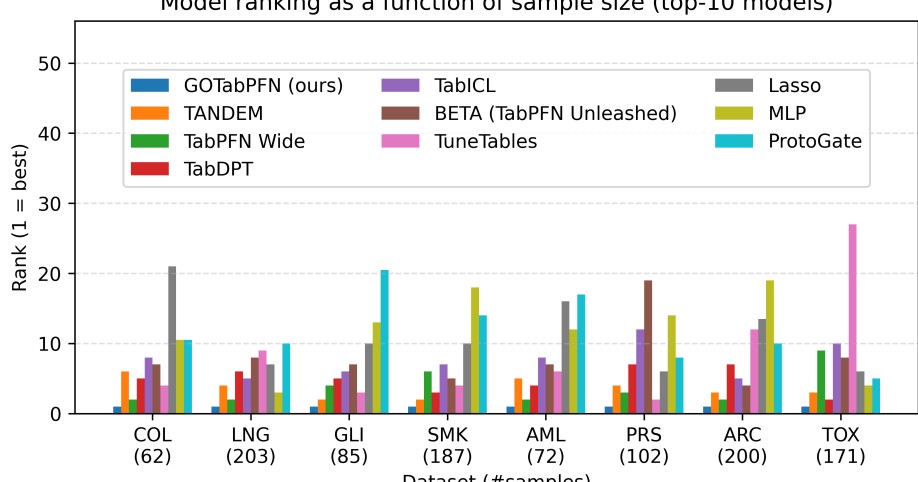

*Figure J.3.* Model rank versus sample size for the top-10 methods on the 8 HDLSS benchmarks.

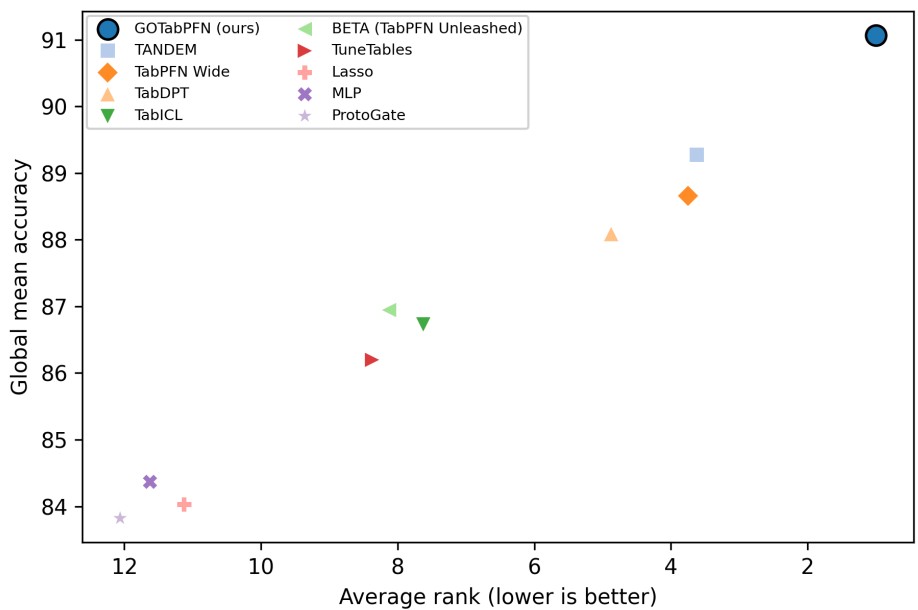

*Figure J.4.* Avg. rank vs. global accuracy.

*Table M.1.* Sanity/stress and stability diagnostics for GOTabPFN on Colon dataset.

| Sanity / stress mode | Accuracy (%) |
|---|---|
| Full input | 98.39 |
| All-zero input | 64.52 |
| Global-mean input | 64.52 |
| Shuffle rows | 54.84 |
| Heavy noise | 95.16 |
| **TTA stability ($n_{\mathrm{aug}}=5$)** | |
| Base accuracy | 98.39 |
| TTA majority-vote accuracy | 98.39 |
| Any label change across aug (%) | 3.23 |
| **Per-class accuracy** | |
| Class 0 (support 40) | 97.5 |
| Class 1 (support 22) | 100.0 |

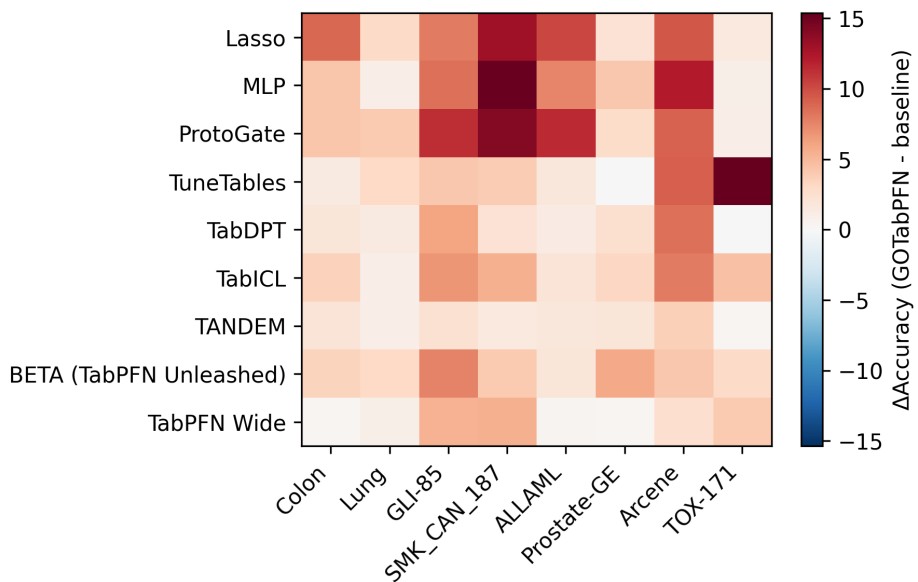

*Figure J.5.* ΔAcc heatmap (ours − baseline).

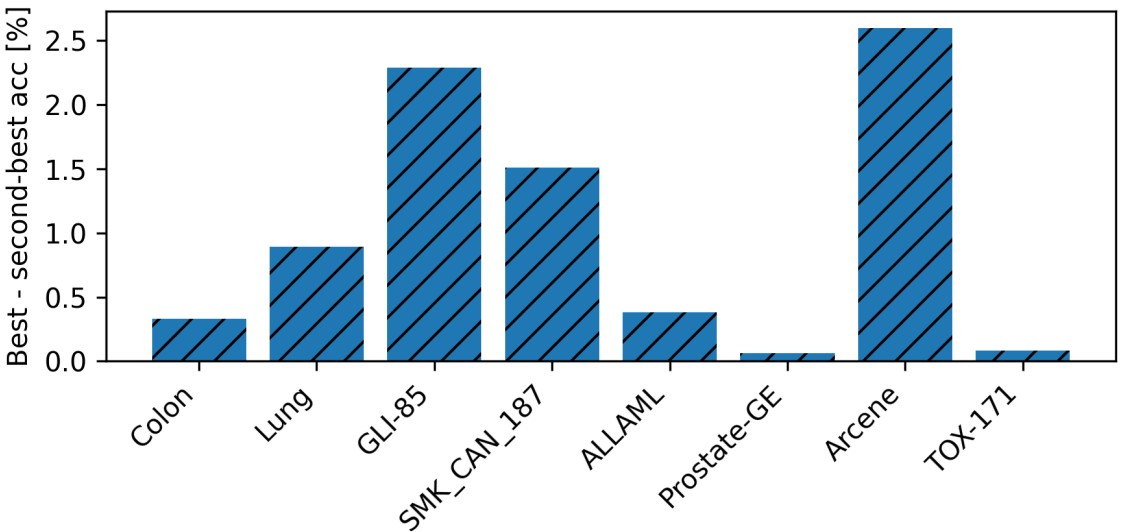

*Figure J.6.* Best-second-best margin across the 8 HDLSS benchmarks.

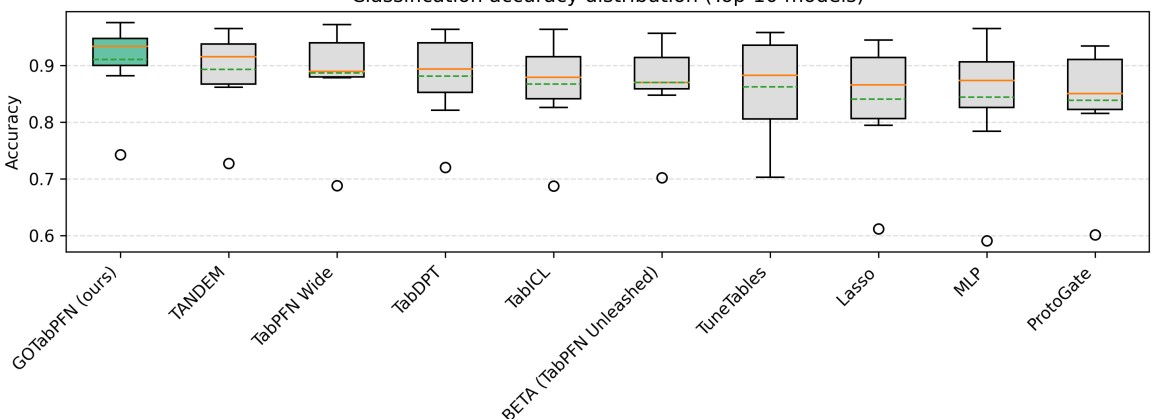

*Figure J.7.* Accuracy distributions across CV splits for the top-10 methods on the 8 HDLSS benchmarks.

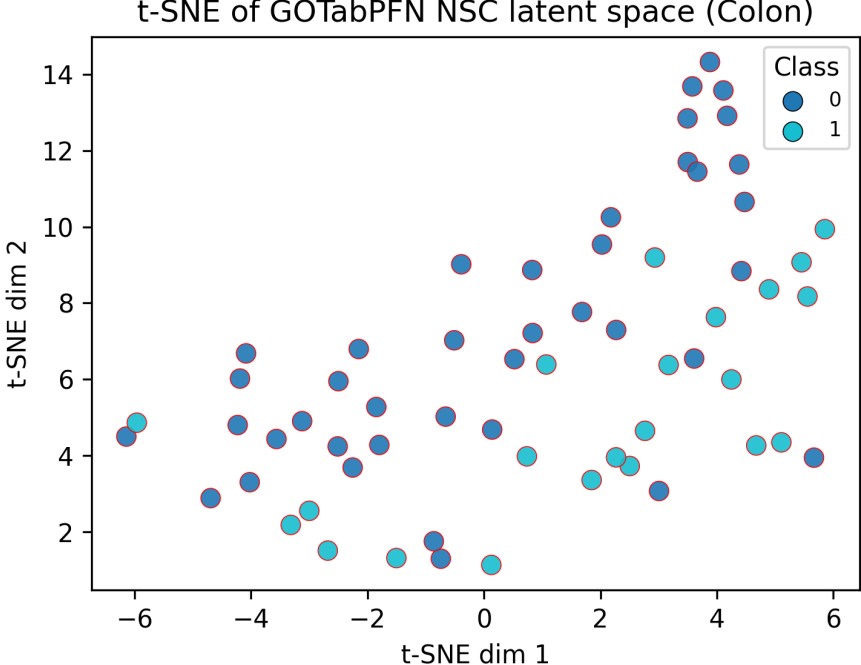

*Figure K.1.* t-SNE visualization of the NSC latent space in GOTabPFN on Colon (colored by class).

*Table R.1.* **GO-LR + NSC as a model-agnostic front-end for TabICL.** We compare GO-LR + NSC + TabICL against vanilla TabICL on 8 HDLSS datasets under $5 \times 5$ CV. Values are mean with subscripted standard deviation for accuracy, ROC-AUC, and macro-F1; runtime is total seconds. Bold denotes the better value between the two methods for each metric/dataset.

| Dataset | Model | Accuracy ↑ | ROC-AUC ↑ | Macro-F1 ↑ | Runtime ↓ |
|---|---|---|---|---|---|
| COL | GO-LR+NSC+TabICL | $\mathbf{86.56}_{\pm 10.77}$ | $\mathbf{89.83}_{\pm 10.36}$ | $\mathbf{81.36}_{\pm 14.87}$ | **382.17** |
| | TabICL | $84.62_{\pm 10.52}$ | $89.38_{\pm 10.03}$ | $78.84_{\pm 13.87}$ | 942.21 |
| LNG | GO-LR+NSC+TabICL | $96.05_{\pm 2.58}$ | $\mathbf{99.62}_{\pm 0.42}$ | $92.17_{\pm 7.57}$ | **382.35** |
| | TabICL | $\mathbf{96.36}_{\pm 2.61}$ | $99.59_{\pm 0.45}$ | $\mathbf{93.72}_{\pm 6.95}$ | 2602.42 |
| GLI | GO-LR+NSC+TabICL | $\mathbf{87.76}_{\pm 7.39}$ | $\mathbf{92.62}_{\pm 6.19}$ | $\mathbf{91.45}_{\pm 5.15}$ | **117.65** |
| | TabICL | $87.06_{\pm 6.23}$ | $91.47_{\pm 6.98}$ | $90.91_{\pm 4.53}$ | 14598.50 |
| SMK | GO-LR+NSC+TabICL | $\mathbf{70.69}_{\pm 5.46}$ | $\mathbf{76.00}_{\pm 6.95}$ | $\mathbf{73.01}_{\pm 5.23}$ | **315.10** |
| | TabICL | $68.73_{\pm 7.28}$ | $73.29_{\pm 7.99}$ | $70.92_{\pm 6.73}$ | 25011.40 |
| AML | GO-LR+NSC+TabICL | $94.78_{\pm 6.35}$ | $\mathbf{98.75}_{\pm 3.49}$ | $92.22_{\pm 9.42}$ | **242.97** |
| | TabICL | $\mathbf{95.52}_{\pm 5.59}$ | $98.27_{\pm 3.96}$ | $\mathbf{92.89}_{\pm 9.10}$ | 2847.26 |
| PRS | GO-LR+NSC+TabICL | $88.25_{\pm 6.94}$ | $93.04_{\pm 6.17}$ | $87.87_{\pm 8.09}$ | **286.36** |
| | TabICL | $\mathbf{90.17}_{\pm 5.93}$ | $\mathbf{93.92}_{\pm 5.45}$ | $\mathbf{90.12}_{\pm 5.80}$ | 2469.72 |
| ARC | GO-LR+NSC+TabICL | $\mathbf{85.20}_{\pm 3.88}$ | $\mathbf{93.10}_{\pm 2.25}$ | $\mathbf{83.21}_{\pm 4.16}$ | **358.59** |
| | TabICL | $82.60_{\pm 6.14}$ | $91.48_{\pm 3.76}$ | $79.61_{\pm 6.26}$ | 7671.89 |
| TOX | GO-LR+NSC+TabICL | $\mathbf{90.87}_{\pm 5.97}$ | $\mathbf{99.08}_{\pm 0.98}$ | $\mathbf{90.95}_{\pm 5.92}$ | **386.01** |
| | TabICL | $88.78_{\pm 5.92}$ | $98.62_{\pm 1.12}$ | $88.89_{\pm 5.79}$ | 2900.00 |

*Table S.1.* **Average accuracy (↑) across the 8 HDLSS datasets for different TabPFN seeds.** Values are mean accuracy with subscripted standard deviation across datasets.

| Model | Seed 42 | Seed 77 | Seed 82 | Seed 93 | Seed 147 |
|---|---|---|---|---|---|
| TabPFN-Wide | $87.51_{\pm 6.04}$ | $88.43_{\pm 4.87}$ | $88.33_{\pm 5.59}$ | $87.59_{\pm 5.60}$ | $87.69_{\pm 5.11}$ |
| BETA | $85.03_{\pm 8.70}$ | $86.10_{\pm 8.33}$ | $85.36_{\pm 8.36}$ | $85.87_{\pm 7.54}$ | $85.70_{\pm 7.65}$ |
| TuneTables | $88.12_{\pm 5.72}$ | $87.90_{\pm 5.65}$ | $88.12_{\pm 5.70}$ | $88.47_{\pm 5.08}$ | $88.20_{\pm 5.21}$ |
| GOTabPFN | $\mathbf{90.57}_{\pm 5.54}$ | $\mathbf{89.66}_{\pm 5.72}$ | $\mathbf{89.93}_{\pm 5.17}$ | $\mathbf{89.79}_{\pm 5.30}$ | $\mathbf{89.66}_{\pm 5.59}$ |

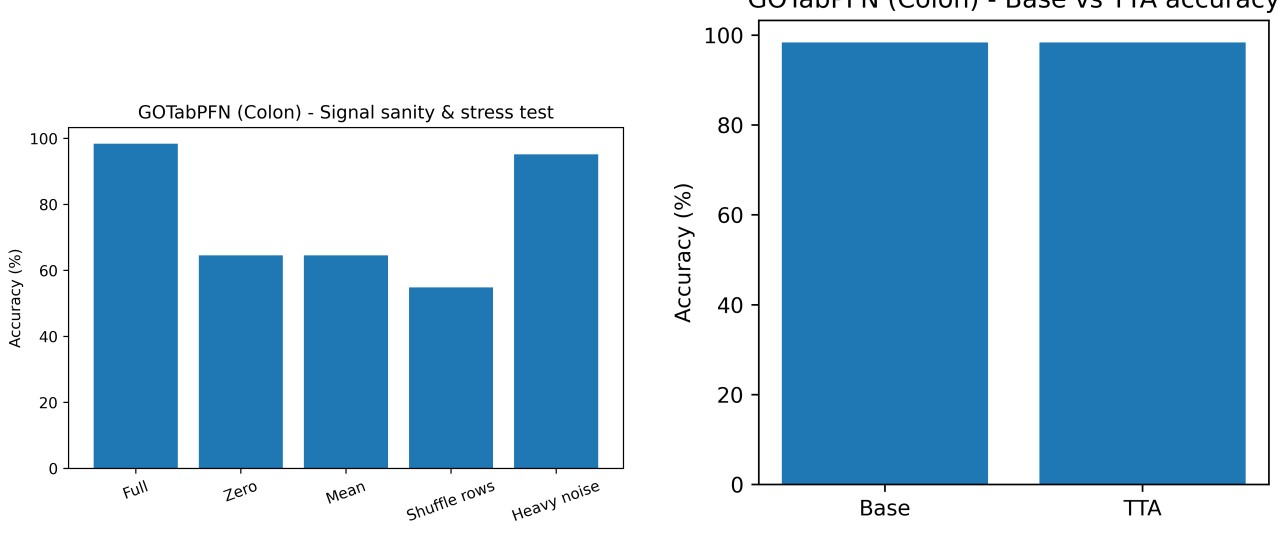

*(a)* Confidence vs. accuracy & coverage.

*(b)* Robustness to perturbations.

*(c)* Reliability diagram (ECE).

*(d)* kNN label agreement.

*Figure L.1.* GOTabPFN (Colon): confidence/coverage, calibration, robustness, and latent-space neighborhood agreement.

*(a)* Signal sanity & stress tests.

*(b)* Base vs. TTA accuracy.

*Figure M.1.* Sanity and stress diagnostics on Colon.

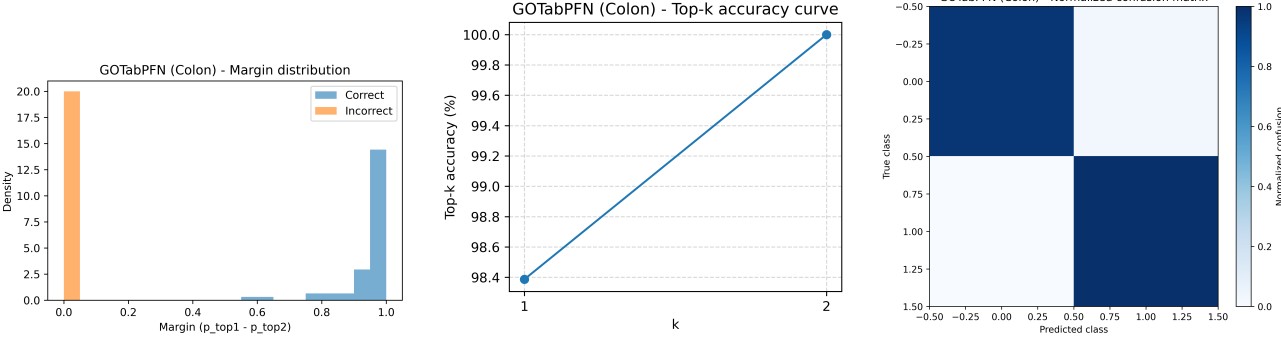

*(a)* Margin distribution ($p_{\text{top1}} - p_{\text{top2}}$).      *(b)* Top-$k$ accuracy curve.      *(c)* Normalized confusion matrix.

*Figure N.1.* Extra reliability diagnostics for GOTabPFN on Colon. We report (left) margin separation between correct vs. incorrect predictions, (middle) Top-$k$ accuracy, and (right) normalized confusion matrix.

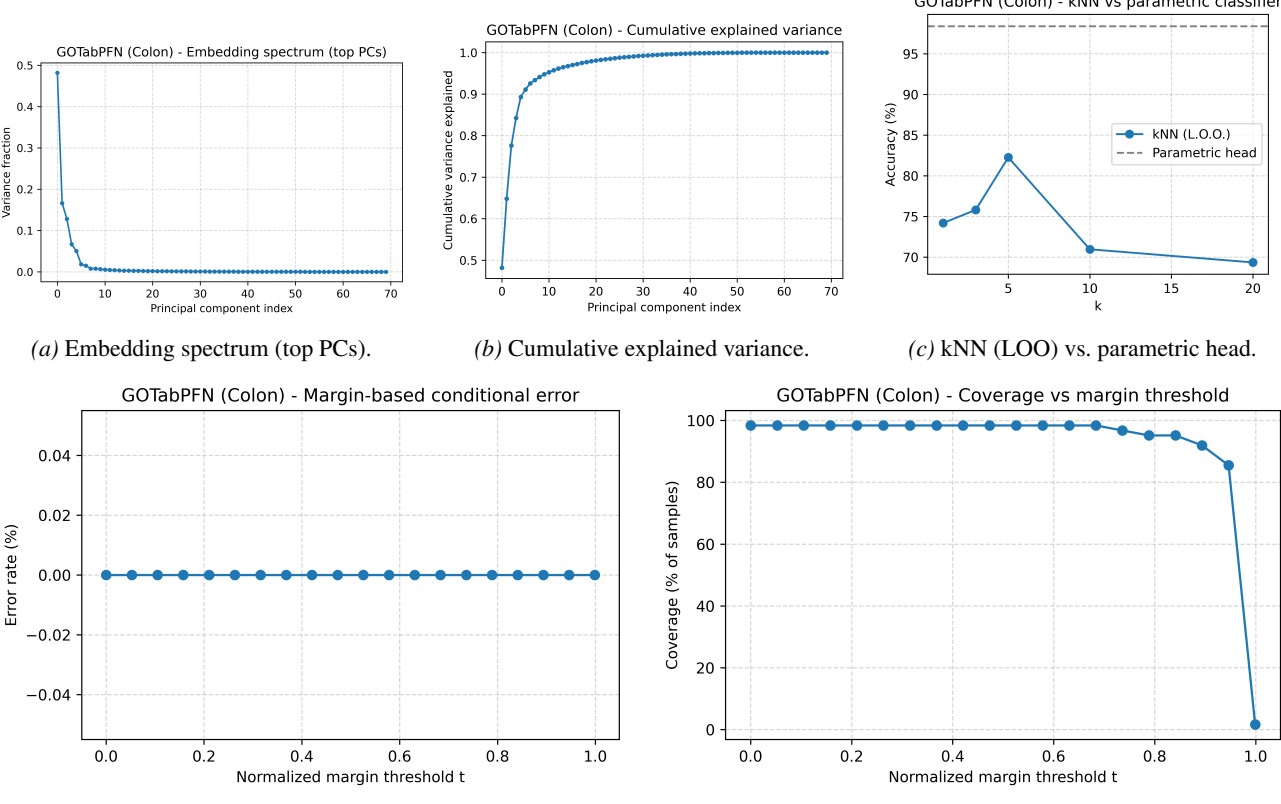

*(a)* Embedding spectrum (top PCs).      *(b)* Cumulative explained variance.      *(c)* kNN (LOO) vs. parametric head.

*(d)* Margin-thresholded conditional error.             *(e)* Coverage vs. margin threshold.

*Figure O.1.* **Theory-inspired representation diagnostics for GOTabPFN (Colon).** (a) Variance spectrum indicates a sharp concentration of energy in the leading PCs. (b) Cumulative explained variance shows rapid saturation. (c) Leave-one-out kNN in embedding space underperforms the parametric head, suggesting performance is not explained by simple local geometry alone. (d-e) Margin-based conditional error and coverage demonstrate high-confidence predictions over most samples.

*Table S.2.* **TabPFN-seed robustness on 8 cross-domain datasets.** Average accuracy across the 8 additional cross-domain datasets for different TabPFN seeds. Values are mean with subscripted standard deviation across datasets. BETA is omitted because it was not run on all datasets and required substantially longer runtime.

| Model | Seed 42 | Seed 77 | Seed 82 | Seed 93 | Seed 147 |
|---|---|---|---|---|---|
| TabPFN-W | $86.60_{\pm 2.95}$ | $84.84_{\pm 3.35}$ | $85.72_{\pm 3.04}$ | $83.71_{\pm 3.24}$ | $86.84_{\pm 3.10}$ |
| TuneTables | $83.65_{\pm 2.10}$ | $82.43_{\pm 2.33}$ | $83.19_{\pm 2.20}$ | $81.68_{\pm 2.28}$ | $83.60_{\pm 2.22}$ |
| GOTabPFN | $\mathbf{86.85}_{\pm 2.55}$ | $\mathbf{86.58}_{\pm 2.85}$ | $\mathbf{86.94}_{\pm 2.50}$ | $\mathbf{86.90}_{\pm 2.55}$ | $\mathbf{86.88}_{\pm 2.66}$ |

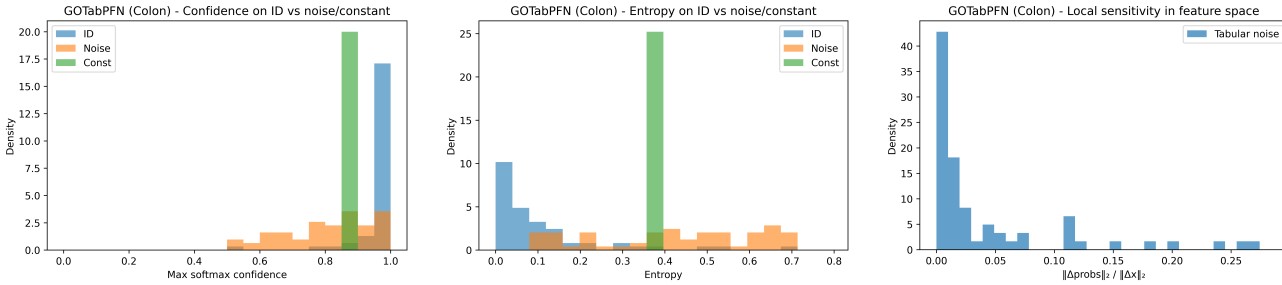

*(a)* Max-softmax confidence: ID vs. Noise/Const.

*(b)* Predictive entropy: ID vs. Noise/Const.

*(c)* Local sensitivity in feature space.

*Figure P.1.* OOD and sensitivity-style diagnostics for GOTabPFN on Colon. Confidence/entropy histograms compare in-distribution (ID) inputs with synthetic OOD tabular perturbations (Noise, Const), while the right panel reports a Lipschitz-like local sensitivity score $\|\Delta\mathrm{probs}\|_2/\|\Delta x\|_2$ under small feature-space noise.

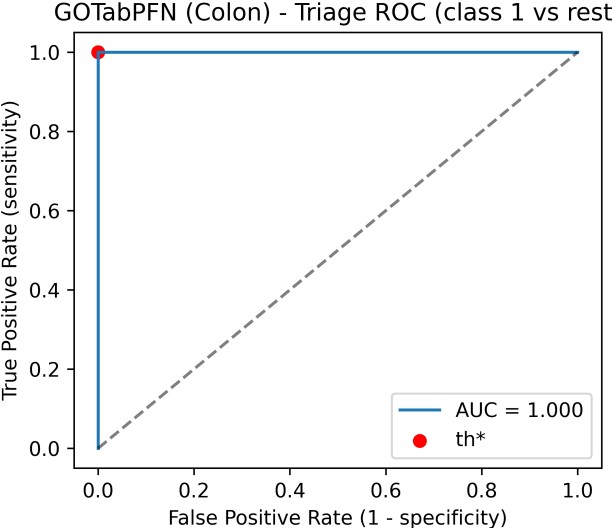

*Figure Q.1.* Deployment-style triage diagnostic on Colon (class 1 vs rest). ROC curve for treating class 1 as the "high-risk" positive class. The marked operating point (th*) corresponds to the selected threshold achieving the target sensitivity criterion, with the best specificity among feasible thresholds.

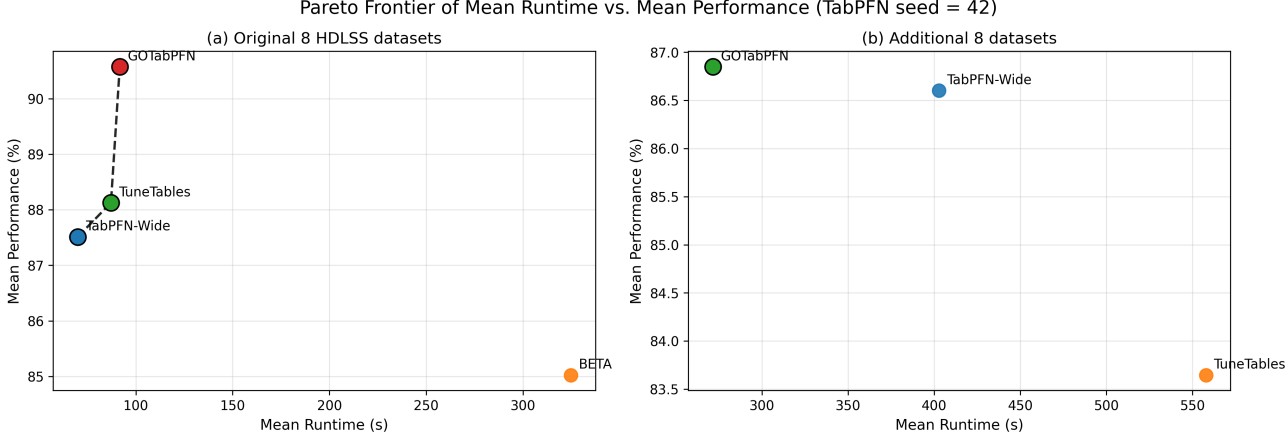

*Figure S.1.* **Pareto frontier of mean runtime vs. mean performance for high-dimensionality compatible TabPFN-family methods.** Results use TabPFN seed 42. Lower runtime and higher performance are better. Dashed lines connect Pareto optimal methods. Left: original 8 HDLSS datasets. Right: additional 8 cross-domain datasets. BETA is excluded from the right panel because it was not run on Cell Cycle and DrivFace-Regression and required substantially longer runtime.

*Table S.3.* **Accuracy comparison across 8 HDLSS datasets for different TabPFN seeds.** Values are mean accuracy with subscripted standard deviation over $5 \times 5$ CV. The Avg. column reports the mean across the 8 datasets, with the subscript giving the mean of the corresponding standard deviations.

| Model | COL | LNG | AML | TOX | PRS | ARC | SMK | GLI | Avg. |
|---|---|---|---|---|---|---|---|---|---|
| **Seed 42** | | | | | | | | | |
| TabPFN-Wide | $87.05_{\pm 7.44}$ | $96.04_{\pm 2.85}$ | $95.90_{\pm 3.74}$ | $84.15_{\pm 6.17}$ | $92.19_{\pm 7.21}$ | $86.00_{\pm 2.85}$ | $70.53_{\pm 9.73}$ | $88.24_{\pm 8.32}$ | $87.51_{\pm 6.04}$ |
| BETA | $84.80_{\pm 13.01}$ | $95.00_{\pm 3.41}$ | $89.40_{\pm 5.82}$ | $83.20_{\pm 12.73}$ | $88.40_{\pm 9.89}$ | $84.50_{\pm 6.20}$ | $69.80_{\pm 10.10}$ | $85.10_{\pm 8.40}$ | $85.03_{\pm 8.70}$ |
| TuneTables | $86.20_{\pm 6.80}$ | $96.30_{\pm 2.50}$ | $94.10_{\pm 5.20}$ | $92.10_{\pm 4.80}$ | $91.10_{\pm 6.30}$ | $85.40_{\pm 3.90}$ | $71.20_{\pm 8.80}$ | $88.60_{\pm 7.50}$ | $88.12_{\pm 5.72}$ |
| GOTabPFN | $\mathbf{88.18}_{\pm 10.05}$ | $\mathbf{97.44}_{\pm 2.32}$ | $\mathbf{97.54}_{\pm 4.31}$ | $\mathbf{92.75}_{\pm 4.07}$ | $91.95_{\pm 7.38}$ | $\mathbf{90.30}_{\pm 3.70}$ | $\mathbf{73.61}_{\pm 5.71}$ | $\mathbf{92.82}_{\pm 6.81}$ | $\mathbf{90.57}_{\pm 5.54}$ |
| **Seed 77** | | | | | | | | | |
| TabPFN-Wide | $85.38_{\pm 7.07}$ | $96.55_{\pm 2.85}$ | $95.81_{\pm 3.83}$ | $90.10_{\pm 5.94}$ | $\mathbf{92.24}_{\pm 5.44}$ | $88.50_{\pm 4.87}$ | $70.60_{\pm 3.10}$ | $88.24_{\pm 5.88}$ | $88.43_{\pm 4.87}$ |
| BETA | $83.00_{\pm 10.37}$ | $94.20_{\pm 2.93}$ | $91.40_{\pm 8.52}$ | $88.00_{\pm 12.46}$ | $87.40_{\pm 9.77}$ | $85.20_{\pm 7.10}$ | $70.10_{\pm 6.30}$ | $89.50_{\pm 9.20}$ | $86.10_{\pm 8.33}$ |
| TuneTables | $84.90_{\pm 8.50}$ | $96.10_{\pm 2.90}$ | $94.80_{\pm 5.40}$ | $91.20_{\pm 4.50}$ | $90.90_{\pm 5.80}$ | $86.40_{\pm 5.10}$ | $70.80_{\pm 5.60}$ | $88.10_{\pm 7.40}$ | $87.90_{\pm 5.65}$ |
| GOTabPFN | $\mathbf{87.21}_{\pm 9.28}$ | $\mathbf{97.33}_{\pm 2.38}$ | $\mathbf{96.99}_{\pm 5.18}$ | $\mathbf{92.04}_{\pm 4.91}$ | $89.39_{\pm 6.94}$ | $\mathbf{90.10}_{\pm 3.57}$ | $\mathbf{74.11}_{\pm 6.37}$ | $\mathbf{90.12}_{\pm 7.15}$ | $\mathbf{89.66}_{\pm 5.72}$ |
| **Seed 82** | | | | | | | | | |
| TabPFN-Wide | $86.92_{\pm 11.33}$ | $97.05_{\pm 3.20}$ | $95.71_{\pm 3.91}$ | $88.89_{\pm 5.25}$ | $\mathbf{91.19}_{\pm 3.99}$ | $88.00_{\pm 3.26}$ | $70.63_{\pm 6.55}$ | $88.24_{\pm 7.20}$ | $88.33_{\pm 5.59}$ |
| BETA | $82.80_{\pm 13.01}$ | $94.80_{\pm 1.83}$ | $90.60_{\pm 9.99}$ | $86.20_{\pm 10.96}$ | $86.40_{\pm 9.48}$ | $84.80_{\pm 6.80}$ | $70.40_{\pm 7.20}$ | $86.90_{\pm 7.60}$ | $85.36_{\pm 8.36}$ |
| TuneTables | $85.10_{\pm 9.40}$ | $96.80_{\pm 3.20}$ | $94.30_{\pm 6.10}$ | $92.80_{\pm 4.90}$ | $90.20_{\pm 4.70}$ | $85.90_{\pm 4.80}$ | $71.50_{\pm 5.90}$ | $88.40_{\pm 6.60}$ | $88.12_{\pm 5.70}$ |
| GOTabPFN | $\mathbf{87.21}_{\pm 9.06}$ | $\mathbf{97.34}_{\pm 2.15}$ | $\mathbf{98.10}_{\pm 3.67}$ | $\mathbf{93.10}_{\pm 4.03}$ | $90.19_{\pm 6.19}$ | $\mathbf{89.60}_{\pm 3.59}$ | $\mathbf{73.58}_{\pm 6.59}$ | $\mathbf{90.35}_{\pm 6.09}$ | $\mathbf{89.93}_{\pm 5.17}$ |
| **Seed 93** | | | | | | | | | |
| TabPFN-Wide | $84.10_{\pm 10.90}$ | $96.06_{\pm 1.34}$ | $97.14_{\pm 3.91}$ | $90.62_{\pm 3.87}$ | $\mathbf{91.16}_{\pm 3.37}$ | $85.50_{\pm 6.22}$ | $70.00_{\pm 11.73}$ | $86.12_{\pm 3.42}$ | $87.59_{\pm 5.60}$ |
| BETA | $82.80_{\pm 13.01}$ | $95.20_{\pm 2.80}$ | $94.00_{\pm 5.10}$ | $86.20_{\pm 9.93}$ | $87.20_{\pm 8.35}$ | $84.90_{\pm 7.00}$ | $70.30_{\pm 8.50}$ | $86.34_{\pm 5.65}$ | $85.87_{\pm 7.54}$ |
| TuneTables | $86.40_{\pm 7.90}$ | $96.30_{\pm 2.10}$ | $95.60_{\pm 4.30}$ | $\mathbf{94.50}_{\pm 3.80}$ | $91.30_{\pm 5.20}$ | $84.80_{\pm 5.10}$ | $71.00_{\pm 6.80}$ | $87.90_{\pm 5.40}$ | $88.47_{\pm 5.08}$ |
| GOTabPFN | $\mathbf{87.85}_{\pm 9.04}$ | $\mathbf{97.34}_{\pm 2.58}$ | $\mathbf{97.52}_{\pm 3.88}$ | $93.57_{\pm 4.38}$ | $90.16_{\pm 6.69}$ | $\mathbf{89.80}_{\pm 3.88}$ | $\mathbf{72.42}_{\pm 5.73}$ | $\mathbf{89.65}_{\pm 6.19}$ | $\mathbf{89.79}_{\pm 5.30}$ |
| **Seed 147** | | | | | | | | | |
| TabPFN-Wide | $83.97_{\pm 5.25}$ | $96.56_{\pm 2.20}$ | $95.81_{\pm 3.83}$ | $87.71_{\pm 4.38}$ | $\mathbf{91.34}_{\pm 3.82}$ | $89.50_{\pm 2.09}$ | $68.42_{\pm 12.07}$ | $88.24_{\pm 7.20}$ | $87.69_{\pm 5.11}$ |
| BETA | $80.20_{\pm 11.60}$ | $95.00_{\pm 3.41}$ | $93.00_{\pm 4.43}$ | $86.54_{\pm 10.30}$ | $88.60_{\pm 8.05}$ | $85.10_{\pm 6.50}$ | $69.90_{\pm 9.80}$ | $87.30_{\pm 7.10}$ | $85.70_{\pm 7.65}$ |
| TuneTables | $83.20_{\pm 5.90}$ | $96.20_{\pm 2.30}$ | $94.40_{\pm 5.60}$ | $\mathbf{93.80}_{\pm 3.90}$ | $90.80_{\pm 4.20}$ | $86.70_{\pm 4.40}$ | $71.60_{\pm 9.30}$ | $88.90_{\pm 6.10}$ | $88.20_{\pm 5.21}$ |
| GOTabPFN | $\mathbf{88.49}_{\pm 9.60}$ | $\mathbf{97.14}_{\pm 2.12}$ | $\mathbf{97.28}_{\pm 4.77}$ | $92.28_{\pm 4.85}$ | $89.18_{\pm 6.98}$ | $\mathbf{90.10}_{\pm 3.78}$ | $\mathbf{72.94}_{\pm 6.36}$ | $\mathbf{89.88}_{\pm 6.24}$ | $\mathbf{89.66}_{\pm 5.59}$ |

*Table S.4.* **Accuracy comparison across 8 cross-domain datasets for different TabPFN seeds.** Values are mean accuracy with subscripted standard deviation over $5 \times 5$ CV. The Avg. column reports the mean across datasets within each seed block, with the subscript giving the mean of the corresponding standard deviations. BETA is reported only for seed 42 and its average is computed over 6 datasets because Cell Cycle and DrivFace-Regression were omitted due to substantially longer runtime.

| Model | ORL | BAS | REL | PCM | CCY | CIF | DF-R | DF-C | Avg. |
|---|---|---|---|---|---|---|---|---|---|
| **Seed 42** | | | | | | | | | |
| TabPFN-W | $95.80_{\pm 6.30}$ | $96.70_{\pm 0.70}$ | $\mathbf{89.20}_{\pm 3.60}$ | $88.65_{\pm 0.60}$ | $77.90_{\pm 2.30}$ | $88.20_{\pm 0.60}$ | $\mathbf{70.10}_{\pm 7.00}$ | $86.27_{\pm 2.50}$ | $86.60_{\pm 2.95}$ |
| TuneTables | $92.40_{\pm 1.10}$ | $92.90_{\pm 1.00}$ | $89.10_{\pm 2.10}$ | $83.87_{\pm 2.80}$ | $77.50_{\pm 1.20}$ | $78.20_{\pm 0.20}$ | $68.90_{\pm 6.20}$ | $86.30_{\pm 2.20}$ | $83.65_{\pm 2.10}$ |
| BETA | $93.00_{\pm 5.10}$ | $96.80_{\pm 1.17}$ | $\mathbf{89.20}_{\pm 2.04}$ | $89.13_{\pm 1.94}$ | - | $88.20_{\pm 0.75}$ | - | $80.60_{\pm 1.36}$ | $89.49_{\pm 2.06}†$ |
| GOTabPFN | $\mathbf{100.00}_{\pm 0.00}$ | $\mathbf{97.04}_{\pm 1.11}$ | $88.79_{\pm 1.37}$ | $\mathbf{89.28}_{\pm 2.25}$ | $\mathbf{79.34}_{\pm 2.68}$ | $\mathbf{88.50}_{\pm 0.93}$ | $65.51_{\pm 9.83}$ | $86.34_{\pm 2.22}$ | $\mathbf{86.85}_{\pm 2.55}$ |
| **Seed 77** | | | | | | | | | |
| TabPFN-W | $96.30_{\pm 6.90}$ | $96.50_{\pm 0.80}$ | $85.60_{\pm 4.10}$ | $89.30_{\pm 0.80}$ | $\mathbf{79.40}_{\pm 2.60}$ | $\mathbf{87.50}_{\pm 0.50}$ | $60.20_{\pm 8.20}$ | $83.90_{\pm 2.90}$ | $84.84_{\pm 3.35}$ |
| TuneTables | $94.80_{\pm 1.00}$ | $93.80_{\pm 1.10}$ | $86.20_{\pm 2.50}$ | $86.90_{\pm 3.20}$ | $79.20_{\pm 1.00}$ | $77.60_{\pm 0.10}$ | $58.20_{\pm 7.10}$ | $82.70_{\pm 2.60}$ | $82.43_{\pm 2.33}$ |
| GOTabPFN | $\mathbf{100.00}_{\pm 0.00}$ | $\mathbf{97.05}_{\pm 1.13}$ | $\mathbf{88.56}_{\pm 1.60}$ | $\mathbf{89.52}_{\pm 2.35}$ | $79.57_{\pm 2.85}$ | $86.27_{\pm 2.39}$ | $\mathbf{65.39}_{\pm 10.08}$ | $\mathbf{86.27}_{\pm 2.39}$ | $\mathbf{86.58}_{\pm 2.85}$ |
| **Seed 82** | | | | | | | | | |
| TabPFN-W | $94.60_{\pm 6.40}$ | $\mathbf{97.80}_{\pm 0.70}$ | $88.40_{\pm 3.80}$ | $87.70_{\pm 0.70}$ | $76.80_{\pm 2.20}$ | $\mathbf{88.90}_{\pm 0.60}$ | $65.50_{\pm 7.50}$ | $86.10_{\pm 2.40}$ | $85.72_{\pm 3.04}$ |
| TuneTables | $93.10_{\pm 1.10}$ | $93.20_{\pm 1.00}$ | $88.70_{\pm 2.20}$ | $84.50_{\pm 3.00}$ | $78.40_{\pm 1.20}$ | $78.40_{\pm 0.20}$ | $64.10_{\pm 6.60}$ | $85.10_{\pm 2.30}$ | $83.19_{\pm 2.20}$ |
| GOTabPFN | $\mathbf{100.00}_{\pm 0.00}$ | $96.99_{\pm 1.01}$ | $\mathbf{88.80}_{\pm 1.62}$ | $\mathbf{89.63}_{\pm 2.05}$ | $\mathbf{79.32}_{\pm 2.34}$ | $88.66_{\pm 0.81}$ | $\mathbf{65.66}_{\pm 9.60}$ | $\mathbf{86.44}_{\pm 2.53}$ | $\mathbf{86.94}_{\pm 2.50}$ |
| **Seed 93** | | | | | | | | | |
| TabPFN-W | $90.20_{\pm 6.80}$ | $96.90_{\pm 0.60}$ | $84.90_{\pm 4.00}$ | $88.80_{\pm 0.70}$ | $\mathbf{81.30}_{\pm 2.50}$ | $87.80_{\pm 0.50}$ | $57.10_{\pm 8.00}$ | $82.70_{\pm 2.80}$ | $83.71_{\pm 3.24}$ |
| TuneTables | $91.90_{\pm 1.00}$ | $92.70_{\pm 1.10}$ | $87.30_{\pm 2.40}$ | $82.80_{\pm 3.10}$ | $76.90_{\pm 1.10}$ | $77.80_{\pm 0.10}$ | $60.10_{\pm 6.90}$ | $83.90_{\pm 2.50}$ | $81.68_{\pm 2.28}$ |
| GOTabPFN | $\mathbf{100.00}_{\pm 0.00}$ | $\mathbf{97.09}_{\pm 1.02}$ | $\mathbf{88.63}_{\pm 1.41}$ | $\mathbf{89.52}_{\pm 2.02}$ | $79.44_{\pm 2.67}$ | $\mathbf{88.38}_{\pm 0.89}$ | $\mathbf{65.62}_{\pm 9.69}$ | $\mathbf{86.54}_{\pm 2.70}$ | $\mathbf{86.90}_{\pm 2.55}$ |
| **Seed 147** | | | | | | | | | |
| TabPFN-W | $97.60_{\pm 6.20}$ | $96.70_{\pm 0.70}$ | $\mathbf{91.10}_{\pm 3.70}$ | $88.90_{\pm 0.70}$ | $77.90_{\pm 2.40}$ | $87.80_{\pm 0.60}$ | $\mathbf{68.60}_{\pm 7.90}$ | $86.10_{\pm 2.60}$ | $86.84_{\pm 3.10}$ |
| TuneTables | $93.50_{\pm 1.00}$ | $93.10_{\pm 1.00}$ | $90.00_{\pm 2.30}$ | $84.10_{\pm 2.90}$ | $79.50_{\pm 1.20}$ | $78.30_{\pm 0.20}$ | $65.30_{\pm 6.80}$ | $85.00_{\pm 2.40}$ | $83.60_{\pm 2.22}$ |
| GOTabPFN | $\mathbf{99.80}_{\pm 1.00}$ | $\mathbf{97.08}_{\pm 1.08}$ | $88.75_{\pm 1.78}$ | $\mathbf{89.31}_{\pm 2.12}$ | $\mathbf{79.64}_{\pm 2.58}$ | $\mathbf{88.34}_{\pm 0.74}$ | $65.55_{\pm 9.69}$ | $\mathbf{86.54}_{\pm 2.33}$ | $\mathbf{86.88}_{\pm 2.66}$ |

† For BETA at seed 42, Cell Cycle and DrivFace-Regression are unavailable because the runs were prohibitively time-consuming; its average is computed over the remaining 6 datasets.

*Table S.5.* **ROC-AUC comparison across 8 HDLSS datasets for different TabPFN seeds.** Values are mean ROC-AUC with subscripted standard deviation over $5 \times 5$ CV. The Avg. column reports the mean across the 8 datasets, with the subscript giving the mean of the corresponding standard deviations.

| Model | COL | LNG | AML | TOX | PRS | ARC | SMK | GLI | Avg. |
|---|---|---|---|---|---|---|---|---|---|
| **Seed 42** | | | | | | | | | |
| TabPFN-Wide | $88.25_{\pm 6.27}$ | $99.49_{\pm 0.55}$ | $98.40_{\pm 3.58}$ | $97.72_{\pm 1.77}$ | $\mathbf{96.07}_{\pm 5.63}$ | $93.56_{\pm 2.63}$ | $77.97_{\pm 8.85}$ | $\mathbf{94.76}_{\pm 4.93}$ | $93.28_{\pm 4.28}$ |
| BETA | $\mathbf{92.60}_{\pm 5.54}$ | $99.00_{\pm 1.10}$ | $94.40_{\pm 3.38}$ | $96.60_{\pm 2.94}$ | $95.20_{\pm 4.90}$ | $91.80_{\pm 3.40}$ | $76.10_{\pm 8.20}$ | $93.40_{\pm 6.10}$ | $92.39_{\pm 4.44}$ |
| TuneTables | $90.10_{\pm 6.20}$ | $99.10_{\pm 0.80}$ | $97.20_{\pm 3.10}$ | $98.80_{\pm 1.20}$ | $95.80_{\pm 4.90}$ | $92.10_{\pm 3.10}$ | $76.80_{\pm 8.00}$ | $94.20_{\pm 6.20}$ | $93.01_{\pm 4.19}$ |
| GOTabPFN | $91.40_{\pm 10.12}$ | $\mathbf{99.58}_{\pm 0.60}$ | $\mathbf{99.44}_{\pm 1.47}$ | $\mathbf{99.23}_{\pm 0.63}$ | $94.32_{\pm 5.39}$ | $\mathbf{95.85}_{\pm 2.29}$ | $\mathbf{80.92}_{\pm 5.63}$ | $90.97_{\pm 7.09}$ | $\mathbf{93.96}_{\pm 4.15}$ |
| **Seed 77** | | | | | | | | | |
| TabPFN-Wide | $86.75_{\pm 11.31}$ | $\mathbf{99.84}_{\pm 0.27}$ | $\mathbf{99.60}_{\pm 0.89}$ | $98.21_{\pm 1.89}$ | $\mathbf{97.13}_{\pm 3.14}$ | $95.08_{\pm 4.13}$ | $76.03_{\pm 7.48}$ | $91.39_{\pm 8.83}$ | $93.00_{\pm 4.74}$ |
| BETA | $89.80_{\pm 7.49}$ | $99.40_{\pm 0.49}$ | $99.20_{\pm 1.60}$ | $96.80_{\pm 2.71}$ | $94.80_{\pm 4.70}$ | $91.20_{\pm 4.60}$ | $75.20_{\pm 7.50}$ | $\mathbf{92.30}_{\pm 7.90}$ | $92.34_{\pm 4.62}$ |
| TuneTables | $89.20_{\pm 9.80}$ | $99.60_{\pm 0.60}$ | $99.10_{\pm 1.40}$ | $98.90_{\pm 1.50}$ | $95.90_{\pm 4.20}$ | $92.80_{\pm 4.10}$ | $75.90_{\pm 7.30}$ | $92.60_{\pm 7.40}$ | $93.00_{\pm 4.54}$ |
| GOTabPFN | $\mathbf{91.75}_{\pm 9.86}$ | $99.65_{\pm 0.50}$ | $99.12_{\pm 2.39}$ | $\mathbf{99.18}_{\pm 0.72}$ | $93.93_{\pm 5.13}$ | $\mathbf{95.82}_{\pm 2.26}$ | $\mathbf{81.03}_{\pm 5.66}$ | $91.07_{\pm 6.94}$ | $\mathbf{93.94}_{\pm 4.18}$ |
| **Seed 82** | | | | | | | | | |
| TabPFN-Wide | $83.38_{\pm 15.27}$ | $\mathbf{99.75}_{\pm 0.29}$ | $\mathbf{99.56}_{\pm 0.99}$ | $97.83_{\pm 1.64}$ | $\mathbf{95.78}_{\pm 4.29}$ | $96.61_{\pm 2.55}$ | $78.74_{\pm 4.11}$ | $\mathbf{95.21}_{\pm 6.63}$ | $93.36_{\pm 4.47}$ |
| BETA | $90.40_{\pm 6.62}$ | $99.20_{\pm 0.75}$ | $95.80_{\pm 5.15}$ | $96.80_{\pm 2.71}$ | $95.10_{\pm 3.90}$ | $92.00_{\pm 2.80}$ | $78.20_{\pm 4.80}$ | $92.80_{\pm 6.20}$ | $92.54_{\pm 4.12}$ |
| TuneTables | $88.90_{\pm 7.20}$ | $99.70_{\pm 0.40}$ | $98.90_{\pm 1.80}$ | $99.10_{\pm 1.10}$ | $95.60_{\pm 3.50}$ | $93.40_{\pm 2.70}$ | $\mathbf{79.10}_{\pm 4.50}$ | $93.40_{\pm 6.40}$ | $93.51_{\pm 3.45}$ |
| GOTabPFN | $\mathbf{91.03}_{\pm 10.30}$ | $99.51_{\pm 0.84}$ | $99.04_{\pm 2.52}$ | $\mathbf{99.32}_{\pm 0.68}$ | $94.20_{\pm 5.32}$ | $95.84_{\pm 2.22}$ | $79.06_{\pm 5.57}$ | $90.47_{\pm 7.82}$ | $\mathbf{93.56}_{\pm 4.41}$ |
| **Seed 93** | | | | | | | | | |
| TabPFN-Wide | $\mathbf{91.13}_{\pm 10.24}$ | $\mathbf{99.65}_{\pm 0.46}$ | $\mathbf{99.56}_{\pm 0.99}$ | $97.85_{\pm 1.70}$ | $\mathbf{97.85}_{\pm 3.48}$ | $93.86_{\pm 5.30}$ | $76.68_{\pm 5.77}$ | $\mathbf{93.33}_{\pm 7.36}$ | $\mathbf{93.74}_{\pm 4.41}$ |
| BETA | $90.40_{\pm 8.01}$ | $99.20_{\pm 0.75}$ | $96.40_{\pm 2.65}$ | $96.00_{\pm 2.61}$ | $95.30_{\pm 4.10}$ | $91.00_{\pm 3.90}$ | $75.50_{\pm 6.90}$ | $92.50_{\pm 7.20}$ | $92.04_{\pm 4.52}$ |
| TuneTables | $90.50_{\pm 10.20}$ | $99.50_{\pm 0.50}$ | $98.90_{\pm 1.60}$ | $99.00_{\pm 1.30}$ | $96.20_{\pm 4.00}$ | $92.40_{\pm 3.50}$ | $75.80_{\pm 6.40}$ | $92.90_{\pm 7.00}$ | $93.15_{\pm 4.31}$ |
| GOTabPFN | $90.68_{\pm 10.34}$ | $99.61_{\pm 0.60}$ | $99.28_{\pm 2.07}$ | $\mathbf{99.24}_{\pm 0.76}$ | $94.28_{\pm 5.41}$ | $\mathbf{95.67}_{\pm 2.47}$ | $\mathbf{79.93}_{\pm 5.40}$ | $90.98_{\pm 6.88}$ | $93.71_{\pm 4.24}$ |
| **Seed 147** | | | | | | | | | |
| TabPFN-Wide | $83.87_{\pm 11.80}$ | $99.44_{\pm 0.42}$ | $\mathbf{99.56}_{\pm 0.99}$ | $98.23_{\pm 1.44}$ | $\mathbf{96.42}_{\pm 3.44}$ | $96.11_{\pm 1.51}$ | $79.18_{\pm 12.89}$ | $\mathbf{95.36}_{\pm 5.21}$ | $93.52_{\pm 4.71}$ |
| BETA | $90.20_{\pm 6.01}$ | $99.40_{\pm 0.49}$ | $97.80_{\pm 4.40}$ | $96.80_{\pm 3.12}$ | $95.60_{\pm 3.80}$ | $92.40_{\pm 3.10}$ | $78.50_{\pm 12.10}$ | $93.20_{\pm 5.90}$ | $92.99_{\pm 4.86}$ |
| TuneTables | $88.10_{\pm 11.40}$ | $99.30_{\pm 0.60}$ | $98.70_{\pm 2.20}$ | $\mathbf{99.20}_{\pm 1.10}$ | $95.90_{\pm 3.60}$ | $93.10_{\pm 3.20}$ | $78.90_{\pm 11.80}$ | $93.80_{\pm 5.50}$ | $93.38_{\pm 4.93}$ |
| GOTabPFN | $\mathbf{91.23}_{\pm 10.75}$ | $\mathbf{99.57}_{\pm 0.56}$ | $99.20_{\pm 2.24}$ | $99.08_{\pm 0.99}$ | $93.88_{\pm 5.35}$ | $95.55_{\pm 2.47}$ | $79.64_{\pm 5.94}$ | $90.92_{\pm 7.02}$ | $\mathbf{93.63}_{\pm 4.42}$ |

*Table S.6.* **TabPFN inference-seed sensitivity of GOTabPFN on Colon.** GO-LR, NSC, preprocessing, and the exact same $5 \times 5$ CV splits are fixed; only the TabPFN inference seed is varied. Values are mean accuracy with subscripted standard deviation over $5 \times 5$ CV.

| Seed | 0 | 1 | 2 | 3 | 4 | 7 | 11 | 17 | 23 | 42 |
|---|---|---|---|---|---|---|---|---|---|---|
| GOTabPFN | $87.51_{\pm 8.72}$ | $86.56_{\pm 9.25}$ | $87.18_{\pm 8.72}$ | $86.90_{\pm 8.34}$ | $86.87_{\pm 8.71}$ | $86.85_{\pm 8.36}$ | $87.15_{\pm 9.34}$ | $87.51_{\pm 9.36}$ | $87.21_{\pm 8.73}$ | $\mathbf{88.18}_{\pm 10.05}$ |
| Across seeds | Mean = 87.19, std. = 0.46, range = 86.56-88.18 (1.62 pp) | | | | | | | | | |

*Table T.1.* **Additional cross-domain datasets.** Datasets are categorized using the empirical regime rule in Appendix F, following DynaTab (Habib et al., 2026b). Here, $n$ is the number of instances, $m$ is the number of features, and $\rho = m/n$ is the feature-to-sample ratio. CIFAR-10 uses subsampled ResNet-50 image embeddings.

| Dataset | Source | Type | $n$ | $m$ | #Classes | $\rho$ | Category |
|---|---|---|---|---|---|---|---|
| BASEHOCK | scikit-feature (Li et al., 2018)[1] | Text (bag-of-words) | 1993 | 4862 | 2 | 2.44 | MixedRegime |
| PCMAC | scikit-feature (Li et al., 2018)[1] | Text (bag-of-words) | 1943 | 3289 | 2 | 1.69 | MixedRegime |
| RELATHE | scikit-feature (Li et al., 2018)[1] | Text (bag-of-words) | 1427 | 4322 | 2 | 3.03 | MixedRegime |
| orlraws10P | scikit-feature (Li et al., 2018)[1] | Face image | 100 | 10304 | 10 | 103.04 | HDLSS |
| DrivFace-C | UCI (Hernández-Sabat et al., 2016)[2] | Camera sensor | 606 | 6400 | 7 | 10.56 | HDLSS |
| DrivFace-R | UCI (Hernández-Sabat et al., 2016)[2] | Camera sensor | 606 | 6400 | Reg. | 10.56 | HDLSS |
| CIFAR-10 | Kaggle (Cukierski, 2013)[3] | Image features | 11000 | 2048 | 10 | 0.19 | HDHSS |
| Cell Cycle | NCBI (Mahdessian et al., 2021)[4] | RNA-seq | 1067 | 42728 | 3 | 40.05 | MixedRegime |

[1] scikit-feature dataset repository: https://jundongl.github.io/scikit-feature/datasets.html. [2] DrivFace dataset: https://archive.ics.uci.edu/dataset/378/drivface. [3] Kaggle CIFAR-10 dataset: https://www.kaggle.com/competitions/cifar-10. [4] Cell Cycle GEO accession: https://www.ncbi.nlm.nih.gov/geo/query/acc.cgi?acc=GSE146773. **Categorization rule.** A dataset is labeled **HDLSS** if $m > 1000$, $n < 1000$, and $\rho > 2$; **HDHSS** if $m > 1000$, $n > 10^4$, and $0.005 < \rho \le 2$; **LDHSS** if $m \le 100$, $n > 10^4$, and $\rho \le 0.01$; **LDLSS** if $m \le 100$, $n \le 1000$, and $\rho \le 0.05$; and **MixedRegime** otherwise.

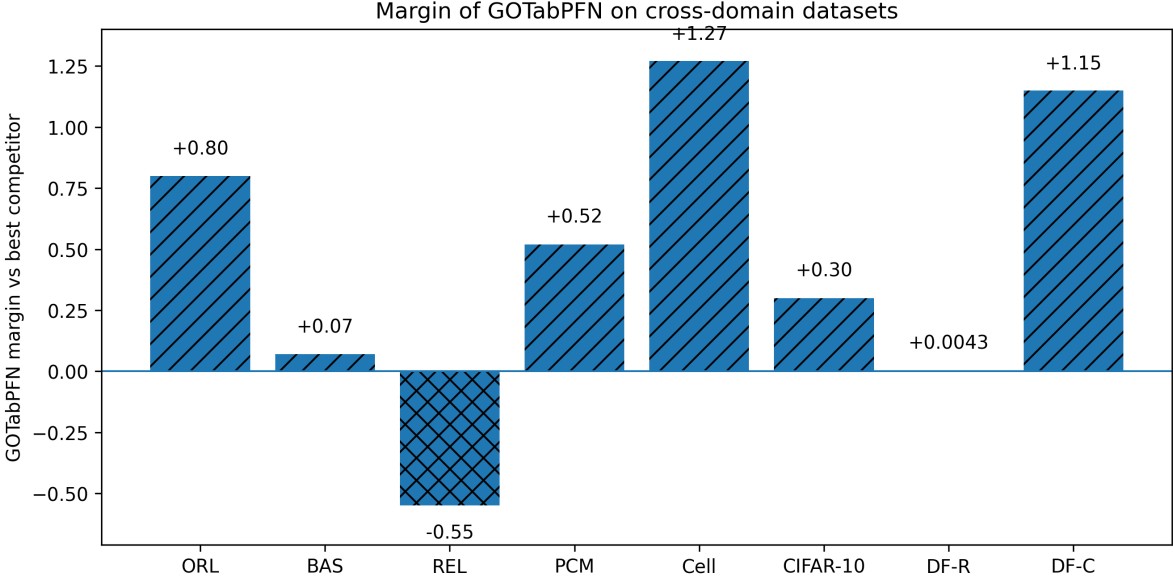

*Figure T.1.* **Signed margin of GOTabPFN on cross-domain datasets.** Positive bars show GOTabPFN's margin over the runner-up when it ranks first; the negative bar shows how far it trails the best baseline when it does not rank first. GOTabPFN ranks first on 7/8 additional datasets, with especially strong margins on ORL, Cell Cycle, and DrivFace-C, and remains competitive on RELATHE.

*Table T.2.* List of 55 baseline models and their source URLs.

| Model | Source URL | Model | Source URL |
|---|---|---|---|
| Naive Bayes | https://scikit-learn.org/stable/supervised_learning.html | AutoInt | https://github.com/OpenTabular/DeepTab |
| KNN | https://scikit-learn.org/stable/supervised_learning.html | TabR | https://github.com/OpenTabular/DeepTab |
| SVM | https://scikit-learn.org/stable/supervised_learning.html | ProtoGate | https://github.com/SilenceX12138/ProtoGate |
| Decision Tree | https://scikit-learn.org/stable/supervised_learning.html | LSPIN | https://github.com/jcyang34/lspin |
| Lasso | https://scikit-learn.org/stable/supervised_learning.html | LLSPIN | https://github.com/jcyang34/lspin |
| MLP | https://scikit-learn.org/stable/supervised_learning.html | INVASE | https://github.com/vanderschaarlab/INVASE |
| 1-D CNN | https://github.com/harryjdavies/Python1D_CNNs | L2X | https://github.com/Jianbo-Lab/L2X |
| Random Forest | https://scikit-learn.org/stable/supervised_learning.html | Mambular | https://github.com/OpenTabular/DeepTab |
| AdaBoost | https://scikit-learn.org/stable/supervised_learning.html | DANets | https://github.com/manujosephv/pytorch_tabular |
| GBM | https://scikit-learn.org/stable/supervised_learning.html | STG | https://github.com/runopti/stg |
| LGBM | https://github.com/microsoft/LightGBM | REAL-X | https://github.com/rajesh-lab/realx |
| XGBoost | https://github.com/dmlc/xgboost | TabM | https://github.com/OpenTabular/DeepTab |
| CatBoost | https://github.com/catboost/catboost | ModernNCA | https://github.com/OpenTabular/DeepTab |
| TabNet | https://github.com/dreamquark-ai/tabnet | Trompt | https://github.com/OpenTabular/DeepTab |
| TabTransformer | https://github.com/lucidrains/tab-transformer-pytorch | TabulaRNN | https://github.com/OpenTabular/DeepTab |
| FT-Transformer | https://github.com/lucidrains/tab-transformer-pytorch | MambAttention | https://github.com/OpenTabular/DeepTab |
| TabSeq | https://github.com/zadid6pretam/TabSeq | MambaTab | https://github.com/OpenTabular/DeepTab |
| TANGOS | https://github.com/OpenTabular/DeepTab | NDTF | https://github.com/OpenTabular/DeepTab |
| NODE | https://github.com/OpenTabular/DeepTab | ENODE | https://github.com/OpenTabular/DeepTab |
| SAINT | https://github.com/OpenTabular/DeepTab | ResNetTabular | https://github.com/OpenTabular/DeepTab |
| DeepFM | https://github.com/shenweichen/DeepCTR-Torch | CategoryEmbedding | https://github.com/manujosephv/pytorch_tabular |
| DCN | https://github.com/shenweichen/DeepCTR-Torch | TANDEM | https://github.com/erelnaor3/tandem |
| TabICL | https://github.com/soda-inria/tabicl | LoCalPFN | https://github.com/layer6ai-labs/LoCalPFN |
| BETA | https://github.com/LAMDA-Tabular/BETA | TuneTables | https://github.com/penfever/TuneTables |
| TabPFN-Wide | https://github.com/pfeiferAI/TabPFN-Wide | RealMLP | https://github.com/dholzmueller/pytabkit |
| MLP-PLR | https://github.com/dholzmueller/pytabkit | TabDPT | https://github.com/layer6ai-labs/TabDPT-inference |
| TabPFN v1 | https://github.com/PriorLabs/TabPFN | TabPFN v2 | https://github.com/PriorLabs/TabPFN |
| TabPFN-2.5 | https://github.com/PriorLabs/TabPFN | | |

