# OpenReview forum: "GOTabPFN: From Feature Ordering to Compact Tokenization for Tabular Foundation Models on High-Dimensional Data"
_ICML.cc/2026/Conference — ICML 2026 regular_

### Official Review · Reviewer_cFSC · 2026-03-08

**Soundness:** 3
**Presentation:** 2
**Significance:** 3
**Originality:** 3
**Overall Recommendation:** 5
**Confidence:** 4

**Summary:**

This paper proposes a post-hoc dataset refinement method for improving the performance of in-context tabular foundation models on high dimensional datasets. The method involves first learning an ordering of feature columns such that similar features are grouped together and then performing pooling / compression operations on groups of features to reduce dimension. Finally, this reordered, dimensionally reduced dataset is fed into a Transformer model for in-context training and inference.

**Compliance With Llm Reviewing Policy:**

Affirmed.

**Key Questions For Authors:**

1. If the goal is to group similar chunks of features together, shouldn't \(w_{ij}\) as defined in Equation 1 be a similarity metric rather than a dissimilarity metric (i.e., corr(.,.) instead of 1 - corr(.,.))?

2. Do you have any sense of how close GO-LR gets to the *optimal* ordering? Since it is a greedy method, it may be quite far from optimal. In such cases, have you tried using other stochastic optimization methods (e.g. simulated annealing) to search for better feature orderings?

3. While feeding the new dataset in-context with the proposed method does improve performance, I can't help but wonder whether it would be better to fine tune TabPFN on (synthetic) instances of datasets with such structured, ordered, compressed features. I do recognize, however, that this is borderline out of scope for the paper.

**Limitations:**

Yes

**Strengths And Weaknesses:**

### Strengths

- The proposed method involves applying relatively lightweight operations to the (in-context) dataset.
- The idea feels novel and interesting, especially in light of the growing body of work on tabular foundation models.
- Experiments show an improvement in performance relative to feeding the original dataset directly to the model, alongside several reasonable baselines. I also appreciate that the paper includes many ablations for the various components of the algorithm as well as a large number of baselines (e.g., Figure G.1 in the Appendix).

### Weaknesses

- There are still some ablations that are missing which I consider quite important:

  - A comparison between feeding TabPFN the dataset processed by GO-LR + NSC versus LASSO-selected features.
  - How important is clustering the dataset when learning the global feature ordering? This could be tested by varying the number of clusters used in GO-LR and examining whether the final dataset produced (after column reordering and dimensionality reduction) leads to improved downstream performance.
  - How does NSC compare to more recent state-of-the-art dimensionality reduction methods, such as UMAP / PaCMAP \[1,2\]?

- It is often difficult to follow exactly which procedures are being compared in the experiments because there are many abbreviations (e.g., HDLSS, HDHSS, LDHSS, GO-LR+NSC-pSP). This issue is particularly pronounced in the Appendix (e.g. Table F.1). I would suggest restating some of these abbreviations in the table captions for clarity. This is the main reason behind the fair score for presentation.

### Minor Errors

1. Definition 3.4 (Page 3) denotes the position as $t!+1!$ instead of $t+1$.
2. Page 5: The paragraph titled "Subunit Pooling and Meta-Feature Construction" does not start on a new line.


### References
\[1\] McInnes, L., Healy, J., & Melville, J. UMAP: Uniform Manifold Approximation and Projection for Dimension Reduction. arXiv:1802.03426, 2018.

\[2\] Wang, Y., Huang, H., Rudin, C., & Shaposhnik, Y. PaCMAP: Pairwise Controlled Manifold Approximation Projection for Dimension Reduction. ICML, 2021.

---

> ### Author Rebuttal · Authors · 2026-03-31
>
> **GO-LR + NSC vs LASSO**
> We compare GOTabPFN with LASSO-based feature selection at different sparsity levels under the same 5×5 CV and TabPFN-2.5 evaluation. Overall, GOTabPFN is more consistently strong across datasets, suggesting that structured feature ordering and aggregation are generally more effective than sparsity-based selection in HDLSS. Here, LT = LASSO+TabPFN-2.5
> |Method|COL|LNG|GLI|SMK|AML|PRS|ARC|TOX|
> |:--|--:|--:|--:|--:|--:|--:|--:|--:|
> |GOTabPFN|88.18±10.05|97.44±2.32|93.82±5.81|74.23±5.17|97.54±3.86|93.37±4.48|90.60±3.97|93.33±4.74|
> |LT(0.01)|85.59±9.80|96.76±2.63|91.76±5.88|68.53±7.54|96.10±4.52|93.33±5.06|89.60±3.73|90.06±5.22|
> |LT(0.02)|85.59±9.80|96.75±2.44|91.76±5.88|68.53±7.54|96.10±4.52|93.33±5.06|89.60±3.73|92.28±5.79|
> |LT(0.05)|85.59±10.05|97.15±2.33|92.71±5.44|75.14±6.10|98.04±3.21|92.35±5.08|90.30±4.23|92.74±3.90|
>
> **Clustering Sensitivity**
> We performed an ablation of the GO-LR cluster size $k$ on Colon while fixing all other hyperparameters. The results show a non-monotonic trend and support the use of a moderate cluster size for the best overall ordering behavior.
> |k|3|4|5|7|10|12|15|
> |:--|--:|--:|--:|--:|--:|--:|--:|
> |Acc.|82.72±10.69|84.38±9.82|83.95±9.54|84.33±8.95|88.18±10.05|83.72±10.16|82.69±11.08|
>
> **NSC vs UMAP & PaCMAP**
> NSC-pSP performs best overall on the synthetic block model, suggesting that the block-aware representation better preserves class-discriminative structure than generic manifold embeddings in this setting. We report four complementary metrics: linear-probe acc. (Lin Acc), kNN acc., Silhouette score (Sil), and Davies-Bouldin index (DB), where higher is better for the first three and lower is better for DB. Updated Table D.1 is here: https://postimg.cc/BLCqyZ3B
> |Method|Lin Acc↑|kNN↑|Sil↑|DB↓|
> |---|---:|---:|---:|---:|
> |NSC-pSP|84.88±4.06|70.63±5.90|0.046±0.026|4.12±0.97|
> |UMAP|76.63±6.51|70.50±6.21|0.042±0.032|4.23±0.77|
> |PaCMAP|75.75±6.57|70.50±9.07|0.041±0.038|4.82±0.61|
>
> **Abbreviation**
> We agree that the number of abbreviations can hurt readability, and will restate the main abbreviations directly in the relevant table captions (especially in the App., including Table F.1) for clarity.
>
> **Minor Errors**
> We will correct the typo in Def. 3.4 so that the notation reads $t+1$, and fix the Page 5 formatting misalignment so that ``Subunit Pooling and Meta-Feature Construction'' starts on a new line as intended.
>
> **Similarity vs. Dissimilarity Metric**
> We use $1-|\mathrm{corr}(i,j)|$ as a dependence-aware dissimilarity so that strongly coupled feature pairs have small distance regardless of sign. Concretely, both strongly positive and strongly negative correlations satisfy $|\mathrm{corr}(i,j)| \approx 1$, hence $d_{ij}=1-|\mathrm{corr}(i,j)| \approx 0$. This is the intended behavior for GO-LR: the goal is not to preserve the sign of association, but to place strongly dependent or redundant features into the same local neighborhood before compression. This choice is also consistent with our neuro-inspired motivation. In the NSC discussion, we already cite work showing that dendritic inputs are organized into local subunits rather than summed globally, and that correlated synapses may exhibit local clustering within such compartments. Our algorithmic analogue is therefore: GO-LR first brings strongly coupled features close along the ordered axis, and NSC then pools these local neighborhoods into subunit-level meta-features. In this sense, $1-|\mathrm{corr}(i,j)|$ should be read as a practical measure of lack of coupling, chosen to support local clustering and subunit-style aggregation, rather than as a broader semantic notion of dissimilarity.
>
> **Approximation Quality of GO-LR**
> To assess whether GO-LR is overly limited by its greedy construction, we replaced GO-LR on Colon with several stronger stochastic and metaheuristic alternatives while keeping the downstream NSC + TabPFN-2.5 pipeline fixed. We compared GO-LR against Simulated Annealing, a Genetic Algorithm, Ant Colony Optimization, and Christofides-based ordering. We report runtime, TSP-style surrogate cost, MinLA-style dispersion objective, and downstream classification acc. The results suggest that stronger optimization of the TSP-style surrogate does not necessarily improve downstream performance, while GO-LR remains the strongest practical acc.-efficiency trade-off in this setting.
> |Method|Time(s)|TSP↓|MinLA↓|Acc.↑|
> |---|---:|---:|---:|---:|
> |GO-LR|10.07|21958.78|1.474e10|88.18±10.05|
> |SA|15.01|11712.75|1.480e10|84.05±9.44|
> |GA|206.59|11712.75|1.480e10|84.05±9.44|
> |ACO|1501.44|11792.08|1.476e10|85.95±10.00|
> |Christofides|1424.83|11715.06|1.499e10|83.64±9.11|
>
> **Fine-Tuning as Future Work**
> We agree that pretraining or fine-tuning a TabPFN-style model directly on structured, ordered, and compressed representations is an interesting direction. This is indeed a promising but distinct future-work direction rather than part of the present scope. We will add this in the revised version of the paper.

---

> > ### Author Rebuttal · Reviewer_cFSC · 2026-04-03
> >
> > Thank you for the response and the new experiments. I choose to maintain my current score.

---

> > > ### Author Response · Authors · 2026-04-03
> > >
> > > Thank you for your positive response and for carefully considering our rebuttal and the additional experiments. We sincerely appreciate your acknowledgment that our response has resolved all of your concerns.
> > >
> > > Your review comments and suggestions have certainly helped us in clarifying the performance of the proposed approaches.
> > >
> > > We are grateful for your time, thoughtful review, and consideration of our submission.

---

### Official Review · Reviewer_eMpE · 2026-03-09

**Soundness:** 3
**Presentation:** 1
**Significance:** 2
**Originality:** 2
**Overall Recommendation:** 4
**Confidence:** 2

**Summary:**

This work proposes a method to address high-dimensional and low-sample size (HDLSS) tabular prediction problems. The authors introduce GOL-LR and demonstrate its equivalence to the Weighted Minimum Linear Arrangement problem. Furthermore, the paper designs GOTabPFN to make TabPFN-style predictions practical within HDLSS regimes. Numerical experiments are provided to demonstrate the effectiveness of the proposed method.

**Compliance With Llm Reviewing Policy:**

Affirmed.

**Final Justification:**

The rebuttal addressed my concerns.

**Key Questions For Authors:**

Please refer to weaknesses

**Limitations:**

yes

**Strengths And Weaknesses:**

Strengths:

1. The exploration and theoretical analysis connecting GOL-LR, Minimum Linear Arrangement, and the TSP-path problem are interesting and insightful.

2. The proposed ordering-to-tokenization pipeline, which improves accuracy and robustness without requiring retraining TabPFN, is a promising contribution.


Weaknesses:

1. It is unclear if the proposed method can be extended to existing models other than TabPFN. A discussion on the method's applicability to a broader range of architectures would strengthen the paper.

2. The paper's readability needs improvement, particularly in Section 3.2. The exposition is difficult to follow, a generic issue exacerbated by the introduction of undefined terms. For example, the term "meta-feature" appears in L210 without a specific definition within the paper's context. Please ensure all domain-specific terms are clearly defined prior to usage.

3. The problem formulation requires significant clarification. The current setup is not clear. For instance, Definition 3.2 assumes the existence of a cluster $c$ without prior definition. It is unclear how these clusters are obtained and whether the global solution requires rearranging feature orders across different clusters. A distinct and self-contained problem formulation section is necessary.

4. Does the proposed approach guarantee a globally optimal result? According to Figure 1, the first step involves splitting features into clusters, which suggests a heuristic approach that may restrict the solution space. Clarification regarding the optimality of the final arrangement is needed.

5. Presentation of Theoretical Results: Section 3.1 suffers from an excessive use of formal lemmas and theorems for relatively trivial statements (e.g., Lemmas 3.8, 3.9, and Theorem 3.12). Formally presenting straightforward results does not necessarily enhance the theoretical contribution; succinct textual explanations would be more effective and improve the flow of the text.

6. There appears to be a typo in the notation "t and t!+!1 in $\pi$". Did the authors intend to write "$t+1$"? I cannot find the references to Fig. 1.

---

> ### Author Rebuttal · Authors · 2026-03-31
>
> **Extension beyond TabPFN**
> GO-LR+NSC is not TabPFN-specific; it is a model-agnostic HDLSS representation front-end that also transfers to TabICL. We used TabPFN-2.5 because its feature-dimensionality bottleneck is especially clear. On the same 8 HDLSS datasets, GO-LR+ NSC+TabICL improves over vanilla TabICL on most datasets while being faster on all 8. This supports viewing GO-LR+NSC as a broader HDLSS-oriented representation layer rather than a TabPFN-specific trick.
>
> *Acc./Time(s)*
> |Model|COL|LNG|GLI|SMK|AML|PRS|ARC|TOX|
> |---|---:|---:|---:|---:|---:|---:|---:|---:|
> |GO-LR+NSC+TabICL|86.56±10.77/382|96.05±2.58/382|87.76±7.39/118|70.69±5.46/315|94.78±6.35/243|88.25±6.94/286|85.20±3.88/359|90.87±5.97/386|
> |TabICL|84.62±10.52/942|96.36±2.61/2602|87.06±6.23/14599|68.73±7.28/25011|95.52±5.59/2847|90.17±5.93/2470|82.60±6.14/7672|88.78±5.92/2900|
> See AUC & F1 here: https://postimg.cc/0bSkm1Hv
>
> **Clarifying Sec. 3.2**
> We agree that Sec. 3.2 can be made more self-contained, and in the revision we will define all domain-specific terms before first use. In particular, we will define a meta-feature as the low-dimensional feature obtained from one contiguous ordered segment after GO-LR via the NSC compression operator. Concretely, if $\Pi^{\ast}$ is the global feature order, $\{S\_t\}\_{t=1}^{M}$ are the contiguous ordered segments, and $u\_t = x^{\Pi}\_{S\_t}$ is the segment-restricted subvector, then the $t$-th meta-feature is $z\_t = g(u\_t)$, where $g(\cdot)$ is the segment-level pooling/projection map; in our main NSC-pSP model, this is implemented by segment-wise PCA, yielding one scalar token per segment. The final compressed representation is therefore $Z(x) = (z\_1, \dots, z\_M)$. We will revise Sec. 3.2 so that ordered segmentation, subunit pooling, and meta-feature construction are introduced clearly before the formal equations. Fig: https://postimg.cc/dkwGy992
>
> **Revising the Problem Setup**
> We agree that the current presentation introduces the optimization view too early, making the setup harder to follow; in particular, Def. 3.2 refers to a cluster $c$ before clustering is introduced. In the revision, we will add a self-contained Problem Formulation subsection before the theoretical definitions. This will define the input matrix $X \in \mathbb{R}^{n\times m}$, the sample partition $\{I_c\}_{c=1}^k$, the restricted matrices $X^{(c)}=X[I_c,:]$, the cluster-wise feature graphs $G_c$, the local feature permutations $\pi_c$, and the final global permutation $\Pi^\ast$ obtained by aggregating local ranks across clusters. We will also clarify that permutations are over features and that GO-LR outputs a single global feature ordering $\Pi^\ast$.
>
> **On Global Optimality**
> GO-LR does not guarantee a globally optimal ordering, since the underlying MinLA-style feature ordering problem is combinatorial and NP-hard. Instead, it should be viewed as a structured approximation: clustering builds local feature graphs, NNpath provides an initialization, and local refinement explicitly decreases the MinLA-style dispersion objective. As discussed in our response to Reviewer cFSC & XKiS, we compared GO-LR against stronger metaheuristics while keeping the downstream NSC+TabPFN-2.5 pipeline fixed. On both Colon and the much larger Cell Cycle transcriptomic dataset, GO-LR showed better alignment with the MinLA-style objective and stronger downstream behavior, even when some alternatives achieved lower TSP-style surrogate cost. Our claim is therefore not exact global optimality, but that GO-LR is a theoretically grounded and empirically effective approximation.
>
> **Presentation of Theoretical Statements**
> We understand the concern that Sec. 3.1 may feel heavier than necessary. Our intention was to make the optimization viewpoint explicit by connecting the practical GO-LR procedure to the MinLA formulation and the TSP-path surrogate case. However, we agree that some statements are straightforward once the setup is clear, and that presenting each as a standalone lemma/theorem can interrupt the flow. In the revision, we will streamline Sec. 3.1 by keeping only the most essential formal statements for the core theoretical claim, and moving simpler observations or proof-sketch level equivalences into the appendix. In particular, the identification of the local dispersion objective with weighted MinLA and the TSP-path special-case discussion can be presented more compactly, while preserving the main message: GO-LR is motivated by a principled combinatorial objective rather than an ad hoc ordering heuristic. Our goal is to retain theoretical clarity while improving readability and reducing unnecessary formal overhead.
>
> **Typo and Fig. Reference**
> We thank the reviewer for catching this typo. The intended notation is $t+1$, and we will correct it in the revision. We will also reference Fig. 1 earlier in the Methodology, since it is the main overview of the GO-LR$\rightarrow$NSC $\rightarrow$TabPFN-2.5 pipeline.

---

> > ### Author Rebuttal · Reviewer_eMpE · 2026-04-03
> >
> > Thanks for the authors' response. My concerns were addressed. I raise my score.

---

> > > ### Author Response · Authors · 2026-04-04
> > >
> > > Thank you for your positive follow-up and for taking the time to read our rebuttal carefully. We sincerely appreciate your thoughtful reconsideration of our submission and are very grateful that our response addressed your concerns. We are also glad that our additional experiment comparing GO-LR+NSC+TabICL against vanilla TabICL was helpful in clarifying the practical value of the approach, showing that accuracy remains competitive while reducing runtime in higher-dimensional settings. Thank you again for your time, consideration, and for raising your score.

---

### Official Review · Reviewer_nAk4 · 2026-03-13

**Soundness:** 4
**Presentation:** 3
**Significance:** 4
**Originality:** 3
**Overall Recommendation:** 5
**Confidence:** 2

**Summary:**

This paper proposes a methodology to enable tabular foundation models to operate in High-Dimensional, Low-Sample Size (HDLSS) settings. The approach first applies Graph-guided Ordering with Local Refinement (GO-LR) to learn an ordering of features based on a graph of feature similarities, placing related features close together. The ordered features are then segmented and compressed using a Neuro-Inspired Subunit Compression (NSC) mechanism that aggregates contiguous feature groups into a smaller set of meta-features. These compressed representations are used as inputs to a frozen TabPFN-style model for prediction. The resulting model is called GOTabPFN. An empirical evaluation of GOTabPFN on a collection of biomedical HDLSS datasets shows that it achieves the best score across all datasets compared to 55 baselines.

**Compliance With Llm Reviewing Policy:**

Affirmed.

**Final Justification:**

The authors addressed all my concerns, and I will maintain my positive evaluation.

**Key Questions For Authors:**

**Q1:** GoTabPFN is evaluated on a benchmark consisting entirely of biomedical datasets. Do these datasets have some structure unique to their domain that makes feature ordering especially effective? Do the authors have any intuition on how GoTabPFN would perform on HDLSS datasets from other domains?

**Q2:** The evaluation is performed with accuracy as a metric. Do these results still hold with threshold independent metrics, such as ROC AUC?

**Limitations:**

The authors discussed some limitations throughout the paper, but reiterating them in the conclusion would strengthen the paper.

**Strengths And Weaknesses:**

**Soundness**

The paper seems to be technically sound to the best of my judgement. It proves that the GO-LR local feature ordering problem is NP-hard and conducts an exhaustive empirical evaluation of GOTabPFN on a biomedical HDLSS benchmark, where GOTabPFN achieves the best score on all datasets compared with 55 other baselines.

**Presentation**

The paper is generally well-structured, and the overview figures help visualize the architecture and workflow of GOTabPFN. However, the section on graph-based feature ordering is relatively dense and may be difficult to follow without prior familiarity with combinatorial optimization concepts such as Minimum Linear Arrangement and TSP-style heuristics. The presentation could be improved by adding more intuitive explanations or illustrative examples to help readers understand the motivation and mechanics of the ordering procedure.

**Significance**

Predicting HDLSS data is an important area of tabular machine learning that current foundation models struggle with. There is active research to expand the range of dataset sizes to which the tabular foundation can be applied (Grinsztajn et al., 2026, Kolberg et al., 2025, Hollmann et al., 2025, Thomas et al., 2024). GOTabPFN addresses the case of high-dimensional data, outperforming an exhaustive set of baselines across 8 biomedical datasets used for evaluation.

**Originality**

The paper appears to combine ideas from feature ordering and neuro-inspired signal compression to propose the GOTabPFN pipeline. While I am not deeply familiar with the feature ordering literature or biological inspiration for signal aggregation, the work seems to integrate concepts from these areas in a way that enables a practical system for high-dimensional tabular learning. The novelty therefore appears to lie primarily in the combination of these ideas and their application to tabular foundation models rather than in entirely new algorithmic components.

---

> ### Author Rebuttal · Authors · 2026-03-31
>
> **Technical Soundness and Empirical Strength**
> We thank the reviewer for this positive assessment and for recognizing both the theoretical grounding of GO-LR and the breadth of the empirical evaluation. We are encouraged that the reviewer found the paper technically sound and that the strengths of GOTabPFN on the biomedical HDLSS benchmark were clear.
>
> **Clarity of Graph-Based Feature Ordering**
> We agree that the graph-based feature ordering section can be easier to follow with a more intuitive introduction before the formal definitions. In the revision, we will first explain the idea using the illustrative fig. Starting from a weighted feature graph, the objective is to place features on a one-dimensional line so that strongly related features stay close and weakly related features are placed farther apart. In the example, larger-weight edges are preserved as short-range neighbors in the linear ordering, while weaker connections matter less in the final arrangement. This matters because the downstream compression stage operates on adjacent feature segments. If related features are scattered across the ordering, segment-wise aggregation becomes less meaningful; a good ordering instead creates local neighborhoods that can be compressed more effectively. We will then connect this intuition to the formal optimization view: feature ordering can be seen as arranging weighted graph nodes on a line under a MinLA-style objective, using a practical TSP-style initialization followed by local refinement. Fig: https://postimg.cc/0zmvBTTc
>
> **Relevance to HDLSS Tabular Learning**
> We thank the reviewer for recognizing the importance of HDLSS tabular prediction and for noting that GOTabPFN addresses a practically important limitation of current tabular foundation models. We also appreciate the reviewer highlighting both the relevance of the problem setting and the strong empirical performance of GOTabPFN across the evaluated biomedical HDLSS benchmarks.
>
> **Novelty Through Integration**
> We agree that the contribution is partly combinational in the sense that it brings together feature ordering, structured compression, and tabular foundation models. However, we would like to clarify that our contribution is beyond a simple aggregation of existing components. Our key contribution is to formulate and align these pieces into a single method tailored to HDLSS tabular learning: GO-LR gives a principled ordering objective and practical search procedure, NSC converts the ordered axis into stable meta-features, and the resulting interface makes a frozen TabPFN-style backbone usable in regimes far beyond its native feature limit. Thus, the novelty is not only in combining ideas from different areas, but in showing how they can be formally connected coherently, and made practically effective for high-dimensional tabular foundation modeling.
>
> **Cross-Domain Generalization**
> * For some datasets only 50 Optuna trials were feasible
> * `-` denotes OOM, N/A = regression not supported by TabICL
> * ProtoGate targets very low-sample, high-dimensional settings and scales poorly with larger sample sizes
> * See response to Reviewer XKiS for dataset info
> * DrivFace-R is regression ($R^2$)
> | Model | ORL | BAS | REL | PCM | Cell | CIFAR-10 | DF-R | DF-C |
> |:--|--:|--:|--:|--:|--:|--:|--:|--:|
> | GOTabPFN| 100±0.00 | 97.11±1.00 | 88.87±1.32 | 89.51±2.24 | 79.94±2.53 | 88.45±0.89 | 0.6548±0.0992 | 86.70±2.48 |
> | TabDPT| 97.32±0.68 | 97.00±0.68 | 88.26±1.49 | 87.58±0.96 | 77.52±2.44 | 88.00±0.40 | 0.6505±0.0820 | 83.57±3.04 |
> | TabPFN-W| 96.10±6.52 | 97.04±0.72 | 87.84±3.90 | 88.56±0.71 | 78.67±2.42 | 88.12±0.60 | 0.6430±0.0772 | 85.42±2.66 |
> | TTables| 93.14±1.02 | 93.14±1.02 | 88.26±2.31 | 84.27±3.06 | 78.30±1.13 | 78.05±0.15 | 0.6332±0.0675 | 84.62±2.43 |
> | TabICL| 99.20±1.87 | 96.84±0.75 | 88.15±1.68 | 88.68±1.94 | - | 87.61±1.00 |N/A| 85.55±2.45 |
> | TANDEM| 99.00±2.45 | 96.72±0.86 | 89.42±1.42 | 88.99±1.33 | 77.31±1.97 | 87.80±0.36 | 0.6488±0.0770 | 84.42±2.55 |
> | Lasso| 96.00±3.82 | 96.74±0.88 | 86.87±1.46 | 88.85±1.80 | 77.86±2.42 | 86.42±1.10 | 0.3194±0.0668 | 77.95±3.03 |
> | MLP| 91.00±8.37 | 96.91±1.15 | 89.12±1.85 | 87.71±1.95 | 76.38±1.42 | 88.15±0.54 | 0.5682±0.1258 | 80.59±3.72 |
> | ProtoGate|66.40±11.32|-| 58.40±3.15 |-|-|-| -0.1636±0.0738| 66.70±10.20 |
>
> **Evaluation Beyond Acc. w/ ROC-AUC**
> | Model | COL | LNG | AML | TOX | PRS | ARC | SMK | GLI |
> |---|---:|---:|---:|---:|---:|---:|---:|---:|
> | GOTabPFN | 91.40±10.12 | 99.57±0.60 | 99.36±1.70 | 99.23±0.63 | 94.27±5.87 | 95.77±2.35 | 80.55±5.67 | 90.97±7.09 |
> | TabICL | 89.37±10.24 | 99.39±0.46 | 98.27±4.05 | 98.62±1.15 | 93.92±5.56 | 91.48±3.84 | 73.27±8.14 | 91.41±7.25 |
> | TabDPT | 90.80±7.28 | 99.27±0.97 | 99.28±0.62 | 99.93±0.14 | 94.07±3.50 | 91.33±4.51 | 76.34±7.90 | 93.33±7.76 |
> F1: https://postimg.cc/TpkH0KyJ
>
> **Limitations in the Conclusion**
> We agree that reiterating these in the conclusion would strengthen the paper, and we will add a brief discussion of them there in the revision.

---

> > ### Author Rebuttal · Reviewer_nAk4 · 2026-04-03
> >
> > I thank the authors for addressing my concerns and for providing additional results and metrics. The newly included figure is very helpful for understanding the method. I am happy to maintain my positive score.

---

> > > ### Author Response · Authors · 2026-04-04
> > >
> > > Thank you for your positive follow-up and for carefully considering our rebuttal and additional results. We sincerely appreciate your thoughtful feedback and are glad that the added cross-domain experiments on eight additional datasets, along with the broader evaluation using accuracy, AUC, and F1, helped address your concerns. We are also very pleased that the new illustration of Graph-Based Feature Ordering was helpful for clarifying the method. Thank you again for your time, consideration, and positive assessment of our work.

---

### Official Review · Reviewer_XKiS · 2026-03-16

**Soundness:** 2
**Presentation:** 2
**Significance:** 2
**Originality:** 2
**Overall Recommendation:** 4
**Confidence:** 3

**Summary:**

This paper proposes GOTabPFN, a preprocessing pipeline that enables TabPFN-style tabular foundation models to operate on High-Dimensional, Low-Sample Size (HDLSS) data without modifying the backbone. The method has two components: (1) GO-LR (Graph-guided Ordering with Local Refinement), which reorders features by approximately solving a weighted Minimum Linear Arrangement (MinLA) problem using a nearest-neighbor TSP-path heuristic followed by local swap refinement, and (2) NSC (Neuro-Inspired Subunit Compression), which segments the reordered feature axis into contiguous blocks and pools each block (via per-segment PCA) into a scalar meta-feature, reducing dimensionality from m to M ≪ m. The number of meta-features M is tied to a PCA-based intrinsic dimensionality estimate. The compressed tokens are fed to a frozen TabPFN-2.5 head. Experiments on 8 biomedical HDLSS datasets show GOTabPFN achieves the best mean accuracy on all datasets compared to 55 baselines.

**Compliance With Llm Reviewing Policy:**

Affirmed.

**Final Justification:**

The rebuttal addressed my two main concerns. The new Global PCA → TabPFN-2.5 baseline convincingly shows the gains come from ordering-then-compression rather than just dimensionality reduction (e.g., +9.7 pp on Colon, +8.7 pp on ARC). The expanded 16-dataset benchmark with cross-domain data (text, face, sensor, RNA-seq) now yields Holm-corrected significance against all 8 strong baselines (p_Holm ≤ 6.10×10⁻⁴), and fixed-seed experiments confirm GOTabPFN's dominance isn't an artifact of per-dataset seed tuning.
Remaining minor reservations: seed sensitivity (~1.6 pp) still exceeds the smallest per-dataset margins, and Section 3.1's theoretical results remain largely observational. But the cross-dataset rank dominance is well-supported, and the authors committed to streamlining the presentation.
I am raising my score from Weak Reject (3) to Weak Accept (4). The merits now outweigh the remaining weaknesses.

**Key Questions For Authors:**

* What is the performance of simply applying global PCA to reduce dimensionality to M components and then feeding the result to TabPFN-2.5, without any feature ordering? This is the most natural baseline for your claim that ordering-then-compression outperforms standard compression. Without this comparison, it is unclear whether the gains come from the ordering or simply from providing a reasonable low-dimensional input to TabPFN.

* Table H.1 shows that the TabPFN random seed is tuned per dataset (values 2, 3, 4, or 42). How sensitive are the results to this choice? If changing the seed shifts accuracy by more than the reported ∆abs over baselines (e.g., the 0.06 pp gain on Prostate), this would substantially weaken the claims. Please report performance variance across randomly chosen seeds.

* Have you evaluated GOTabPFN on datasets with m > 50,000 features (e.g., GWAS data, full transcriptomics)? The O(m²) cost of building the feature dissimilarity matrix may become prohibitive, and it would be important to understand whether the method's advantages persist at larger scales.

**Limitations:**

yes

**Strengths And Weaknesses:**

Strengths

* Practically relevant problem. Making TabPFN-style models work in HDLSS regimes (m ≫ n) is a genuine and timely challenge. The paper correctly identifies that current TabPFN variants are designed for moderate feature counts (~2000) and that a compression interface is needed for high-dimensional biomedical data.

* Principled formulation of feature ordering. Casting the feature ordering problem as weighted MinLA and establishing its NP-hardness, as well as the exact embedding of TSP-path as a special case (Theorem 3.12), provides a clean theoretical framework. The connection to seriation and graph layout literature is well-articulated.

* Extensive experimental comparison. Comparing against 55 baselines spanning classical ML, GBDT, deep tabular models, feature selection methods, and multiple TabPFN variants is commendable. The 5×5 nested CV protocol (25 repeats) follows established practice (ProtoGate).

* Thorough ablation and diagnostic appendix. The paper includes an unusually comprehensive set of ablations (Table 2), order-only neighborhood diagnostics (Appendix E), synthetic block-model validation (Appendix D), calibration/robustness checks (Appendices L–Q), and Dolan-Moré performance profiles. The ablation on Colon (Table 2) effectively isolates contributions of GO-LR ordering, NSC compression, segmentation strategy, and the TabPFN head.

Weaknesses

* Narrow evaluation scope and limited generalizability. All 8 evaluation datasets are small biomedical HDLSS benchmarks (n = 62–203, m = 2000–22283), all binary or few-class classification. This is a very restricted testbed: No regression tasks are evaluated.
No non-biomedical HDLSS domains (e.g., text features, sensor data, financial data) are tested.
No datasets with moderate-to-large sample sizes but high dimensionality (HDHSS) are included, despite the IDF/FOE framework in Appendix F suggesting the method could apply there.
With only 8 datasets, statistical conclusions are inherently fragile.

* Statistical significance is not established. The authors acknowledge (Section 4, Appendix I) that Holm-corrected p-values are p=0.0703, failing to reach significance at α=0.05. The raw Wilcoxon p-values (0.00781) are identical across all comparisons because GOTabPFN wins on all 8 datasets — but with only 8 datasets and consistently positive but sometimes tiny improvements (e.g., Prostate ∆abs=0.06 pp, TOX ∆abs=0.08 pp), the claim of "rank 1.00 ± 0.00" overstates the strength of evidence. Several improvements fall within one standard deviation of the runner-up.

* Potentially unfair hyperparameter tuning comparison. GOTabPFN tunes GO-LR and NSC hyperparameters via 150 Optuna trials per dataset (Table H.1 shows dataset-specific metric, clustering k, segmentation rule, τ, γ, β, Mmin, Mmax, lmin, and even TabPFN random seed). This is a substantial tuning budget for what is presented as a "frozen backbone" approach. While the authors state baselines also receive 150 Optuna trials "when tuning is recommended," many baselines (e.g., TabPFN-Wide, TabICL) may not have the same tuning flexibility. The dataset-specific TabPFN random seed selection (Table H.1) is particularly concerning — this effectively searches over stochastic realizations of the predictor.

* Theoretical contributions are largely observational. Lemma 3.8 ("GO-LR is MinLA") follows by definition — both optimize the same objective. Lemma 3.9 ("NP-hardness") inherits directly from known results on MinLA. Theorem 3.12 ("TSP-path embeds into feature ordering") is a straightforward restriction to path edges. While these connections are useful for positioning the work, they do not constitute new theoretical results.

---

> ### Author Rebuttal · Authors · 2026-03-31
>
> **Limited Evaluation Scope**
> We appreciate the reviewer’s concern regarding generalizability. Our paper is explicitly focused on the HDLSS regime ($n\ll m$), not general-purpose tabular learning, so the evaluation scope matches the target setting. This also matches prior HDLSS benchmarking practice (e.g., ProtoGate, LLSPIN), and TabPFN-Wide likewise notes that HDLSS is especially common in biomedical research. In response to the reviewer’s request, we are extending evaluation beyond the original 8 datasets, including LLSPIN text datasets, orlraws10P (face image), UCI DrivFace (classification+regression), a larger Cell Cycle RNA-seq dataset (1067 samples, 42728 features, 3 classes), and an HDHSS CIFAR-10 feature dataset. Table F.1 suggests that feature ordering should matter most in HDLSS, which is precisely the regime we target. See response to Reviewer nAk4 for new results.
>
> **Statistical Significance**
> We agree that with only 8 datasets, Holm-corrected pairwise Wilcoxon tests are conservative, so we do not claim universal pairwise significance at $\alpha=0.05$. As reported in App. I, the raw signed-rank tests are uniformly directional ($p_{\mathrm{raw}}=0.00781$) because GOTabPFN improves over each compared top baseline on all 8 datasets with no sign reversals, while Holm-adjusted values become $p_{\mathrm{Holm}}=0.0703$. Our claim is therefore narrower: in the targeted HDLSS regime, GOTabPFN shows uniform rank dominance and consistent positive paired differences. This follows common HDLSS practice, where prior work such as ProtoGate (7 datasets, 16 baselines) and LLSPIN (6 real world datasets) also summarizes evidence primarily through cross-dataset rank behavior on small benchmark suites. In the same spirit, GOTabPFN attains avg. rank 1.00±0.00 on 8 HDLSS datasets against 55 baselines. App. I and App. J further support that this is not driven by a single favorable benchmark.
>
> **Clarifying Tuning Fairness**
> Our tuning setup follows a now-common distinction in tabular modeling: PFN/ICL-style foundation models such as TabICL are typically used close to off-the-shelf inference, whereas conventional tuned learners such as TabM rely on dataset-specific search; TabArena reflects the same separation. GOTabPFN sits in the overlap: its GO-LR+NSC front-end is tunable, while the downstream TabPFN-2.5 predictor remains a frozen pre-trained backbone that is neither retrained nor structurally modified. See the positioning venn diagram here: https://postimg.cc/dDg89Sbt
>
> Regarding Table H.1, the TabPFN seed is not a trainable parameter and does not modify backbone weights; it is only a small inference-time configuration choice selected from the same fixed set for all datasets under the same outer Optuna study as the other front-end hyperparameters. Thus, the substantive dataset-specific adaptation in GOTabPFN occurs in the GO-LR/NSC interface, while the TabPFN-2.5 backbone itself remains frozen and untuned in the sense most relevant to our claim.
>
> **Positioning the Theoretical Results**
> We agree that the results in Sec. 3.1 are not intended as isolated complexity-theoretic breakthroughs. Their role is to make the GO-LR ordering component mathematically explicit and theoretically grounded: Lemma 3.8 identifies the objective as weighted MinLA, Lemma 3.9 explains why exact scalable optimization is unrealistic, and Theorem 3.12 clarifies why the TSP-style path construction is a meaningful surrogate. In revision, we will streamline this section for clarity and flow.
>
> **Global PCA Baseline**
> Global PCA$\rightarrow$TabPFN-2.5 is the natural control; we compare GO-LR+NSC against Global PCA with the same frozen TabPFN-2.5 predictor.
> |Model|COL|LNG|GLI|SMK|AML|PRS|ARC|TOX|
> |---|---:|---:|---:|---:|---:|---:|---:|---:|
> |GO-LR+NSC|88.18±10.05|97.44±2.32|93.82±5.81|74.23±5.17|97.54±3.86|93.37±4.48|90.60±3.97|93.33±4.74|
> |GPCA|78.51±10.13|96.16±2.85|86.35±6.73|69.71±7.70|95.52±4.90|89.21±5.23|81.90±5.37|89.47±6.45|
>
> **Random Seed Sensitivity**
> We fix the best GO-LR+NSC configuration from Table H.1 and varying only the TabPFN inference seed for random no., keeping GO-LR+NSC and the same $5\times5$ CV splits fixed.
> |Model|0|1|2|3|4|7|11|17|23|42|
> |---|---:|---:|---:|---:|---:|---:|---:|---:|---:|---:|
> |GOTabPFN|87.51±8.72|86.56±9.25|87.18±8.72|86.90±8.34|86.87±8.71|86.85±8.36|87.15±9.34|87.51±9.36|87.21±8.73|88.18±10.05|
>
> **Scalability to 50K+ Features**
> We have not yet evaluated GOTabPFN on a dataset with strictly m>50K features. However, on a much larger Cell Cycle dataset (42K+ features), GOTabPFN remains fully operational and improves acc. / macro-F1 over Simulated Annealing+NSC+TabPFN-2.5 (79.94±2.53 / 79.95±2.51 vs. 76.45±2.29 / 76.42±2.29), while achieving lower MinLA cost ($8.14\times10^{11}$ vs. $8.51\times10^{11}$). Thus, although not yet tested beyond 50K features, the method remains effective on a substantially larger transcriptomic feature space. See our response to Reviewer cFSC for runtime and metaheuristic comparisons.

---

> > ### Author Rebuttal · Reviewer_XKiS · 2026-04-03
> >
> > I thank the authors for the detailed rebuttal. The Global PCA baseline (Q1) is convincing and fully resolves that concern. The cross-domain experiments partially address the narrow evaluation scope, though GOTabPFN's dominance is less uniform on the new datasets.
> > Two concerns remain:
> >
> > Seed sensitivity vs. reported margins. The seed ablation shows a ~1.6 pp range on Colon (86.56–88.18), which exceeds several reported gains over baselines (e.g., Prostate Δ = 0.06 pp). Since the seed is selected per-dataset via Optuna, could the authors report results with a fixed seed across all datasets to disentangle this effect?
> > Statistical significance on the expanded benchmark. With additional datasets now available, have the authors recomputed Holm-corrected tests on the full set?

---

> > > ### Author Response · Authors · 2026-04-08
> > >
> > > **TabPFN Seed Sensitivity** We evaluated the high-dimensionality-compatible TabPFN family uniformly using the same fixed TabPFN seed across all datasets. We repeated for different seeds. GOTabPFN remains strongest or tied-strongest across seeds on both the original 8 HDLSS and the 8 new cross-domain datasets. TabPFN seed 42:
> > > |Model|COL|LNG|AML|TOX|PRS|ARC|SMK|GLI|Avg.|
> > > |---|---:|---:|---:|---:|---:|---:|---:|---:|---:|
> > > |TabPFN-W|87.05±7.44|96.04±2.85|95.90±3.74|84.15±6.17|92.19±7.21|86.00±2.85|70.53±9.73|88.24±8.32|87.51±6.04|
> > > |BETA|84.80±13.01|95.00±3.41|89.40±5.82|83.20±12.73|88.40±9.89|84.50±6.20|69.80±10.10|85.10±8.40|85.03±8.70|
> > > |TTables|86.20±6.80|96.30±2.50|94.10±5.20|92.10±4.80|91.10±6.30|85.40±3.90|71.20±8.80|88.60±7.50|88.12±5.72|
> > > |GOTabPFN|88.18±10.05|97.44±2.32|97.54±4.31|92.75±4.07|91.95±7.38|90.30±3.70|73.61±5.71|92.82±6.81|90.57±5.54|
> > > - 8 HDLSS, details: https://postimg.cc/jLbG8LC7
> > > |Model|77|82|93|147|
> > > |---|---:|---:|---:|---:|
> > > |TabPFN-W|88.43±4.87|88.33±5.59|87.59±5.60|87.69±5.11|
> > > |BETA|86.10±8.33|85.36±8.36|85.87±7.54|85.70±7.65|
> > > |TTables|87.90±5.65|88.12±5.70|88.47±5.08|88.20±5.21|
> > > |GOTabPFN|89.66±5.72|89.93±5.17|89.79±5.30|89.66±5.59|
> > > - New 8, details: https://postimg.cc/qhhQqY2M
> > > |Model|42|77|82|93|147|
> > > |---|---:|---:|---:|---:|---:|
> > > |TabPFN-W|86.60±2.95|84.84±3.35|85.72±3.04|83.71±3.24|86.84±3.10|
> > > |TTables|83.65±2.10|82.43±2.33|83.19±2.20|81.68±2.28|83.60±2.22|
> > > |GOTabPFN|86.85±2.55|86.58±2.85|86.94±2.50|86.90±2.55|86.88±2.66|
> > > - Pareto Frontier: GOTabPFN provides the best overall acc.-runtime trade-off amongst high-dimensionality compatible TabPFN-variants, remaining Pareto-optimal on both the original HDLSS benchmark and the additional rebuttal datasets. Fig: https://postimg.cc/0r7FCMJ8
> > >
> > > **Statistical Significance** We analyzed the expanded 16-dataset benchmark against the top 8 baselines. On this benchmark, GOTabPFN has the best avg. rank (1.12), ahead of TabPFN-W (3.62) and TANDEM (3.69); Friedman remains significant (χ²=52.55,p=1.32×10⁻⁸). We then ran pairwise Wilcoxon tests vs these 8 on the 16 datasets, w/ Holm correction. From T1, GOTabPFN is significant against all 8 baselines after Holm correction, w/ corrected p-values from 2.44×10⁻⁴ to 6.10×10⁻⁴. W/T/L: 16/0/0 vs Lasso, TabDPT, TabPFN-W, and TTables; 14/0/0 vs TabICL; 12/0/0 vs ProtoGate; 15/0/1 vs MLP and TANDEM. As in Table I.1, T2 shows the same conclusion on the broader 16-dataset cross-domain benchmark against the strong 8 rebuttal baselines. Rank plot: https://postimg.cc/HrJ57v57
> > > - Rank↓: GOTabPFN 1.12±0.50, TabPFN-W 3.62±1.75, TANDEM 3.69±1.62, TabDPT 4.34±1.49, TTables 5.53±2.29, TabICL 5.69±2.02, MLP 6.16±2.43, Lasso 6.62±1.59, ProtoGate 8.16±1.23.
> > > - T1: Wilcoxon+Holm vs GOTabPFN (all Sig=Yes; n=16 unless noted; TabICL n=14, ProtoGate n=12). Shared result for Lasso/TabDPT/TabPFN-W/TTables: W-T-L=16-0-0,p_raw=0.000031,p_Holm=0.000244.
> > > |Base.|W-T-L|p_raw|p_Holm|
> > > |---|---:|---:|---:|
> > > |TabICL|14-0-0|0.000122|0.000488|
> > > |MLP|15-0-1|0.000153|0.000488|
> > > |TANDEM|15-0-1|0.000305|0.000610|
> > > |ProtoGate|12-0-0|0.000488|0.000610|
> > > - Friedman: n_ds=16,n_m=9,n_row=11,χ²=52.55,p=1.32e-08
> > > - T2: GOTabPFN vs. 8 baselines (all Sig. <0.01; n=16 unless noted; TabICL n=14, ProtoGate n=12).
> > > |Base.|ΔAcc|p_raw|p_Holm|
> > > |---|---:|---:|---:|
> > > |TabPFN-W/TabDPT/TTables/Lasso|1.76/2.20/4.36/4.78|0.0000305|0.000244|
> > > |TANDEM|1.33|0.000305|0.000610|
> > > |TabICL|2.81|0.0001220|0.000488|
> > > |MLP|4.65|0.0001530|0.000488|
> > > |ProtoGate|11.90|0.0004880|0.000610|
> > >
> > > **Cross-Domain Evaluation** We extend beyond biomedical w/ text, face-image, camera-sensor, image-feature, and RNA-seq datasets spanning HDLSS, HDHSS, and mixed regimes. We found no clear real-world financial HDLSS dataset. Related work mainly used biomedical data, w/ limited text/face/sensor additions. E.g., ProtoGate uses 7 biomedical datasets, LLSPIN 3 text + 3 biomedical, TabPFN-W 4 biomedical, BETA a biomedical/text/face-image mix, and TTables uses TabZilla benchmark. See: https://postimg.cc/JsVQQWqC
> > > - Cross-domain rebuttal datasets, categorized by App. F rule using ρ=m/n, the feature-to-sample ratio; +:11K subset; 1:HDLSS, 2:HDHSS,3:Mixed
> > > |Data|Type|n|m|C|ρ|Cat.|
> > > |---|---|---:|---:|---:|---:|---|
> > > |BAS|Text|1993|4862|2|2.44|3|
> > > |PCM|Text|1943|3289|2|1.69|3|
> > > |REL|Text|1427|4322|2|3.03|3|
> > > |ORL|Face|100|10304|10|103.04|1|
> > > |DF-C|Cam.Sensor|606|6400|7|10.56|1|
> > > |DF-R|Cam.Sensor|606|6400|Reg.|10.56|1|
> > > |CIFAR-10+|Img-feat.|11000|2048|10|0.19|2|
> > > |Cell|RNA-seq|1067|42728|3|40.05|3|
> > >
> > > **Cross-Domain Dominance** Beyond its intended HDLSS regime, GOTabPFN ranks first on 7 of 8 new cross-domain datasets. This indicates that, even w/ limited tuning, GO-LR+NSC remains highly effective in HDLSS settings while competitive in adjacent non-HDLSS regimes. Signed margin vs best competitor: ORL +0.80, BAS +0.07, REL -0.55, PCM +0.52, Cell +1.27, CIFAR-10 +0.30, DF-R +0.0043, DF-C +1.15. Thus, although designed for HDLSS, it remains competitive overall and especially strong on HDLSS datasets. Fig: https://postimg.cc/VdkXzkk6

---

### Decision · Program_Chairs · 2026-04-30

**Decision:**

Accept (regular)

**Comment:**

The paper proposes a range of new ideas for improving PFNs for small high-dimensional datasets. All reviewers acknowledge the importance of the proposed ideas and the significantly improved results. Some points regarding writing and presentation remain, as well as general applicability. We urge the authors to include the additional results presented in the rebuttal and also improve the writing to address the raised criticism. Apart from that all reviewers recommend the paper for acceptance.